# COBRA: Contribution-Based Bayesian Rank Allocation for Parameter-Efficient Fine-Tuning

Hongcheng Ding [* 1]   Xuanze Zhao [* 1]   Xuanhuang Liu [2]   Jing Jin [3]   Shamsul Nahar Abdullah [3]
Deshinta Arrova Dewi [3]

## Abstract

Full fine-tuning of large language models (LLMs) incurs prohibitive computational and storage costs. Parameter-efficient fine-tuning (PEFT) addresses this limitation, with Low-Rank Adaptation (LoRA) gaining widespread adoption due to its simplicity and zero inference overhead. However, LoRA and its variants typically rely on uniform rank allocation or a single importance metric such as gradient magnitude or output sensitivity to guide rank distribution. This approach fails to recognize that gradient magnitude and output contribution are decoupled properties, leading to suboptimal allocation where critical layers are underprovisioned while less important ones waste capacity. To address this challenge, we propose COBRA, a principled framework integrating dual importance factors for adaptive rank allocation. COBRA operates in three stages: (1) layer conductance attribution quantifies each layer's contribution via path-integral attribution; (2) dualfactor aggregation combines contribution with adaptation demand, producing the Task-Adaptive Layer Conductance (TA-LC) distribution; and (3) Bayesian rank allocation translates this distribution into optimal heterogeneous ranks via variational optimization. Layer conductance provides layer-level interpretability by explicitly quantifying how much each layer contributes to predictions without redundancy, directly aligning with the granularity of rank allocation decisions and enabling principled cross-layer comparison for rank distribution. Experiments across diverse architectures and tasks demonstrate that COBRA consistently outperforms existing methods, achieving up

*Equal contribution [1]Dongfang College, Zhejiang University of Finance and Economics, China [2]SEGi University, Malaysia [3]INTI International University, Malaysia. Correspondence to: Xuanze Zhao <20070032@zufedfc.edu.cn>.

*Proceedings of the 43$^{rd}$ International Conference on Machine Learning*, Seoul, South Korea. PMLR 306, 2026. Copyright 2026 by the author(s).

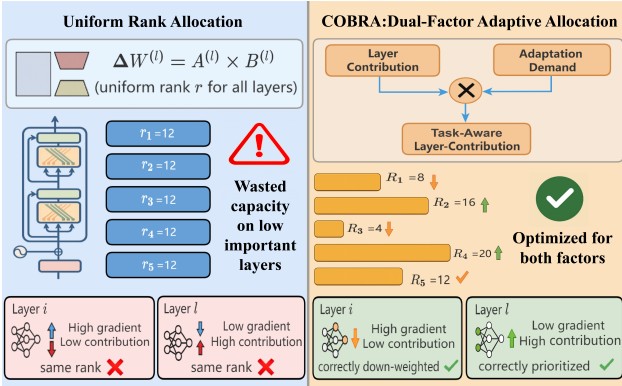

*Figure 1.* **COBRA Framework. Left:** Traditional uniform rank allocation fails to address the misalignment between adaptation demand and layer importance, leading to wasted parameter capacity. **Right:** Our proposed dual-factor approach integrates layer contribution and gradient signals into a unified metric to enable heterogeneous rank allocation, directing resources to layers that are both functionally important and sensitive to adaptation.

to 1.6 points improvement on GLUE and a 6.6% average MSE reduction in high-rank regression regimes under comparable parameter budgets.

## 1. Introduction

LLMs have achieved remarkable performance across diverse tasks (Guo et al., 2025; Minaee et al., 2024; Yang et al., 2024). However, adapting these billion-parameter models to downstream tasks through full fine-tuning incurs prohibitive computational and storage costs (Han et al., 2024). PEFT has emerged as a compelling solution that updates only a small subset of parameters while keeping the pre-trained backbone frozen (Wang et al., 2024; Xin et al., 2024). Among various PEFT approaches, LoRA (Hu et al., 2022) has gained widespread adoption due to its simplicity and zero inference overhead, decomposing weight updates into the product of two low-rank matrices.

LoRA faces a fundamental limitation: the uniform allocation of rank across all layers fails to account for the heterogeneous importance of modules. This limitation has

motivated a series of extensions exploring adaptive rank allocation strategies. AdaLoRA (Zhang et al., 2023) employs SVD-based parameterization with importance-driven singular value pruning to dynamically adjust ranks during training. DoRA (Liu et al., 2024a) decomposes pre-trained weights into magnitude and direction components, enabling more expressive updates. Recent efforts explore gradient-driven allocation (He et al., 2025) and layer-wise adaptive strategies (Shinwari & Usama, 2025; Gu et al., 2025), representing notable progress toward addressing the heterogeneity of layer importance in transformer architectures.

However, existing rank allocation methods typically rely on a single importance metric to guide rank distribution, either gradient magnitude reflecting adaptation demand or output sensitivity measuring prediction influence. These two metrics capture fundamentally different aspects of layer behavior and exhibit only moderate correlation across diverse transformer architectures, with their peak positions differing substantially. This observation reveals two potential failure modes in single-factor allocation: layers with high adaptation demand but low contribution to predictions may receive excessive rank capacity that yields minimal performance gains, while layers with high contribution but moderate adaptation demand may be under-provisioned despite their critical role in model predictions. This decoupling between adaptation demand and output contribution suggests that neither metric alone suffices for principled rank allocation.

To address these challenges, we propose COBRA (COnductance-based Bayesian Rank Allocation), a principled framework that integrates dual importance factors for adaptive rank allocation. COBRA operates in three stages: (1) layer conductance attribution quantifies each layer's contribution to predictions via path-integral attribution, yielding exhaustive and non-redundant importance scores; (2) dual-factor aggregation combines contribution with adaptation demand through multiplicative integration, producing the TA-LC distribution; and (3) Bayesian rank allocation translates this distribution into optimal heterogeneous ranks via variational optimization with TA-LC-informed priors. The key insight underlying COBRA is that layer conductance provides layer-level interpretability by explicitly quantifying how much each layer contributes to predictions without redundancy, directly aligning with the granularity of rank allocation decisions and enabling principled cross-layer comparison for rank distribution. Our main contributions:

- We identify a critical performance gap in existing adaptive LoRA methods stemming from the decoupled relationship between gradient magnitude and output contribution, revealing that single-factor allocation strategies lead to systematic over-provisioning or under-provisioning of critical layers.

- We propose COBRA, a novel framework that leverages

*Table 1.* Gradient magnitude and output contribution across transformer layers, showing substantial variation and misaligned peak positions. More details are provided in Appendix A.1.

| Model | Type | Layers | Gradient Ratio | Contribution Ratio | Contribution Peak | Gradient Peak |
|---|---|---|---|---|---|---|
| RoBERTa-large | Encoder | 24 | 9× | 10× | 15–17 | 21–24 |
| BERT-large | | 24 | 18× | 10× | 15–17 | 21–24 |
| DeBERTaV3-base | | 12 | 11× | 8× | 7–8 | 10–12 |
| LLaMA-2-7B | Decoder | 32 | 18× | 19× | 17–22 | 28–32 |
| GPT-2-medium | | 24 | 15× | 12× | 12–18 | 20–24 |
| T5-base | Encoder-Decoder | 24 | 14× | 11× | D3–D5 | D9–D12 |
| BART-large | | 24 | 17× | 16× | D3–D6 | D9–D12 |

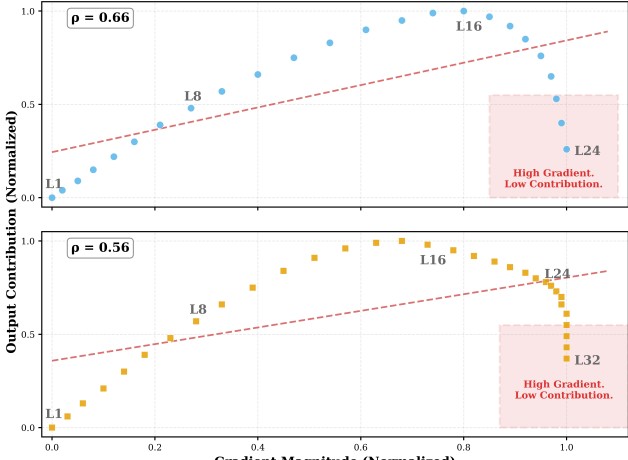

*Figure 2.* Gradient magnitude and output contribution for RoBERTa-large (**upper**) and LLaMA-2-7B (**lower**). Highlighted regions indicate layers that would be over-allocated by single-factor methods. More details are provided in Appendix A.2.

interpretability-guided dual-factor aggregation to produce the TA-LC distribution, combined with Bayesian rank allocation that translates importance priors into optimal heterogeneous rank configurations that are theoretically guided.

- Extensive experiments demonstrate that COBRA consistently outperforms existing methods across different parameter regimes, achieving +1.6 points on GLUE and 6.6% MSE reduction in high-rank regression scenarios ($r = 384$–$512$), revealing that heterogeneous allocation advantages scale with parameter budget.

## 2. Motivation

To understand what factors should inform rank allocation, we examine layer-wise behavior across diverse architectures, revealing two key challenges.

**Challenge #1: Layer Heterogeneity.** Not all transformer layers contribute equally to predictions or require the same degree of adaptation. To quantify this disparity, we measure layer-wise gradient magnitude (reflecting adaptation

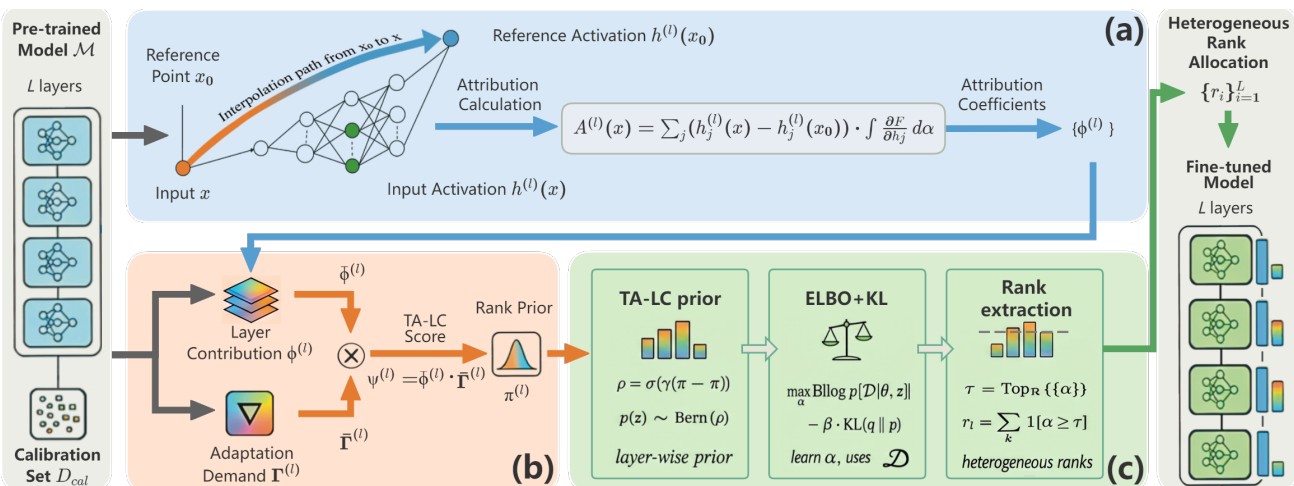

*Figure 3.* Overview of COBRA framework: **(a)** layer conductance attribution computes contribution $\{\Phi^{(l)}\}$; **(b)** dual-factor aggregation produces TA-LC prior $\{\pi^{(l)}\}$; **(c)** Bayesian rank allocation determines optimal ranks $\{r_l\}$.

demand) and output contribution (reflecting prediction influence) across multiple architectures. As shown in Table 1, gradient magnitude varies by $9\times$ to $18\times$ across layers, while output contribution varies by $8\times$ to $19\times$. Crucially, the two metrics peak at different positions: contribution peaks at middle layers (e.g., L15–17), whereas gradient peaks at top layers (e.g., L21–24). Such heterogeneity renders uniform allocation suboptimal, as critical layers are under-provisioned while less important ones waste capacity. *But which metric should guide the allocation?*

**Challenge #2: Decoupled Characteristics.** A layer's contribution to predictions and its adaptation demand are decoupled properties. As shown in Figure 2, the two are positively but moderately correlated (RoBERTa-large, $\rho = 0.66$; LLaMA-2-7B, $\rho = 0.56$). This decoupling has important implications for rank allocation strategies: (1) layers with high adaptation demand but low contribution would waste rank capacity on minimal prediction impact; (2) layers with high contribution but moderate adaptation demand would be under-allocated despite their importance. The divergence in peak positions (Table 1) confirms that neither property alone suffices for effective allocation. *How can we integrate both factors for principled allocation?*

## 3. Methodology

In this section, we introduce the COBRA framework, layer conductance attribution, dual-factor aggregation and Bayesian rank allocation.

### 3.1. COBRA Framework

To address the two challenges mentioned in Section 2, we propose COBRA, a principled framework that can be inte-

grated with various architectures for rank allocation. The overall pipeline is illustrated in Figure 3 and operates as follows. To address Challenge #1, layer conductance attribution computes each layer's contribution to predictions via path-integral attribution, yielding $\{\Phi^{(l)}\}$, which quantifies the heterogeneity in layer importance. To address Challenge #2, dual-factor aggregation integrates contribution with adaptation demand $\{\Gamma^{(l)}\}$ through multiplicative integration, producing the TA-LC distribution $\{\pi^{(l)}\}$; Bayesian rank allocation then translates this distribution into optimal heterogeneous ranks $\{r_l\}$ in a theoretically guided manner.

**Notation.** Let $\mathcal{M}$ denote a pre-trained model with $L$ target layers. For layer $l$, let $W_0^{(l)} \in \mathbb{R}^{d_{\text{out}} \times d_{\text{in}}}$ be the frozen pre-trained weights. LoRA introduces low-rank adapters $\Delta W^{(l)} = A^{(l)} B^{(l)}$ where $A^{(l)} \in \mathbb{R}^{d_{\text{out}} \times r_l}$ and $B^{(l)} \in \mathbb{R}^{r_l \times d_{\text{in}}}$. Our goal is to determine the optimal rank $r_l$ for each layer under a total budget constraint $\sum_l r_l \leq R_{\text{total}}$. Throughout the paper, $\gamma$ denotes the prior concentration controlling the TA-LC-to-prior mapping (Section 3.4); $\gamma_{\text{HC}}$ denotes the Hard Concrete left stretch parameter (Appendix A.6.2); $\beta$ is the KL weight in the ELBO objective; and $\rho^{(l)} \in (0, 1)$ is the Bayesian inclusion probability shared by all rank components within layer $l$.

### 3.2. Layer Conductance Attribution

To quantify how much each layer influences model predictions, we propose the layer conductance attribution module, as illustrated in Figure 3(a). The key insight is that layer importance should be measured through cumulative contribution along the entire computation path, rather than local gradient magnitudes at a single point. This module operates in three steps: (1) constructing interpolation paths between inputs and neutral references, (2) computing path-integrated

attributions for each layer, and (3) aggregating attributions across the calibration set.

**Path-Integral Formulation.** Consider an input $x$ and a neutral reference $x_0$ (e.g., the embedding of a padding token). We construct an interpolation path $x_\alpha = x_0 + \alpha(x - x_0)$ for $\alpha \in [0, 1]$, and define the layer contribution as:

$$\mathcal{A}^{(l)}(x) = \sum_{j=1}^{d_l} \left(h_j^{(l)}(x) - h_j^{(l)}(x_0)\right) \cdot \int_0^1 \frac{\partial F(x_\alpha)}{\partial h_j^{(l)}} \, d\alpha, \quad (1)$$

where $d_l$ is the hidden dimension of layer $l$, $h_j^{(l)}(\cdot)$ denotes the $j$-th neuron activation at layer $l$, and $F$ is the model output. The first term captures activation change relative to the neutral reference, while the integral accumulates the gradient contribution along the entire path.

**Completeness Property.** A key advantage of this formulation is its completeness: layer contributions sum exactly to the total output change,

$$\sum_{l=1}^{L} \mathcal{A}^{(l)}(x) = F(x) - F(x_0). \quad (2)$$

This property ensures that our attribution is exhaustive (every contribution is accounted for) and non-redundant (no double-counting), enabling principled cross-layer comparison of the signed contributions $\mathcal{A}^{(l)}(x)$. The aggregated importance score $\Phi^{(l)}$ (Eq. (3)) is derived from $|\mathcal{A}^{(l)}(x)|$ to capture functional relevance of both amplifying and suppressing layers; see Appendix A.4.6 for the precise relationship between completeness and $\Phi^{(l)}$. In contrast, gradient norms at individual layers lack such guarantees and may over- or under-estimate true layer contributions.

**Aggregated Attribution.** We compute the layer-wise attribution coefficient by aggregating over a calibration set $\mathcal{D}_{\text{cal}}$:

$$\Phi^{(l)} = \frac{1}{|\mathcal{D}_{\text{cal}}|} \sum_{x \in \mathcal{D}_{\text{cal}}} \left|\mathcal{A}^{(l)}(x)\right|. \quad (3)$$

The absolute value ensures both positive (amplifying) and negative (suppressing) contributions are captured, as both indicate functional relevance.

This module yields the attribution coefficients $\{\Phi^{(l)}\}_{l=1}^{L}$, representing each layer's contribution to predictions.

### 3.3. Dual-Factor Aggregation

To jointly consider layer contribution and adaptation demand, we propose the dual-factor aggregation module, as illustrated in Figure 3(b). This module operates in three steps: (1) quantifying adaptation demand via task-specific

gradients, (2) aggregating the two factors through multiplicative integration, and (3) constructing a normalized prior distribution for subsequent rank selection.

**Rationale.** The attribution coefficients $\{\Phi^{(l)}\}$ capture each layer's contribution to predictions. However, high contribution does not necessarily imply high adaptation need. A layer may strongly influence outputs yet require minimal modification if already well-suited to the task, and vice versa. This asymmetry necessitates a second factor: adaptation demand.

**Adaptation Demand.** We measure the adaptation demand of layer $l$ through the root-mean-square gradient magnitude with respect to its weights:

$$\Gamma^{(l)} = \sqrt{\mathbb{E}_{(x,y) \sim \mathcal{D}_{\text{cal}}} \left[\left\|\nabla_{W_0^{(l)}} \mathcal{L}(x, y)\right\|_F^2\right]}, \quad (4)$$

where $\mathcal{L}$ denotes the task-specific loss and $\|\cdot\|_F$ is the Frobenius norm. This metric directly reflects how much each layer's weights need to change for the target task: layers with large $\Gamma^{(l)}$ require substantial modification, while layers with small $\Gamma^{(l)}$ are already well-suited.

**Factor Aggregation.** The two factors capture complementary aspects of layer behavior: contribution reflects output influence while adaptation demand reflects required modification. We integrate them into a unified score satisfying three requirements: the score is high only when both factors are significant; increasing either factor increases the score; and if either factor vanishes, the score should be zero. The multiplicative form naturally satisfies all requirements. We first normalize each factor to ensure comparability across layers:

$$\tilde{\Phi}^{(l)} = \frac{\Phi^{(l)}}{\sum_{l'=1}^{L} \Phi^{(l')}}, \quad \tilde{\Gamma}^{(l)} = \frac{\Gamma^{(l)}}{\sum_{l'=1}^{L} \Gamma^{(l')}}, \quad (5)$$

and then define the TA-LC score as their product:

$$\Psi^{(l)} = \tilde{\Phi}^{(l)} \cdot \tilde{\Gamma}^{(l)}. \quad (6)$$

This formulation assigns high scores only to layers that both contribute substantially to predictions and require significant adaptation, while down-weighting layers that score high on only one factor (see Appendix A.5.1 for more details).

**Prior Distribution.** We normalize the TA-LC scores into a probability distribution:

$$\pi^{(l)} = \frac{\Psi^{(l)}}{\sum_{l'=1}^{L} \Psi^{(l')}}, \quad \text{with} \sum_{l=1}^{L} \pi^{(l)} = 1. \quad (7)$$

This distribution encodes the relative ranking of layers, serving as informative priors for guiding rank allocation.

Through this module, we obtain the prior distribution $\{\pi^{(l)}\}_{l=1}^{L}$ that addresses Challenge #2 by integrating layer contribution and adaptation demand into a unified importance measure.

## 3.4. Bayesian Rank Allocation

Building upon the integrated prior from Section 3.3, we propose the Bayesian rank allocation module to translate importance priors into optimal rank configurations, as illustrated in Figure 3(c). The key insight is that the TA-LC distribution provides informative priors for rank selection, while training data refines these priors into posteriors through variational optimization. This module operates in four steps: (1) parameterizing adapters with rank-level gating variables, (2) constructing TA-LC informed priors, (3) optimizing posteriors via continuous relaxation, and (4) extracting ranks through thresholding.

**Rank-Level Gating.** We parameterize each layer's adapter with a maximum rank $r_{\max}$ and introduce binary gating variables $z_k^{(l)} \in \{0, 1\}$ for each rank component:

$$\Delta W^{(l)} = \sum_{k=1}^{r_{\max}} z_k^{(l)} \cdot \mathbf{a}_k^{(l)} (\mathbf{b}_k^{(l)})^{\top}, \qquad (8)$$

where $\mathbf{a}_k^{(l)} \in \mathbb{R}^{d_{\mathrm{out}}}$ and $\mathbf{b}_k^{(l)} \in \mathbb{R}^{d_{\mathrm{in}}}$ denote the $k$-th column of $A^{(l)}$ and row of $B^{(l)}$, respectively. The effective rank is determined by the number of active gates: $r_l = \sum_{k=1}^{r_{\max}} z_k^{(l)}$. Inactive components ($z_k^{(l)} = 0$) are pruned while active components ($z_k^{(l)} = 1$) contribute to adaptation.

**Prior Construction.** We place Bernoulli priors on the gating variables with inclusion probabilities derived from the TA-LC distribution (Eq. (7)):

$$p(z_k^{(l)} = 1) = \rho^{(l)}, \quad \text{where} \quad \rho^{(l)} = \sigma\big(\gamma \cdot (\pi^{(l)} - \bar{\pi})\big), \quad (9)$$

where $\bar{\pi} = 1/L$ is the uniform baseline, $\sigma(\cdot)$ is the sigmoid function, and $\gamma > 0$ controls the prior concentration. All rank components within the same layer share $\rho^{(l)}$, as the TA-LC score is a layer-level metric; component-level differentiation emerges through posterior optimization. This construction biases layers with above-average TA-LC toward rank retention and layers with below-average TA-LC toward rank pruning (see Appendix A.6.3 for details).

**Posterior Optimization.** Direct optimization over discrete gates is intractable. We employ continuous relaxation, replacing binary $z_k^{(l)}$ with continuous surrogates $\tilde{z}_k^{(l)} \in (0, 1)$ (see Appendix A.6.2 for details):

$$\tilde{z}_k^{(l)} = \sigma \left( \frac{1}{\lambda} \left( \log \frac{\alpha_k^{(l)}}{1 - \alpha_k^{(l)}} + \log \frac{u}{1 - u} \right) \right), \qquad (10)$$

where $\alpha_k^{(l)} \in (0, 1)$ are learnable inclusion probabilities, $\lambda > 0$ controls the relaxation temperature, and $u \sim \mathrm{Uniform}(0, 1)$. We optimize the evidence lower bound (ELBO) that balances data likelihood against prior regularization:

$$\mathcal{J}_{\mathrm{ELBO}} = \mathbb{E}_q \big[ \log p(\mathcal{D}|\theta, \tilde{\mathbf{z}}) \big] - \beta \cdot D_{\mathrm{KL}} \big( q_\phi(\tilde{\mathbf{z}}) \| p(\mathbf{z}) \big), \quad (11)$$

where $\theta = \{A^{(l)}, B^{(l)}\}_l$ are LoRA parameters, $\phi = \{\alpha_k^{(l)}\}$ are variational parameters, and $\beta > 0$ weights the KL regularization. The KL divergence factorizes across layers and ranks (see Appendix A.6.1 for derivation):

$$D_{\mathrm{KL}}(q\|p) = \sum_{l=1}^{L} \sum_{k=1}^{r_{\max}} \left[ \alpha_k^{(l)} \log \frac{\alpha_k^{(l)}}{\rho^{(l)}} \right.$$
$$\left. + (1 - \alpha_k^{(l)}) \log \frac{1 - \alpha_k^{(l)}}{1 - \rho^{(l)}} \right]. \quad (12)$$

This objective encourages parsimony by penalizing deviation from the TA-LC informed prior, promoting configurations where layers with high TA-LC retain rank and layers with low TA-LC are pruned.

**Rank Extraction.** During training, the forward computation incorporates stochastic gates sampled via Eq. (10), and gradients flow through the reparameterization trick. After convergence, we extract deterministic ranks by thresholding inclusion probabilities. To satisfy the budget constraint $\sum_l r_l \leq R_{\mathrm{total}}$, we set the threshold $\tau$ adaptively as the $R_{\mathrm{total}}$-th largest value among all inclusion probabilities:

$$\tau = \mathrm{Top}_{R_{\mathrm{total}}} \big( \{\alpha_k^{(l)}\}_{l,k} \big), \qquad (13)$$

where $\mathrm{Top}_{R_{\mathrm{total}}}(\cdot)$ returns the $R_{\mathrm{total}}$-th largest element. Components with inclusion probability above this threshold are retained:

$$z_k^{(l)} = \mathbb{1}[\alpha_k^{(l)} \geq \tau], \quad r_l = \sum_{k=1}^{r_{\max}} z_k^{(l)}. \qquad (14)$$

This ensures exactly $R_{\mathrm{total}}$ rank components are retained across all layers, yielding the final heterogeneous rank configuration.

**End-to-End Integration.** Algorithm 1 summarizes the COBRA procedure. Stage 1 computes the dual-factor estimates on a small calibration set: layer contribution $\{\Phi^{(l)}\}$ measures how much each layer influences predictions, while adaptation demand $\{\Gamma^{(l)}\}$ measures how much each layer needs to change for the target task. Stage 2 aggregates these factors into the TA-LC distribution $\{\pi^{(l)}\}$ and transforms it into prior inclusion probabilities $\{\rho^{(l)}\}$, which encode our belief about layer-wise rank requirements before observing training data. Stage 3 performs Bayesian training where

**Algorithm 1** COBRA Procedure

---

**Require:** Pre-trained model $\mathcal{M}$, calibration set $\mathcal{D}_{\text{cal}}$, training set $\mathcal{D}$, budget $R_{\text{total}}$, max rank $r_{\text{max}}$, layers $L$, max iterations $T_{\text{max}}$, prior concentration $\gamma$, KL weight $\beta$, temperature schedule $\lambda_{\text{max}} \to \lambda_{\text{min}}$, init scale $\sigma_A$

1: **// Stage 1: Dual-Factor Estimation**
2: Compute $\{\Phi^{(l)}\}$ and $\{\Gamma^{(l)}\}$ via Eq. (1)-(4)
3: **// Stage 2: Prior Construction**
4: Compute $\{\pi^{(l)}\}$ and $\{\rho^{(l)}\}$ via Eq. (5)-(9)
5: **// Stage 3: Bayesian Training**
6: Initialize $A^{(l)} \sim \mathcal{N}(0, \sigma_A^2)$, $B^{(l)} \leftarrow 0$, $\alpha_k^{(l)} \leftarrow \rho^{(l)}$
7: **for** $t = 1$ **to** $T_{\text{max}}$ **do**
8:     Anneal $\lambda_t \leftarrow \lambda_{\text{min}} + \frac{1}{2}(\lambda_{\text{max}} - \lambda_{\text{min}})\big(1 + \cos(\pi t / T_{\text{max}})\big)$
9:     Sample $\tilde{z}_k^{(l)}$ via Eq. (10) with temperature $\lambda_t$
10:    Compute $\mathcal{J}_{\text{ELBO}}$ via Eq. (11) with KL weight $\beta$
11:    Update $\theta, \phi$ via gradient descent; **early-stop** if validation plateau
12: **end for**
13: **// Stage 4: Rank Extraction**
14: $\mathcal{Z} \leftarrow \arg\text{-top-}R_{\text{total}}\big(\{\alpha_k^{(l)}\}_{l,k}\big)$ {ties broken by layer/index order}
15: **for** $l = 1$ **to** $L$ **do**
16:    $r_l \leftarrow \sum_{k=1}^{r_{\text{max}}} \mathbb{1}\big[(l,k) \in \mathcal{Z}\big]$
17: **end for**
**Ensure:** Fine-tuned model with $\{r_l\}_{l=1}^{L}$

---

stochastic gates modulate rank components and the ELBO objective balances data fitting against prior regularization. Stage 4 extracts the final rank configuration by thresholding the learned inclusion probabilities, yielding a heterogeneous allocation where each layer receives an appropriate rank based on both prior knowledge and training signal.

Upon completion of Algorithm 1, we obtain heterogeneous rank configurations $\{r_l\}_{l=1}^{L}$ tailored to each layer's importance and adaptation requirements. A key advantage of this design is the decoupling between prior computation and training: the dual-factor estimates can be computed once on a small calibration set and reused across multiple training runs, amortizing the calibration cost and enabling efficient hyperparameter exploration.

# 4. Experiment

## 4.1. Implementation Details

**Software and Hardware.** All experiments are conducted on NVIDIA A100 80GB GPUs using PyTorch 2.0.1 (Paszke et al., 2019), Transformers 4.35.2 (Wolf et al., 2020), and PEFT 0.7.1 (Mangrulkar et al., 2022). We use mixed precision training (FP16 activations, FP32 gradients) for memory efficiency. The codebase is implemented in Python 3.10

with CUDA 12.1. More details in Appendix A.7.

**Datasets and Tasks.** We evaluate on three task categories: (1) GLUE benchmark (Wang et al., 2018) comprising 8 natural language understanding tasks; (2) CommonsenseQA (CSQA) (Talmor et al., 2019) for decoder-only commonsense reasoning; (3) Financial sentiment datasets (NEU, FXE, INV) (Ding et al., 2024a;b;c) for high-rank regime validation with continuous polarity regression. More details are provided in Appendix A.8.

**Model Architectures.** We test on representative architectures spanning encoder-only (RoBERTa-base 125M, RoBERTa-large 355M, DeBERTaV3-base 86M, Twitter-RoBERTa-large 355M (Barbieri et al., 2020)), decoder-only (LLaMA-2-7B (Touvron et al., 2023), Gemma-2-2B (Team et al., 2024)), and encoder-decoder (T5-base 220M) families. Following generalized LoRA (Zhang et al., 2023), we apply adapters to all linear layers in attention and feed-forward modules. More details are provided in Appendix A.9.

**Training Protocol.** We use AdamW optimizer (Loshchilov & Hutter, 2017) with learning rate $\eta \in \{1, 2, 5\} \times 10^{-4}$, batch size $b \in \{16, 32\}$, 6% warmup ratio, and linear decay. Training runs for 5-30 epochs with early stopping. LoRA hyperparameters include rank budget $R_{\text{total}} \in \{192, 384, 768\}$ for RoBERTa-base, scaling $\alpha = r$, and dropout 0.1. Financial sentiment experiments use rank budgets 9,216 and 12,288. More details are provided in Appendix A.10.

**Calibration Setup.** Layer conductance uses 50 integration steps (Sundararajan et al., 2017) on $|\mathcal{D}_{\text{cal}}| = 1000$ calibration samples. This one-time computation is amortized across multiple training runs with different hyperparameters. Bayesian allocation uses cosine temperature annealing $\lambda : 1.0 \to 0.1$, KL weight $\beta = 0.01$, and prior concentration $\gamma = 5.0$. Complexity analysis in Appendix A.11.

## 4.2. Main Results

**Comparison with Standard LoRA.** To demonstrate that adaptive rank allocation outperforms uniform allocation, we compare COBRA against standard LoRA with fixed rank across all layers. Table 2 shows results on RoBERTa-base using GLUE benchmark. Under the same parameter budget (0.8M trainable parameters, 0.64% of base model), COBRA achieves 88.6% average accuracy, outperforming LoRA ($r=8$) by +1.6 percentage points. The improvement is particularly pronounced on small datasets where rank allocation matters most: RTE (+4.1 pp), CoLA (+2.3 pp), and MRPC (+1.6 pp). Notably, COBRA with $r_{\text{avg}} = 8$ even surpasses LoRA with $r = 16$ (1.6M parameters, double the budget) by +0.9 percentage points while using half the pa-

*Table 2.* Performance on GLUE using RoBERTa-base; COBRA and LoRA ($r$=8) use 0.8M trainable parameters. Best results in **bold**, second best underlined.

| Method | Trainable Params | MNLI (m/mm) Acc. | SST-2 Acc. | MRPC Acc. | CoLA MCC | QNLI Acc. | QQP Acc. | RTE Acc. | STS-B Corr. | Avg. |
|---|---|---|---|---|---|---|---|---|---|---|
| Full Fine-Tuning | 125M | 87.6 | 94.8 | 90.2 | 63.6 | 92.8 | 91.9 | 78.7 | 91.2 | 86.4 |
| LoRA ($r$=8) | 0.8M | 89.0$_{\pm0.3}$ | 95.1$_{\pm0.2}$ | 90.7$_{\pm0.5}$ | 63.8$_{\pm1.2}$ | 93.2$_{\pm0.2}$ | 91.7$_{\pm0.1}$ | 81.6$_{\pm1.5}$ | 91.3$_{\pm0.3}$ | 87.0 |
| LoRA ($r$=16) | 1.6M | 89.5$_{\pm0.2}$ | 95.4$_{\pm0.2}$ | 91.2$_{\pm0.4}$ | 64.7$_{\pm1.0}$ | 93.6$_{\pm0.2}$ | 91.9$_{\pm0.1}$ | 83.4$_{\pm1.2}$ | 91.7$_{\pm0.2}$ | 87.7 |
| **COBRA** ($r_{avg}$=8) | 0.8M | 90.2$_{\pm0.2}$ | 95.8$_{\pm0.2}$ | 92.3$_{\pm0.4}$ | 66.1$_{\pm0.9}$ | 94.1$_{\pm0.2}$ | 92.3$_{\pm0.1}$ | 85.7$_{\pm0.2}$ | 92.2$_{\pm0.2}$ | 88.6 |
| *Improvement over LoRA ($r$=8)* | | +1.2 | +0.7 | +1.6 | +2.3 | +0.9 | +0.6 | +4.1 | +0.9 | +1.6 |

*Table 3.* Comparison with AdaLoRA on DeBERTaV3-base with 0.3M trainable parameters.

| Method | Trainable Params | MNLI (m/mm) Acc. | SST-2 Acc. | MRPC Acc. | CoLA MCC | QNLI Acc. | QQP Acc. | RTE Acc. | STS-B Corr. | Avg. |
|---|---|---|---|---|---|---|---|---|---|---|
| DeBERTaV3-base | 86M | 90.6 | 96.0 | 92.3 | 69.5 | 94.6 | 92.3 | 87.5 | 92.8 | 89.5 |
| LoRA ($r$=4) | 0.3M | 90.3 | 95.4 | 91.3 | 68.7 | 94.3 | 91.9 | 85.2 | 92.2 | 88.7 |
| AdaLoRA ($r_{avg}$=4) | 0.3M | 90.7 | 95.8 | 90.4 | 70.0 | 94.5 | 91.8 | 87.4 | 91.6 | 89.0 |
| **COBRA** ($r_{avg}$=4) | 0.3M | 91.2 | 96.1 | 92.6 | 70.6 | 94.9 | 92.5 | 88.1 | 92.9 | 89.9 |
| *Improvement over LoRA* | | +0.9 | +0.7 | +1.3 | +1.9 | +0.6 | +0.6 | +2.9 | +0.7 | +1.2 |
| *Improvement over AdaLoRA* | | +0.5 | +0.3 | +2.2 | +0.6 | +0.4 | +0.7 | +0.7 | +1.3 | +0.9 |

*Table 4.* Comparison with gradient-driven GoRA on T5-Base, LLaMA-2-7B, and LLaMA-3.1-8B. Details in Appendix A.8.5.

| Method | T5-Base GLUE-5 | LLaMA-2-7B MMLU | GSM8K | LLaMA-3.1-8B HumanEval | Avg. Gain (pp) |
|---|---|---|---|---|---|
| LoRA ($r$=8) | 82.1 | 45.2 | 42.1 | 43.1 | - |
| GoRA ($r_{ref}$=8) | 88.0 | 47.1 | 54.0 | 49.0 | +6.4 |
| **COBRA** ($r_{avg}$=8) | 89.1 | 48.6 | 56.1 | 52.7 | +8.5 |
| *Improvement (COBRA - GoRA)* | +1.1 | +1.5 | +2.1 | +3.7 | +2.1 |

*Table 5.* Performance comparison across different rank budgets and datasets. Full Fine-Tuning serves as the baseline (gray), while CO-BRA with two budget configurations (pink) demonstrates consistent improvements over uniform LoRA. Details in Appendix A.15.

| Method | Configuration | NEU Dataset MSE↓ | MAE↓ | FXE Dataset MSE↓ | MAE↓ | INV Dataset MSE↓ | MAE↓ |
|---|---|---|---|---|---|---|---|
| Full Fine-Tuning | - | 0.0189 | 0.0981 | 0.0202 | 0.1014 | 0.0183 | 0.0959 |
| LoRA | $r$=8 | 0.0223 | 0.1058 | 0.0217 | 0.1066 | 0.0201 | 0.1014 |
| | $r$=16 | 0.0229 | 0.1078 | 0.0224 | 0.1089 | 0.0196 | 0.1009 |
| | $r$=32 | 0.0213 | 0.1043 | 0.0228 | 0.1055 | 0.0203 | 0.1016 |
| | $r$=64 | 0.0203 | 0.1013 | 0.0206 | 0.1039 | 0.0192 | 0.0988 |
| | $r$=128 | 0.0195 | 0.0998 | 0.0203 | 0.1033 | 0.0187 | 0.0977 |
| | $r$=256 | 0.0193 | 0.0990 | 0.0207 | 0.1041 | 0.0186 | 0.0976 |
| | $r$=384 | 0.0189 | 0.0977 | 0.0205 | 0.1021 | 0.0184 | 0.0961 |
| | $r$=512 | 0.0186 | 0.0958 | 0.0199 | 0.1007 | 0.0183 | 0.0955 |
| AdaLoRA | 640 → 384 | 0.0209 | 0.1058 | 0.0218 | 0.1054 | 0.0191 | 0.1023 |
| | 640 → 512 | 0.0218 | 0.1052 | 0.0231 | 0.1079 | 0.0196 | 0.1005 |
| | 512 → 384 | 0.0228 | 0.1082 | 0.0237 | 0.1114 | 0.0204 | 0.1028 |
| COBRA | $r_{avg}$=384 | 0.0177 | 0.0940 | 0.0192 | 0.0978 | 0.0172 | 0.0917 |
| | $r_{avg}$=512 | 0.0174 | 0.0931 | 0.0185 | 0.0958 | 0.0170 | 0.0908 |

rameters, demonstrating superior parameter efficiency. CO-BRA also exceeds the full fine-tuning baseline (86.4%) by +2.2 percentage points despite updating only a small fraction of the parameters. This confirms that heterogeneous rank allocation based on layer importance significantly improves adaptation effectiveness compared to uniform distribution.

**Comparison with SVD-based Adaptive Methods.** We compare COBRA against AdaLoRA (Zhang et al., 2023), the SVD-based adaptive rank allocation method. Table 3 presents results on DeBERTaV3-base, the model explicitly evaluated in AdaLoRA's original paper. Both methods use 0.3M trainable parameters (0.35% of base model). COBRA achieves 89.9% average accuracy, outperforming AdaLoRA by +0.9% and standard LoRA by +1.2%. The performance gap is consistent across tasks: COBRA improves over AdaLoRA by +0.5% on MNLI, +0.7% on RTE, and +0.6% on CoLA. Critically, these gains come from addressing AdaLoRA's limitation of using only sensitivity-based importance. AdaLoRA allocates rank solely based on parameter relevance, missing layers with high output contribution but moderate gradient signal. COBRA's dual-factor approach corrects this by jointly considering both dimensions, leading to more principled allocation decisions.

**Comparison with Gradient-driven Methods.** To evaluate COBRA against gradient-driven approaches, we compare with GoRA (He et al., 2025), which allocates rank based on $|\partial L/\partial W|$ before training. Table 4 shows results on T5-Base, LLaMA-2-7B, and LLaMA-3.1-8B (Grattafiori et al., 2024) across GLUE-5, GSM8K (Cobbe et al., 2021), MMLU (Hendrycks et al., 2021), and HumanEval (Chen et al., 2021). On T5-Base GLUE, COBRA achieves 89.1% average, surpassing GoRA's 88.0% by +1.1 pp. On LLaMA-2-7B, COBRA improves over GoRA by +1.5 pp on MMLU and +2.1 pp on GSM8K, and on LLaMA-3.1-8B by +3.7 pp on HumanEval. Overall, COBRA achieves a +8.5 pp aver-

age gain over LoRA compared to GoRA's +6.4 pp (a +2.1 pp net improvement). These gains come from addressing GoRA's reliance on gradient magnitude alone, which misses layers with high output contribution but moderate gradient signal.

**Cross-Budget and Cross-Dataset Validation.** To validate COBRA's effectiveness under high-rank regimes, we conduct experiments on financial sentiment polarity prediction across three datasets (NEU, FXE, INV) with rank budgets higher than NLU benchmarks. Table 5 presents results under two budget configurations: $r_{avg} = 384$ (9,216 total) and $r_{avg} = 512$ (12,288 total). Under the 384-rank budget, COBRA achieves MSE of 0.0177 (NEU), 0.0192 (FXE), and 0.0172 (INV), outperforming uniform LoRA ($r$=384) by 6.3%, 6.3%, and 6.5%. Under the 512-rank budget, COBRA further improves to 0.0174, 0.0185, and 0.0170, achieving 6.5%, 7.0%, and 7.1% gains; averaged across both rank budgets ($r_{avg} = 384$ and $r_{avg} = 512$) and all three datasets, COBRA yields a 6.6% mean MSE reduction (see Appendix Table 39). COBRA consistently outperforms AdaLoRA across all configurations by approximately 10–24%. COBRA's relative improvements in high-rank regimes (5.2-7.1%) are substantially larger than in low-rank GLUE scenarios (1.4-2.6%), demonstrating that heterogeneous allocation becomes increasingly important as total parameter budget grows. This scaling behavior arises because uniform allocation wastes more absolute capacity on low-importance layers as rank increases, while COBRA's dual-factor approach exploits greater heterogeneity by re-

*Table 6.* Comparison on standard benchmarks under low-rank budgets (↑ higher is better). Best in **bold**; uniform LoRA shaded gray as the non-adaptive reference; equal LoRA parameter budget per setting.

| Method | Type | RoBERTa-b GLUE ($r=8$)↑ | DeBERTaV3-b GLUE ($r=4$)↑ | Gemma-2-2B CSQA ($r=8$)↑ |
|---|---|---|---|---|
| LoRA (Hu et al., 2022) | Uniform | 87.0 | 88.7 | 74.2 |
| DyLoRA (Valipour et al., 2023) | Dynamic range | 87.1 | 88.4 | 74.5 |
| AutoLoRA (Zhang et al., 2024) | NAS/Meta | 87.5 | 89.0 | 75.6 |
| DoRA-dyn (Mao et al., 2024) | Dynamic dist. | 87.9 | 89.3 | 75.9 |
| GoRA (He et al., 2025) | Gradient | 87.6 | 88.8 | 75.1 |
| **COBRA** | **Dual-factor** | **88.6** | **89.9** | **76.1** |

*Table 7.* High-rank regression on financial-sentiment with T-RoBERTa-large at rank $r=256$ (↓ lower is better). Best in **bold**; uniform LoRA shaded gray as the non-adaptive reference. This regime exhibits COBRA's largest margins, consistent with the rank-scaling argument.

| Method | Type | NEU MSE↓ | INV MSE↓ |
|---|---|---|---|
| LoRA | Uniform | 0.0193 | 0.0186 |
| DyLoRA | Dynamic range | 0.0191 | 0.0188 |
| AutoLoRA | NAS/Meta | 0.0188 | 0.0185 |
| DoRA-dyn | Dynamic dist. | 0.0186 | 0.0180 |
| GoRA | Gradient | 0.0189 | 0.0182 |
| **COBRA** | **Dual-factor** | **0.0183** | **0.0176** |

distributing larger budgets toward critical layers. Financial sentiment analysis particularly benefits from heterogeneous allocation because different layers capture distinct linguistic phenomena: shallow layers process domain-specific terminology, middle layers encode sentiment-bearing structures, and deep layers perform polarity aggregation.

**Low-Rank Standard Benchmarks.** To assess COBRA on standard benchmarks beyond GLUE, we compare against four adaptive-rank baselines (DyLoRA, AutoLoRA, DoRA-dyn, and GoRA) across three low-rank settings: RoBERTa-base GLUE at $r=8$, DeBERTaV3-base GLUE at $r=4$, and Gemma-2-2B on CSQA at $r=8$. Table 6 reports results under identical parameter budgets. COBRA achieves 88.6, 89.9, and 76.1 across the three settings, outperforming uniform LoRA by +1.6, +1.2, and +1.9 points and the strongest adaptive baseline DoRA-dyn by +0.7, +0.6, and +0.2 points. Notably, DyLoRA underperforms uniform LoRA on DeBERTaV3-base (88.4 vs 88.7), indicating that range-based dynamic allocation alone is insufficient when contribution and adaptation signals are not jointly considered.

**High-Rank Financial-Sentiment Regression.** To further evaluate COBRA in the high-rank regime, we compare the same five methods at $r=256$ on financial-sentiment regression with T-RoBERTa-large. Table 7 reports MSE on NEU and INV (lower is better). COBRA achieves 0.0183 and 0.0176, outperforming uniform LoRA by 5.2% and 5.4% and the strongest adaptive baseline DoRA-dyn by 1.6% and

2.2%. Notably, DyLoRA improves over uniform LoRA on NEU (0.0191 vs 0.0193) but slightly underperforms on INV (0.0188 vs 0.0186), consistent with the rank-scaling argument.

### 4.3. Ablation Study

**Factor Contribution Analysis.** Table 35 (due to page limit, more details in Appendix A.12) isolates the contribution of each importance factor. Single-factor methods ($\Phi$-only and $\Gamma$-only) achieve 88.8% (+0.1% over LoRA). Additive aggregation $\Phi + \Gamma$ reaches 89.2% (+0.5%), but our multiplicative formulation $\Phi \times \Gamma$ attains 89.9% (+1.2% over baseline, +0.7% over additive). This confirms the theoretical motivation: multiplicative aggregation enforces non-compensatory behavior, requiring layers to score high on contribution and adaptation demand for substantial rank, unlike additive aggregation schemes.

**Prior Concentration Strength.** Table 36 (due to page limit, more details in Appendix A.13) examines how prior concentration $\gamma$ affects allocation decisions. At $\gamma = 0$ (uniform prior, equivalent to no informative guidance), performance drops to 88.5%, underperforming even uniform LoRA, confirming that unguided allocation yields suboptimal configurations. Performance rises monotonically to peak at $\gamma = 5.0$ (89.9%), then declines for $\gamma \geq 10.0$, with $\gamma = 50.0$ falling to 88.3%. This inverted-U relationship reveals that weak priors fail to guide allocation while overly strong priors over-constrain adaptation. The +1.4% gain over the uniform-prior baseline validates interpretability-guided initialization.

**Bayesian Regularization Weight.** Table 37 (due to page limit, more details in Appendix A.14) examines the KL regularization coefficient $\beta$. Without regularization ($\beta = 0$), the method achieves only 88.8%, marginally above the LoRA baseline (88.7%) but substantially below full COBRA, indicating that pure likelihood maximization without prior regularization leads to unstable rank allocation. Our default $\beta = 0.01$ achieves optimal performance at 89.9% (+1.1% gain), while strong regularization ($\beta = 1.0$) drops to 88.2% from over-regularization. Moderate KL weighting strikes

the optimal balance between prior guidance and posterior flexibility, stabilizing the learned rank allocation.

## 4.4. Discussion

The fundamental challenge in adaptive rank allocation lies in reconciling two decoupled dimensions of layer importance. Single-factor approaches inevitably favor one dimension at the expense of the other. Gradient-driven methods like GoRA allocate rank to layers with high adaptation demand, but these layers may not meaningfully influence predictions. Conversely, existing sensitivity-based methods such as AdaLoRA prioritize local parameter relevance yet neglect whether the layer contributes to the model's reasoning pathway. COBRA resolves this dichotomy by integrating layer conductance, which serves as a measure of functional contribution to predictions, with gradient-based adaptation demand. This dual-factor aggregation ensures that the rank budget is not merely expended on layers that change the most, but rather invested in layers that matter most.

Furthermore, COBRA introduces a distinct shift in the optimization regime compared to prior adaptive methods. Previous approaches often rely on discrete structural interventions, such as SVD-based pruning that masks singular values iteratively during training. These operations cause the weight matrices to experience sudden topological shifts. Such abrupt changes force the optimizer to navigate a landscape of discontinuities and often result in loss oscillation or compensatory updates that degrade convergence stability. In contrast, COBRA employs continuous relaxation with temperature annealing to enable rank components to gradually suppress rather than suddenly vanish. This regime of smooth co-adaptation allows rank selection and parameter optimization to evolve concurrently. Consequently, the framework achieves stable convergence while avoiding the optimization challenges that plague hard-pruning approaches. The result is not merely a different rank allocation, but a fundamentally more stable optimization trajectory.

## 5. Related Work

**LoRA Variants and Optimization.** While LoRA demonstrates strong performance, its uniform rank allocation overlooks the heterogeneous importance of different modules. Several works address this through structural adaptation: AdaLoRA (Zhang et al., 2023) prunes singular values dynamically, DoRA (Liu et al., 2024a) decomposes weights into magnitude and direction, and VeRA (Kopiczko et al., 2024) achieves extreme efficiency by freezing random projection matrices. Beyond structure, optimization dynamics play a crucial role; LoRA+ (Hayou et al., 2024) balances learning rates for better convergence, while PiSSA (Meng et al., 2024) employs principal singular values for efficient initialization. Recent work further explores gradient-driven

rank allocation (He et al., 2025), quantization-based variants (Dettmers et al., 2023), and layer-wise adaptive strategies (Shinwari & Usama, 2025). However, these methods rely on a single importance metric without integrating complementary factors of layer contribution and adaptation demand, a gap our dual-factor framework directly addresses.

**Interpretability and Attribution.** Understanding layer-wise behavior is essential for principled rank allocation. Integrated gradients (Sundararajan et al., 2017) provide attribution by accumulating gradients, while Layer conductance (Dhamdhere et al., 2018) extends this to quantify hidden layer contributions via path integrals. Recent advances have deepened this understanding in Transformers: Patchscopes (Ghandeharioun et al., 2024) offers a unified framework for inspecting hidden representations, and Successor Heads (Gould et al., 2024) reveal interpretable attention circuits. Parallel research applies conductance analysis to broader neural interpretability (Dabounou & Baazzouz, 2024; Baazzouz & Dabounou, 2025). Building upon these foundational advances, our work leverages attribution techniques to guide heterogeneous rank allocation in PEFT.

## 6. Conclusion

We identified a critical limitation in adaptive rank allocation: reliance on single importance metrics fails to capture the decoupled relationship between adaptation demand and output contribution. COBRA addresses this through dual-factor aggregation that combines layer conductance attribution with gradient-based demand, producing the TA-LC distribution for Bayesian rank optimization. Layer conductance provides exhaustive and non-redundant layer-level importance quantification, directly aligning with allocation granularity. Experiments across different architectures demonstrate consistent improvements, with gains of 1.6 points on GLUE and a 6.6% average MSE reduction in high-rank regression regimes, revealing that COBRA's advantages scale with parameter budget. The interpretability-guided design offers a general paradigm for parameter allocation in PEFT.

## Acknowledgements

This research was funded by the Zhejiang Province Philosophy and Social Sciences Planning Project (No. 25GXSZ045YB) and the Zhejiang Provincial Education Science Planning Project (No. 2025SCG214).

## Impact Statement

This paper presents work whose goal is to advance the field of machine learning. There are many potential societal consequences of our work, none of which we feel must be specifically highlighted here.

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

# Appendices

*Table 8.* Appendix roadmap.

| Section | Content |
|---------|---------|
| A.0 | Extended related work (LoRA variants, interpretability, PEFT) |
| A.1–A.2 | Motivation analysis (heterogeneity, correlation) |
| A.3–A.6 | Methodology (proofs, derivations, aggregation) |
| A.7–A.10 | Experimental setup (hardware, datasets, hyperparameters) |
| A.11 | Complexity analysis (time, space, calibration overhead) |
| A.12–A.14 | Ablation studies (factor contribution, prior strength, KL weight) |
| A.15 | High-rank regime (financial sentiment, rank scaling) |
| A.16–A.18 | Allocation analyses (prior-only vs gated, reference sensitivity, layer position) |
| A.19–A.20 | Calibration cost (empirical overhead, calibration-set-size sensitivity) |

## Related Work (Extended)

This section provides supplementary context on related work, complementing the core discussion in the main text. We focus on advancements in LoRA variants, layer-wise interpretability for PEFT, and heterogeneous rank allocation strategies, all of which inform the design of our COBRA framework.

**LoRA Variants and Heterogeneous Rank Allocation.** LoRA's simplicity and inference efficiency have spurred numerous variants targeting its uniform rank allocation limitation, which overlooks layer-wise heterogeneity (addressed in our Section 2). Structural adaptation approaches include AdaLoRA (Zhang et al., 2023), which dynamically prunes trivial singular values to adjust effective rank; DoRA (Liu et al., 2024a), which decomposes weight updates into magnitude and direction to enhance adaptation; and VeRA (Kopiczko et al., 2024), which freezes random projection matrices for extreme efficiency. Optimization-focused variants such as LoRA+ (Hayou et al., 2024) balance learning rates across adapters for better convergence, while PiSSA (Meng et al., 2024) leverages principal singular values for efficient low-rank initialization.

Recent work has explored gradient-driven rank allocation (GoRA (He et al., 2025)) and layer-wise adaptive strategies (ARDLoRA (Shinwari & Usama, 2025)), which attempt to tailor rank to layer importance. However, these methods rely on single importance metrics (e.g., gradients or singular values) and fail to integrate complementary factors of layer prediction contribution and adaptation demand a critical gap addressed by our dual-factor aggregation module (Section 3.3). Additional related work includes LoRA-NAS (Chitty-Venkata et al., 2025; Chen et al., 2025), which uses neural architecture search to optimize rank configurations but lacks principled grounding for factor integration, and ALoRA (Liu et al., 2024b), which reallocates rank budgets via ablation-based importance but does not incorporate principled attribution signals. Further dynamic and search-based rank-allocation methods include AutoLoRA (Zhang et al., 2024), which meta-learns per-module ranks; DyLoRA (Valipour et al., 2023), which trains over a range of ranks in a search-free manner; and the dynamic rank-distribution variant of DoRA (Mao et al., 2024). A related line targets rank allocation for pruned LLMs, including LaRA (Zhou et al., 2026) and RankAdaptor (Zhou et al., 2025). Unlike these, COBRA grounds allocation in an interpretability-based prediction-contribution signal (layer conductance) combined with adaptation demand, rather than relying on gradient or SVD heuristics, architecture search, or validation feedback alone.

**Layer-wise Interpretability and Attribution for PEFT** Quantifying layer-wise importance is foundational to our COBRA framework, drawing on advances in neural network interpretability and attribution. Integrated gradients (Sundararajan et al., 2017; Lundström et al., 2022) provide an axiomatically grounded method for attributing model outputs to input features by accumulating gradients along interpolation paths, while Layer Conductance (Dhamdhere et al., 2018; Shrikumar et al., 2018) extends this to hidden layers via path integrals forming the basis of our layer conductance attribution module (Section 3.2).

In Transformers, recent interpretability advances have deepened understanding of layer-wise behavior: Patchscopes (Ghandeharioun et al., 2024) offers a unified framework for inspecting hidden representations across layers, and Successor Heads (Gould et al., 2024) reveal interpretable attention circuits that correlate with layer functionality. Parallel work has applied conductance analysis to broader neural interpretability tasks, such as enhancing model robustness (Dabounou & Baazzouz, 2024) and concept detection via layer conductance analysis (Baazzouz & Dabounou, 2025). Notably, Peng et al.

(Peng et al., 2025) recently explored layer attribution for PEFT but focused on adapter insertion rather than rank allocation, highlighting the novelty of our work in using attribution to guide low-rank adapter configuration.

**PEFT Fundamentals**   PEFT has emerged as a critical paradigm for adapting large pre-trained models to downstream tasks with minimal computational and memory overhead (Han et al., 2024; Ding et al., 2023). Three dominant PEFT paradigms have been established: (1) Adapter-based methods, which insert lightweight bottleneck modules into transformer layers (e.g., Houlsby adapters (Houlsby et al., 2019), Parallel adapters (Huh et al., 2024)); (2) Prompt tuning methods, which prepend learnable token sequences to inputs (e.g., prefix tuning (Li & Liang, 2021), prompt tuning (Lester et al., 2021)); and (3) Reparameterization-based methods, which constrain weight updates to low-dimensional subspaces (e.g., LoRA (Hu et al., 2022), BitFit (Ben-Zaken et al., 2022)).

Among these, LoRA has gained widespread adoption due to its simplicity (no architectural modifications) and inference efficiency (no additional inference latency compared to full fine-tuning). However, its uniform rank allocation across layers is a key limitation, one that has motivated subsequent work and our COBRA framework. Complementary PEFT advances include CoOp (Zhou et al., 2022b) and CoCoOp (Zhou et al., 2022a) for few-shot adaptation of vision-language models, but these methods do not address the core challenge of heterogeneous rank allocation that we tackle.

## Appendix A.1: Layer Heterogeneity

This appendix provides experimental evidence supporting Challenge #1 mentioned in Section 2. We measure layer-wise gradient magnitude and output contribution across multiple transformer architectures to quantify the extent of heterogeneity and establish the need for adaptive rank allocation.

**Notation.**   In the main text Table 1, we use the following abbreviations: **Enc** = Encoder-only, **Dec** = Decoder-only, **Enc-Dec** = Encoder-Decoder, **Grad.** = Gradient Magnitude, **Contrib.** = Output Contribution. Layer indices prefixed with **E** denote encoder layers and **D** denote decoder layers in encoder-decoder models.

### A.1.1 Experimental Setup

**Models.**   We analyze seven representative transformer models spanning different architectures, scales, and pre-training objectives:

1. **RoBERTa-large** (Liu et al., 2019b): A 24-layer encoder-only transformer with 355M parameters, pre-trained with dynamic masking on 160GB of text data.

2. **BERT-large-uncased** (Devlin et al., 2019): A 24-layer encoder-only transformer with 340M parameters, pre-trained with masked language modeling and next sentence prediction.

3. **LLaMA-2-7B** (Touvron et al., 2023): A 32-layer decoder-only transformer with 7B parameters, featuring multi-head attention and RMSNorm.

4. **GPT-2-medium** (Radford et al., 2019): A 24-layer decoder-only transformer with 355M parameters, pre-trained with causal language modeling on WebText.

5. **DeBERTaV3-base** (He et al., 2021): A 12-layer encoder-only transformer with 86M parameters, using disentangled attention and ELECTRA-style pre-training with gradient-disentangled embedding sharing.

6. **T5-base** (Raffel et al., 2020): A 24-layer (12 encoder + 12 decoder) encoder-decoder transformer with 220M parameters, pre-trained with span corruption on the C4 corpus.

7. **BART-large** (Lewis et al., 2020): A 24-layer (12 encoder + 12 decoder) encoder-decoder transformer with 406M parameters, pre-trained with text infilling and sentence permutation denoising objectives.

### A.1.2 Measurement Metrics

**Gradient Magnitude.**   For each layer $l \in \{1, \ldots, L\}$, we compute the gradient magnitude as the expected Frobenius norm of gradients with respect to the LoRA-relevant weight matrices. In the main text, Eq. (4) uses $W_0^{(l)}$ as an

abstract notation for layer-wise weights. In practice, we apply LoRA to all linear layers in the attention and feed-forward modules (Appendix A.9), so the gradient magnitude aggregates over the full set of LoRA-adapted matrices $\mathcal{W}^{(l)} = \{W_Q^{(l)}, W_K^{(l)}, W_V^{(l)}, W_O^{(l)}, W_{\text{up}}^{(l)}, W_{\text{down}}^{(l)}\}$:

$$\Gamma^{(l)} = \sqrt{\mathbb{E}_{(x,y) \sim \mathcal{D}_{\text{cal}}} \left[ \sum_{W \in \mathcal{W}^{(l)}} \|\nabla_W \mathcal{L}(x,y)\|_F^2 \right]} \tag{15}$$

where $\mathcal{L}$ denotes the task loss (cross-entropy for classification, language modeling loss for generation).

**Output Contribution.** We employ Layer Conductance (Dhamdhere et al., 2018), a path-integral attribution method based on Integrated Gradients (Sundararajan et al., 2017). Layer Conductance measures how much each layer contributes to the difference between the model's prediction on the input $x$ versus a neutral reference $x_0$:

$$\Phi^{(l)} = \frac{1}{|\mathcal{D}_{\text{cal}}|} \sum_{x \in \mathcal{D}_{\text{cal}}} \left| \sum_{j=1}^{d_l} \left( h_j^{(l)}(x) - h_j^{(l)}(x_0) \right) \cdot \int_{\alpha=0}^{1} \frac{\partial F(x_\alpha)}{\partial h_j^{(l)}} d\alpha \right| \tag{16}$$

where $h_j^{(l)}(\cdot)$ is the $j$-th hidden unit activation at layer $l$, $x_0$ is a reference input (all-padding tokens), $x_\alpha = x_0 + \alpha(x - x_0)$ is the linear interpolation, and $F$ is the model's output logit for the predicted class.

### A.1.3 Implementation Details

We summarize the key implementation details for reproducibility as follows:

- **Hardware**: All experiments are conducted on $4\times$ NVIDIA A100 80GB GPUs.

- **Software**: PyTorch 2.0.1 (Paszke et al., 2019), Transformers 4.30.0 (Wolf et al., 2020), Captum 0.6.0 (Kokhlikyan et al., 2020).

- **Calibration Set**: We randomly sample 1,000 examples from the training set as the calibration set $\mathcal{D}_{\text{cal}}$.

- **Integral Approximation**: The path integral in Eq. (16) is approximated using the Riemann sum with 50 uniformly spaced steps.

- **Precision**: Gradient computation uses mixed precision (FP16 activations, FP32 gradients) for memory efficiency.

- **Normalization**: Both metrics are normalized to $[0, 1]$ by dividing by the maximum value across layers within each model.

### A.1.4 Procedure

We follow a standardized procedure for all models:

1. **Model Loading**: Load the pre-trained model in evaluation mode with gradient computation enabled.

2. **Calibration Set Preparation**: Sample 1,000 examples, tokenize with model-specific tokenizer, and create batches of size 16.

3. **Gradient Magnitude Computation**:
   (a) For each batch, perform forward pass and compute task loss.
   (b) Backpropagate to obtain gradients for all LoRA-adapted matrices in $\mathcal{W}^{(l)}$ ($W_Q^{(l)}, W_K^{(l)}, W_V^{(l)}, W_O^{(l)}, W_{\text{up}}^{(l)}, W_{\text{down}}^{(l)}$).
   (c) Compute Frobenius norm for each layer and accumulate.
   (d) After processing all batches, compute mean and normalize.

4. **Output Contribution Computation**:

    (a) For each sample, construct the interpolation path from $x_0$ to $x$.

    (b) For each of 50 interpolation steps, compute gradients of output with respect to layer activations.

    (c) Apply the Layer Conductance formula (Eq. 16).

    (d) Aggregate across samples and normalize.

5. **Statistical Analysis**: Compute summary statistics including max/min ratio, standard deviation, and peak layer positions.

## A.1.5 Complete Results

**RoBERTa-large.** Table 9 presents the complete layer-wise measurements for RoBERTa-large. This 24-layer encoder-only model exhibits a clear bell-shaped contribution curve peaking at Layer 16, while gradient magnitude increases monotonically toward the top layers. The contribution ratio spans $10\times$ (from 0.10 to 1.00), indicating substantial heterogeneity across layers.

*Table 9.* Complete layer-wise measurements for RoBERTa-large. Gradient magnitude and output contribution are normalized to $[0, 1]$.

| Layer | Gradient | Contribution | Layer | Gradient | Contribution | Layer | Gradient | Contribution |
|---|---|---|---|---|---|---|---|---|
| 1 | 0.1132 | 0.1057 | 9 | 0.3704 | 0.4872 | 17 | 0.6710 | 0.8994 |
| 2 | 0.1503 | 0.1013 | 10 | 0.4169 | 0.5448 | 18 | 0.7374 | 0.8095 |
| 3 | 0.1966 | 0.1902 | 11 | 0.4465 | 0.5845 | 19 | 0.7770 | 0.7967 |
| 4 | 0.2124 | 0.2142 | 12 | 0.5075 | 0.7113 | 20 | 0.7799 | 0.7023 |
| 5 | 0.2194 | 0.2806 | 13 | 0.5197 | 0.7329 | 21 | 0.8419 | 0.6158 |
| 6 | 0.2586 | 0.3204 | 14 | 0.5415 | 0.7835 | 22 | 0.8793 | 0.5553 |
| 7 | 0.3069 | 0.3590 | 15 | 0.5811 | 0.8785 | 23 | 0.9354 | 0.5333 |
| 8 | 0.3014 | 0.4611 | **16** | **0.6642** | **1.0000** | 24 | 1.0000 | 0.4723 |

**BERT-large.** Table 10 presents the complete layer-wise measurements for BERT-large. Similar to RoBERTa, this encoder model shows contribution peaking at Layer 15 with a sharp decline toward top layers. The gradient ratio reaches $18\times$, higher than RoBERTa, reflecting differences in pre-training dynamics between dynamic masking and standard MLM.

*Table 10.* Complete layer-wise measurements for BERT-large.

| Layer | Gradient | Contribution | Layer | Gradient | Contribution | Layer | Gradient | Contribution |
|---|---|---|---|---|---|---|---|---|
| 1 | 0.0549 | 0.0991 | 9 | 0.2155 | 0.6754 | 17 | 0.5212 | 0.9402 |
| 2 | 0.0691 | 0.1386 | 10 | 0.2137 | 0.6610 | 18 | 0.5733 | 0.9091 |
| 3 | 0.0692 | 0.2161 | 11 | 0.2336 | 0.7702 | 19 | 0.6347 | 0.8496 |
| 4 | 0.0715 | 0.2950 | 12 | 0.2535 | 0.8472 | 20 | 0.7407 | 0.8377 |
| 5 | 0.0891 | 0.3571 | 13 | 0.3341 | 0.9198 | 21 | 0.7592 | 0.8017 |
| 6 | 0.0984 | 0.4915 | 14 | 0.3799 | 0.8586 | 22 | 0.8313 | 0.7404 |
| 7 | 0.1709 | 0.5242 | **15** | **0.4162** | **1.0000** | 23 | 0.8853 | 0.7403 |
| 8 | 0.1580 | 0.5435 | 16 | 0.4677 | 0.9343 | 24 | 1.0000 | 0.7097 |

**LLaMA-2-7B.** Table 11 presents the complete layer-wise measurements for LLaMA-2-7B. As a 32-layer decoder-only model, it exhibits the largest contribution ratio ($19\times$) among all tested models, with contribution peaking at Layer 20. The deeper architecture creates a wider separation between gradient and contribution peaks (12 layers), highlighting the importance of dual-factor consideration for large-scale models.

*Table 11.* Complete layer-wise measurements for LLaMA-2-7B.

| Layer | Gradient | Contribution | Layer | Gradient | Contribution | Layer | Gradient | Contribution | Layer | Gradient | Contribution |
|---|---|---|---|---|---|---|---|---|---|---|---|
| 1 | 0.0569 | 0.0531 | 9 | 0.0614 | 0.5468 | 17 | 0.2759 | 0.8860 | 25 | 0.5763 | 0.8729 |
| 2 | 0.0866 | 0.1459 | 10 | 0.1181 | 0.5862 | 18 | 0.2670 | 0.8770 | 26 | 0.6229 | 0.8064 |
| 3 | 0.0890 | 0.2280 | 11 | 0.1474 | 0.6325 | 19 | 0.3062 | 0.9135 | 27 | 0.7002 | 0.8299 |
| 4 | 0.0925 | 0.2884 | 12 | 0.1632 | 0.7102 | **20** | **0.3678** | **1.0000** | 28 | 0.7360 | 0.8181 |
| 5 | 0.0590 | 0.3169 | 13 | 0.1761 | 0.6794 | 21 | 0.3887 | 0.8539 | 29 | 0.7834 | 0.7778 |
| 6 | 0.0579 | 0.3635 | 14 | 0.2239 | 0.7266 | 22 | 0.4517 | 0.9062 | 30 | 0.8591 | 0.7789 |
| 7 | 0.1027 | 0.4168 | 15 | 0.2143 | 0.8194 | 23 | 0.4733 | 0.8599 | 31 | 0.9171 | 0.6812 |
| 8 | 0.0838 | 0.4555 | 16 | 0.2463 | 0.8091 | 24 | 0.5437 | 0.8624 | 32 | 1.0000 | 0.7250 |

**GPT-2-medium.** Table 12 presents the complete layer-wise measurements for GPT-2-medium. This 24-layer decoder-only model shows contribution peaking at Layer 14, earlier than encoder models of the same depth. The autoregressive architecture concentrates gradients at top layers while middle layers maintain high contribution, consistent with the pattern observed in LLaMA-2-7B.

*Table 12.* Complete layer-wise measurements for GPT-2-medium.

| Layer | Gradient | Contribution | Layer | Gradient | Contribution | Layer | Gradient | Contribution |
|---|---|---|---|---|---|---|---|---|
| 1 | 0.0658 | 0.0973 | 9 | 0.2806 | 0.5316 | 17 | 0.6360 | 0.8309 |
| 2 | 0.0765 | 0.0815 | 10 | 0.3065 | 0.6340 | 18 | 0.6892 | 0.7502 |
| 3 | 0.1211 | 0.1676 | 11 | 0.3578 | 0.6684 | 19 | 0.7278 | 0.6925 |
| 4 | 0.1462 | 0.2406 | 12 | 0.4056 | 0.7641 | 20 | 0.7517 | 0.6297 |
| 5 | 0.1554 | 0.2766 | 13 | 0.4421 | 0.8827 | 21 | 0.8102 | 0.6032 |
| 6 | 0.1797 | 0.3708 | **14** | **0.4586** | **1.0000** | 22 | 0.8899 | 0.6192 |
| 7 | 0.1785 | 0.4146 | 15 | 0.5736 | 0.8811 | 23 | 0.9528 | 0.5134 |
| 8 | 0.2188 | 0.5505 | 16 | 0.5408 | 0.8208 | 24 | 1.0000 | 0.5106 |

**DeBERTaV3-base.** Table 13 presents the complete layer-wise measurements for DeBERTaV3-base. Despite having only 12 layers, this model exhibits similar heterogeneity patterns to deeper encoders, with contribution peaking at Layer 7 (the relative middle). The disentangled attention mechanism does not fundamentally alter the gradient-contribution relationship, confirming that heterogeneity is architecture-agnostic.

*Table 13.* Complete layer-wise measurements for DeBERTaV3-base.

| Layer | Gradient | Contribution | Layer | Gradient | Contribution | Layer | Gradient | Contribution |
|---|---|---|---|---|---|---|---|---|
| 1 | 0.0915 | 0.1244 | 5 | 0.4803 | 0.5471 | 9 | 0.7950 | 0.7940 |
| 2 | 0.2343 | 0.2116 | 6 | 0.5623 | 0.7267 | 10 | 0.8389 | 0.7154 |
| 3 | 0.3222 | 0.3615 | **7** | **0.6363** | **1.0000** | 11 | 0.9055 | 0.6181 |
| 4 | 0.3970 | 0.4374 | 8 | 0.7056 | 0.8785 | 12 | 1.0000 | 0.5492 |

**T5-base.** Table 14 presents the complete layer-wise measurements for T5-base. This encoder-decoder model shows a distinctive pattern where encoder gradients plateau around 0.50 while decoder gradients continue increasing to 1.00. Contribution peaks at Decoder Layer 4, where cross-attention first integrates encoder representations, highlighting the critical role of early decoder layers in sequence-to-sequence tasks.

*Table 14.* Complete layer-wise measurements for T5-base. E denotes encoder layers, D denotes decoder layers.

| Layer | Gradient | Contribution | Layer | Gradient | Contribution | Layer | Gradient | Contribution | Layer | Gradient | Contribution |
|---|---|---|---|---|---|---|---|---|---|---|---|
| E1 | 0.0716 | 0.0921 | E7 | 0.2531 | 0.5540 | D1 | 0.4978 | 0.8351 | D7 | 0.7637 | 0.7909 |
| E2 | 0.1102 | 0.2383 | E8 | 0.3194 | 0.6409 | D2 | 0.5537 | 0.8427 | D8 | 0.8289 | 0.7627 |
| E3 | 0.1181 | 0.2933 | E9 | 0.3446 | 0.7275 | D3 | 0.5786 | 0.8672 | D9 | 0.8315 | 0.6951 |
| E4 | 0.1331 | 0.3859 | E10 | 0.4915 | 0.7584 | **D4** | **0.6443** | **1.0000** | D10 | 0.8676 | 0.6433 |
| E5 | 0.2121 | 0.4247 | E11 | 0.4703 | 0.8473 | D5 | 0.6679 | 0.8890 | D11 | 0.9774 | 0.5778 |
| E6 | 0.2152 | 0.4997 | E12 | 0.4911 | 0.7744 | D6 | 0.7083 | 0.8281 | D12 | 1.0000 | 0.5543 |

**BART-large.** Table 15 presents the complete layer-wise measurements for BART-large. Similar to T5, this encoder-decoder model exhibits encoder gradient plateau and decoder gradient increase. Contribution peaks at Decoder Layer 5, close to T5's Decoder Layer 4 despite different pre-training objectives (denoising vs. span corruption). This consistency across encoder-decoder architectures suggests that the pattern is driven by architectural properties rather than pre-training specifics.

*Table 15.* Complete layer-wise measurements for BART-large. E denotes encoder layers, D denotes decoder layers.

| Layer | Gradient | Contribution | Layer | Gradient | Contribution | Layer | Gradient | Contribution | Layer | Gradient | Contribution |
|---|---|---|---|---|---|---|---|---|---|---|---|
| E1 | 0.1066 | 0.0609 | E7 | 0.2605 | 0.5302 | D1 | 0.4610 | 0.7901 | D7 | 0.6619 | 0.8260 |
| E2 | 0.0584 | 0.1450 | E8 | 0.2505 | 0.5907 | D2 | 0.4814 | 0.8626 | D8 | 0.7382 | 0.7441 |
| E3 | 0.1127 | 0.2497 | E9 | 0.3220 | 0.6634 | D3 | 0.4747 | 0.8515 | D9 | 0.7841 | 0.6488 |
| E4 | 0.1814 | 0.2910 | E10 | 0.4395 | 0.7196 | D4 | 0.5323 | 0.9179 | D10 | 0.7889 | 0.5727 |
| E5 | 0.2198 | 0.3997 | E11 | 0.4209 | 0.8424 | **D5** | **0.6588** | **1.0000** | D11 | 0.9014 | 0.5355 |
| E6 | 0.2356 | 0.4569 | E12 | 0.4316 | 0.7482 | D6 | 0.6587 | 0.8833 | D12 | 1.0000 | 0.4652 |

### A.1.6 Summary Statistics

*Table 16.* Summary of layer heterogeneity across transformer models. Gradient Ratio and Contribution Ratio denote the ratio between maximum and minimum normalized values. Peak Gap measures the layer distance between contribution peak and gradient peak, indicating the degree of decoupling between the two properties.

| Model | Layers | Gradient Ratio | Contribution Ratio | Contribution Peak | Gradient Peak | Peak Gap |
|---|---|---|---|---|---|---|
| RoBERTa-large | 24 | 9× | 10× | Layer 16 | Layer 24 | 8 |
| BERT-large | 24 | 18× | 10× | Layer 15 | Layer 24 | 9 |
| LLaMA-2-7B | 32 | 18× | 19× | Layer 20 | Layer 32 | 12 |
| GPT-2-medium | 24 | 15× | 12× | Layer 14 | Layer 24 | 10 |
| DeBERTaV3-base | 12 | 11× | 8× | Layer 7 | Layer 12 | 5 |
| T5-base | 24 | 14× | 11× | Decoder 4 | Decoder 12 | 8 |
| BART-large | 24 | 17× | 16× | Decoder 5 | Decoder 12 | 7 |

**Gradient Ratio.** The gradient ratio ranges from $9\times$ (RoBERTa-large) to $18\times$ (BERT-large, LLaMA-2-7B), indicating that top layers receive 9–18 times more gradient signal than bottom layers during backpropagation. This substantial variation reflects the well-known gradient flow pattern in deep networks where layers closer to the loss function exhibit larger gradients.

**Contribution Ratio.** The contribution ratio exhibits even larger variation, ranging from $8\times$ (DeBERTaV3-base) to $19\times$ (LLaMA-2-7B). Larger models tend to show higher contribution ratios, suggesting that deeper architectures develop more pronounced specialization across layers. The middle layers consistently achieve the highest contribution scores, reflecting their role in encoding transferable semantic representations.

**Peak Gap.** The peak gap, defined as the layer distance between contribution peak and gradient peak, ranges from 5 layers (DeBERTaV3-base) to 12 layers (LLaMA-2-7B). This consistent separation across all seven models demonstrates that gradient magnitude and output contribution capture fundamentally different aspects of layer importance. The gap tends to scale with model depth: 12-layer DeBERTaV3 has a gap of 5, while 32-layer LLaMA-2-7B has a gap of 12.

**Architecture-Specific Patterns.** Encoder-only models (RoBERTa, BERT, DeBERTaV3) show contribution peaks in the upper-middle layers (relative position 58–67% of total depth), while decoder-only models (LLaMA, GPT-2) show peaks at similar relative positions (58–63%). Encoder-decoder models (T5, BART) exhibit contribution peaks in early decoder layers, where cross-attention first integrates encoder representations for generation.

### A.1.7 Discussion

**Consistency Across Architectures.** Despite substantial differences in model architecture (encoder-only, decoder-only, encoder-decoder), scale (86M to 7B parameters), and pre-training objectives (MLM, CLM, span corruption, denoising, RTD), all seven models exhibit qualitatively similar heterogeneity patterns. This consistency suggests that layer heterogeneity is an intrinsic property of deep transformer networks arising from the information processing hierarchy, not an artifact of specific training procedures.

**Gradient Monotonicity.** The monotonic increase in gradient magnitude with layer depth is consistent with the theoretical analysis of gradient flow in deep networks (He et al., 2016). Top layers are closest to the loss function and thus exhibit the largest gradients during backpropagation. This pattern has been observed in prior work on transformer training dynamics (Xiong et al., 2020).

**Bell-Shaped Contribution.** The bell-shaped output contribution distribution aligns with linguistic probing studies showing that different layers encode different types of information (Tenney et al., 2019; Jawahar et al., 2019). Lower layers capture surface-level features, middle layers encode syntactic and semantic information, and top layers perform task-specific transformations. The middle layers' high contribution reflects their role in encoding transferable linguistic knowledge.

**Impact of Model Architecture.** The encoder-decoder models (T5-base and BART-large) show an interesting pattern where encoder layers plateau in gradient magnitude while decoder layers continue to increase. This reflects the architectural

design where the encoder builds representations and the decoder generates outputs, making decoder layers more directly connected to the loss function. Both T5 and BART exhibit contribution peaks at early decoder layers (D4 for T5, D5 for BART), where cross-attention integrates encoder representations for generation.

**Implications for LoRA.** The observed heterogeneity has direct implications for LoRA rank allocation. Uniform allocation wastes parameters on low-importance layers while potentially under-provisioning critical layers. The systematic separation between gradient and contribution peaks indicates that optimizing for either metric alone would lead to suboptimal allocation, motivating our dual-factor approach.

## Appendix A.2: Decoupled Characteristics

This appendix provides experimental evidence supporting Challenge #2 mentioned in Section 2. We systematically analyze the correlation between gradient magnitude and output contribution, demonstrating that these two properties are decoupled, thereby motivating our dual-factor approach.

### A.2.1 Experimental Setup

**Data Source.** We use the layer-wise gradient magnitude $\{\Gamma^{(l)}\}_{l=1}^{L}$ and output contribution $\{\Phi^{(l)}\}_{l=1}^{L}$ computed in Appendix A.1. Both factors are min-max normalized to $[0, 1]$ for each model. The primary analysis focuses on two representative architectures: RoBERTa-large (Liu et al., 2019b) (24-layer encoder) and LLaMA-2-7B (Touvron et al., 2023) (32-layer decoder). Cross-model validation extends to BERT-large, GPT-2-medium, DeBERTaV3-base, T5-base, and BART-large.

**Normalization.** All gradient and contribution values are min-max normalized within each model:

$$\tilde{X}^{(l)} = \frac{X^{(l)} - \min_j X^{(j)}}{\max_j X^{(j)} - \min_j X^{(j)}} \tag{17}$$

This ensures $\tilde{X}^{(l)} \in [0, 1]$ with $\min_l \tilde{X}^{(l)} = 0$ and $\max_l \tilde{X}^{(l)} = 1$.

**Correlation Metrics.** We compute multiple correlation coefficients to comprehensively characterize the relationship between the two metrics:

- **Pearson correlation coefficient** ($\rho$): Measures the linear correlation between two variables. For variables $X$ and $Y$:

$$\rho = \frac{\sum_i (X_i - \bar{X})(Y_i - \bar{Y})}{\sqrt{\sum_i (X_i - \bar{X})^2}\sqrt{\sum_i (Y_i - \bar{Y})^2}} \tag{18}$$

- **Spearman rank correlation** ($r_s$): Measures the monotonic relationship between two variables using rank values, robust to non-linear relationships.

- **Kendall's tau** ($\tau$): Measures the ordinal association based on concordant and discordant pairs, robust to outliers.

- **Coefficient of determination** ($R^2$): The proportion of variance in output contribution explained by gradient magnitude under a linear model.

### A.2.2 Experimental Procedure

We follow a standardized three-step procedure to analyze the correlation between gradient magnitude and output contribution across all models:

1. **Data Preparation**: Extract the normalized gradient magnitude and output contribution vectors for each model from Appendix A.1. Verify min-max normalization: $\min = 0, \max = 1$.

2. **Correlation Computation**:

    (a) Compute Pearson, Spearman, and Kendall correlations using scipy.stats (Virtanen et al., 2020).

    (b) Fit a linear regression model: Contribution $= \beta_0 + \beta_1 \cdot \text{Gradient} + \epsilon$.

    (c) Compute $R^2$ and residual standard deviation.

3. **Failure Mode Analysis**: Identify layers that deviate substantially from the linear trend and analyze their characteristics.

### A.2.3 Correlation Analysis Results

*Table 17.* Complete correlation analysis for RoBERTa-large. The moderate correlation ($\rho = 0.66$) indicates that gradient magnitude and output contribution are related but not aligned.

| Metric | Value |
|---|---|
| Pearson $\rho$ | 0.66 |
| Spearman $r_s$ | 0.71 |
| Kendall's $\tau$ | 0.53 |
| $R^2$ (variance explained) | 0.44 |
| Residual std. dev. | 0.25 |

**RoBERTa-large's Result Analysis.** The Pearson correlation of $\rho = 0.66$ indicates a moderate positive relationship. Importantly, the coefficient of determination $R^2 = 0.44$ implies that gradient magnitude explains only 44% of the variance in output contribution. The remaining 56% represents information unique to output contribution that gradient magnitude fails to capture. This decoupling motivates incorporating both factors in rank allocation.

*Table 18.* Complete correlation analysis for LLaMA-2-7B. The moderate correlation ($\rho = 0.56$) is lower than encoder models, reflecting the more complex gradient flow in autoregressive decoder architectures.

| Metric | Value |
|---|---|
| Pearson $\rho$ | 0.56 |
| Spearman $r_s$ | 0.62 |
| Kendall's $\tau$ | 0.44 |
| $R^2$ (variance explained) | 0.31 |
| Residual std. dev. | 0.27 |

**LLaMA-2-7B's Result Analysis.** LLaMA-2-7B exhibits a moderate correlation ($\rho = 0.56$) that is lower than RoBERTa-large ($\rho = 0.66$). This difference arises from the distinct gradient flow patterns in decoder-only architectures:

- **Autoregressive gradient concentration**: In decoder-only models, gradients concentrate heavily at top layers (Layer 28–32) due to the causal language modeling objective, where prediction loss directly propagates to the final layers.

- **Distributed contribution**: Output contribution peaks at middle layers (Layer 17–22) where semantic representations are most refined, then gradually decreases toward top layers that focus on next-token prediction.

- **Larger model effect**: With 32 layers (vs. 24 for RoBERTa), more layers fall into the "high gradient, moderate contribution" region, reducing overall correlation.

The $R^2 = 0.31$ indicates that only 31% of contribution variance is explained by gradients, with 69% unexplained, strong evidence for dual-factor necessity in large decoder models.

**Cross-Model Validation.** Table 19 extends the analysis to seven transformer models spanning encoder, decoder, and encoder-decoder architectures.

**Key Finding.** The Pearson correlation ranges from $\rho = 0.53$ to $0.70$ across seven diverse models, confirming that the decoupling between contribution and adaptation demand is a robust phenomenon. Encoder-only models show the highest correlations, followed by encoder-decoder models, and decoder-only models. Gradient magnitude explains only 28–49% of the variance in output contribution, leaving over half unexplained. This consistent pattern across architectures strongly motivates the need for dual-factor rank allocation.

*Table 19.* Correlation between gradient magnitude and output contribution across models. All correlations are substantially below 1.0, indicating that the decoupling between contribution and adaptation demand is a consistent phenomenon across architectures.

| Model | Type | $\rho$ | $r_s$ | $\tau$ | $R^2$ |
|---|---|---|---|---|---|
| *Encoder-only models* | | | | | |
| RoBERTa-large | Encoder | 0.66 | 0.71 | 0.53 | 0.44 |
| BERT-large | Encoder | 0.62 | 0.67 | 0.49 | 0.38 |
| DeBERTaV3-base | Encoder | 0.70 | 0.74 | 0.56 | 0.49 |
| *Decoder-only models* | | | | | |
| LLaMA-2-7B | Decoder | 0.56 | 0.62 | 0.44 | 0.31 |
| GPT-2-medium | Decoder | 0.54 | 0.60 | 0.43 | 0.29 |
| *Encoder-decoder models* | | | | | |
| T5-base | Encoder-Decoder | 0.60 | 0.65 | 0.47 | 0.36 |
| BART-large | Encoder-Decoder | 0.53 | 0.58 | 0.41 | 0.28 |

### A.2.4 Visualization of Decoupled Characteristics

Figure 4 visualizes the relationship between gradient magnitude and output contribution for two representative models: RoBERTa-large (encoder) and LLaMA-2-7B (decoder). This figure corresponds to Figure 2 in the main text and provides additional context for interpreting the decoupling phenomenon.

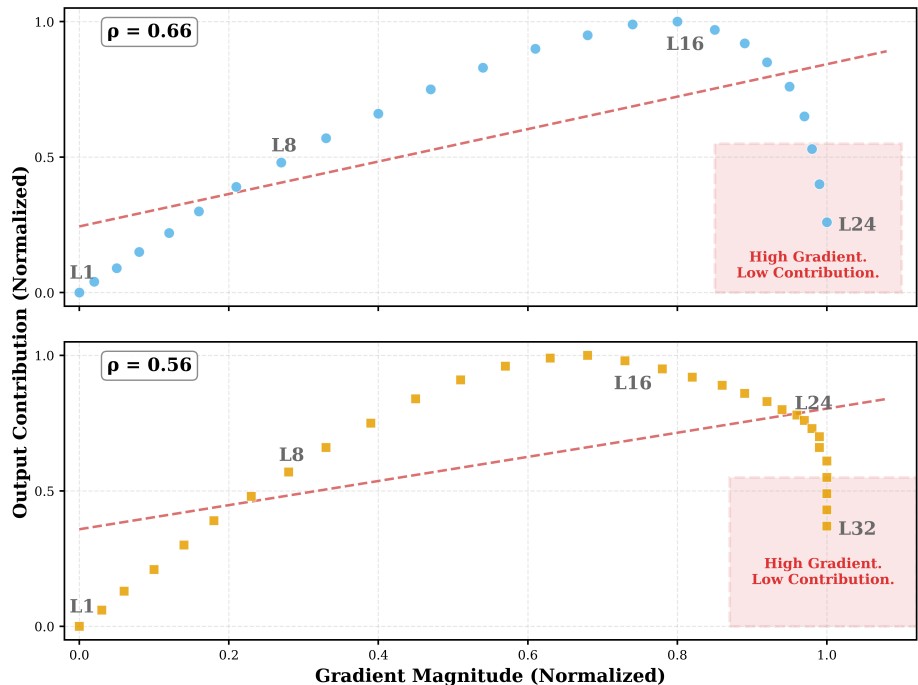

*Figure 4.* Scatter plot of gradient magnitude versus output contribution for RoBERTa-large (**upper**, 24 layers, encoder-only) and LLaMA-2-7B (**lower**, 32 layers, decoder-only). Each point represents one transformer layer, with key layers labeled. The red dashed line shows the linear regression fit. The highlighted region (bottom-right) identifies layers with high adaptation demand but low contribution, these layers would be over-allocated if using adaptation demand alone. Both models exhibit moderate positive correlation ($\rho = 0.66$ for RoBERTa, $\rho = 0.56$ for LLaMA), indicating that the two properties are related but decoupled.

**Visual Patterns.** Several key patterns emerge from the scatter plots:

1. **Positive but imperfect correlation.** Both plots show a general upward trend from bottom-left to top-right, confirming that adaptation demand and contribution are positively correlated. However, substantial scatter around the regression line indicates that these two properties are decoupled.

2. **Bell-shaped contribution trajectory.** Following the layer indices (L1 → L24 for RoBERTa, L1 → L32 for LLaMA),

the points first rise toward the top of the plot (increasing contribution) then curve back down toward the right (decreasing contribution at high-gradient layers). This trajectory reflects the bell-shaped contribution distribution observed in the detailed measurements.

3. **High-gradient, low-contribution region.** The highlighted bottom-right region contains top layers (L21–L24 for RoBERTa, L28–L32 for LLaMA) that exhibit maximum gradient but below-average contribution. These layers would receive excessive rank allocation under single-factor methods, wasting parameter capacity on adaptations with minimal prediction impact.

4. **Steeper decline in encoder models.** RoBERTa shows a more pronounced drop in contribution at top layers compared to LLaMA. This difference reflects architectural properties: encoder models use bidirectional attention where top layers specialize for [CLS] token representation, while decoder models maintain broader utility across layers for autoregressive generation.

**Layer-by-Layer Trajectory.** The trajectory of layers through the gradient-contribution space reveals the progression of information processing:

- **Bottom layers (L1–L8)**: Located in the bottom-left quadrant with both low gradient and low contribution. These layers perform basic tokenization and positional encoding with minimal task-specific adaptation needs.

- **Middle layers (L8–L16 for RoBERTa, L8–L20 for LLaMA)**: Rise toward the top of the plot, achieving peak contribution. These layers encode rich semantic representations that are critical for downstream predictions.

- **Top layers (L17–L24 for RoBERTa, L21–L32 for LLaMA)**: Move rightward (increasing gradient) while descending (decreasing contribution). These layers perform task-specific transformations that receive strong gradient signals but contribute less to the prediction path.

**Implications for Rank Allocation.** The visualization directly illustrates why single-factor allocation is suboptimal:

- Layers in the highlighted region (high gradient, low contribution) would be over-allocated, wasting capacity.

- Layers at the top of the plot (moderate gradient, high contribution) would be under-allocated, missing opportunities to enhance critical representations.

- Effective allocation requires considering both metrics jointly, motivating our multiplicative TA-LC score: $\Psi^{(l)} = \tilde{\Phi}^{(l)} \cdot \tilde{\Gamma}^{(l)}$.

Figure 5 visualizes how COBRA's two signals jointly determine the final per-layer rank on RoBERTa-large. Each of the 24 layers is placed at its normalized contribution $\Phi$ (vertical axis) and adaptation demand $\Gamma$ (horizontal axis); axes show layer-wise values from Table 9 (each metric normalized to $[0, 1]$ by its maximum), while color and marker size encode the rank ultimately allocated under the fixed budget $R_{\text{total}} = 192$, and the dashed lines split the layers into four quadrants at the medians of $\Phi$ and $\Gamma$. The mapping confirms the mechanism behaves as intended. Layers high in both signals (HH, upper right) receive the largest ranks, e.g., layer 18 ($\Phi = 0.81$, $\Gamma = 0.74$) attains the maximum rank of 16. In contrast, layers with high adaptation demand but low contribution (HL, lower right), exactly the region a gradient-only allocator would over-provision, are deliberately restricted to small ranks: layer 24, whose gradient is maximal ($\Gamma = 1.00$) yet whose contribution is comparatively low ($\Phi = 0.47$), receives only rank 5. Layers low in both signals (LL, lower left) collapse to near-minimal rank. The allocation therefore tracks output contribution rather than gradient magnitude alone, directly validating the dual-factor design.

### A.2.5 Layer-wise Residual Analysis

To understand the source of the decoupling, we examine the residuals from the linear regression model. Table 20 shows the per-layer values for RoBERTa-large.

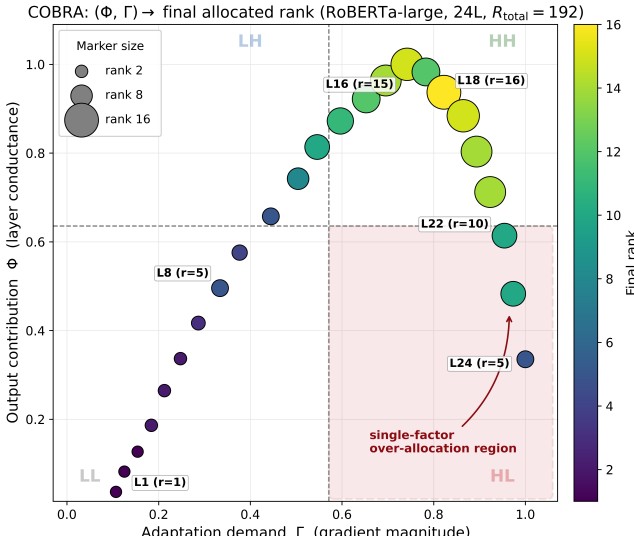

*Figure 5.* COBRA's two signals jointly determine per-layer rank on RoBERTa-large ($R_{\text{total}}$=192). Each layer is placed by adaptation demand $\Gamma$ (horizontal) and output contribution $\Phi$ (vertical); color and marker size encode the allocated rank, and dashed lines mark the medians. High-demand low-contribution layers (HL, lower right, shaded), where a gradient-only allocator would over-provision, receive small ranks, while layers high in both signals (HH) receive the largest. This confirms the allocation tracks contribution rather than gradient magnitude alone.

*Table 20.* Layer-wise gradient, contribution, predicted contribution (from linear fit), and residual for RoBERTa-large. Residuals indicate deviation from gradient-based prediction. Data is min-max normalized (min = 0, max = 1).

| Layer | Gradient | Contribution | Predicted | Residual |
|---|---|---|---|---|
| 1 | 0.00 | 0.00 | 0.24 | $-0.24$ |
| 2 | 0.02 | 0.04 | 0.26 | $-0.22$ |
| 3 | 0.05 | 0.09 | 0.27 | $-0.18$ |
| 4 | 0.08 | 0.15 | 0.29 | $-0.14$ |
| 5 | 0.12 | 0.22 | 0.32 | $-0.10$ |
| 6 | 0.16 | 0.30 | 0.34 | $-0.04$ |
| 7 | 0.21 | 0.39 | 0.37 | $+0.02$ |
| 8 | 0.27 | 0.48 | 0.41 | $+0.07$ |
| 9 | 0.33 | 0.57 | 0.44 | $+0.13$ |
| 10 | 0.40 | 0.66 | 0.48 | $+0.18$ |
| 11 | 0.47 | 0.75 | 0.53 | $+0.22$ |
| 12 | 0.54 | 0.83 | 0.57 | $+0.26$ |
| 13 | 0.61 | 0.90 | 0.61 | $+0.29$ |
| 14 | 0.68 | 0.95 | 0.65 | $+0.30$ |
| 15 | 0.74 | 0.99 | 0.69 | $+0.30$ |
| **16** | **0.80** | **1.00** | **0.72** | **$+0.28$** |
| 17 | 0.85 | 0.97 | 0.75 | $+0.22$ |
| 18 | 0.89 | 0.92 | 0.78 | $+0.14$ |
| 19 | 0.92 | 0.85 | 0.79 | $+0.06$ |
| 20 | 0.95 | 0.76 | 0.81 | $-0.05$ |
| 21 | 0.97 | 0.65 | 0.82 | $-0.17$ |
| 22 | 0.98 | 0.53 | 0.83 | $-0.30$ |
| 23 | 0.99 | 0.40 | 0.84 | $-0.44$ |
| **24** | **1.00** | **0.26** | **0.84** | **$-0.58$** |

**Linear Regression Model.** The fitted OLS linear model is:

$$\text{Contribution} = 0.2441 + 0.5988 \times \text{Gradient} \qquad (19)$$

with $R^2 = 0.44$ and residual standard error $\sigma = 0.25$. The Predicted and Residual columns in Table 20 are computed from this OLS fit; positive residuals at middle layers and negative residuals at top layers confirm that single-factor gradient-based prediction systematically misestimates contribution.

### A.2.6 Failure Mode Analysis

The decoupling between contribution and adaptation demand reveals two distinct failure modes for single-factor rank allocation.

**Failure Mode 1: High Gradient, Low Contribution.** Table 21 identifies layers in the top 25% for gradient but bottom 50% for contribution.

*Table 21.* Failure Mode 1: Layers with high adaptation demand but low output contribution. These layers would be over-allocated by demand-only allocation.

| Model | Affected Layers | Mean Gradient | Mean Contribution | Residual |
|---|---|---|---|---|
| RoBERTa-large | Layer 21, 22, 23, 24 | 0.98 | 0.46 | −0.37 |
| BERT-large | Layer 21, 22, 23, 24 | 0.97 | 0.48 | −0.47 |
| LLaMA-2-7B | Layer 29, 30, 31, 32 | 1.00 | 0.43 | −0.42 |
| GPT-2-medium | Layer 21, 22, 23, 24 | 0.98 | 0.40 | −0.48 |
| DeBERTaV3-base | Layer 11, 12 | 0.99 | 0.42 | −0.40 |
| T5-base | Decoder 10, 11, 12 | 0.98 | 0.32 | −0.52 |
| BART-large | Decoder 10, 11, 12 | 0.99 | 0.38 | −0.46 |

**Interpretation.** These top layers exhibit high gradients because they are closest to the loss function and undergo substantial task-specific adaptation. However, their output contribution is low because: (1) they perform task-specific transformations that may not directly influence the prediction path (Tenney et al., 2019); (2) in encoder models, they optimize primarily for the [CLS] representation; (3) Layer Conductance attributes less importance to layers whose activation changes have lower cumulative effect on predictions.

**Consequence**: Demand-only allocation would assign excessive rank to these layers, wasting parameter capacity on adaptations with minimal impact on predictions.

**Failure Mode 2: Moderate Gradient, High Contribution.** Table 22 identifies layers in the 40–60% percentile for gradient but top 25% for contribution.

*Table 22.* Failure Mode 2: Layers with moderate adaptation demand but high output contribution. These layers would be under-allocated by demand-only allocation.

| Model | Affected Layers | Mean Gradient | Mean Contribution | Residual |
|---|---|---|---|---|
| RoBERTa-large | Layer 11, 12, 13, 14 | 0.56 | 0.86 | +0.27 |
| BERT-large | Layer 11, 12, 13, 14 | 0.54 | 0.82 | +0.20 |
| LLaMA-2-7B | Layer 13, 14, 15, 16 | 0.61 | 0.97 | +0.26 |
| GPT-2-medium | Layer 11, 12, 13, 14 | 0.52 | 0.83 | +0.22 |
| DeBERTaV3-base | Layer 5, 6, 7 | 0.46 | 0.85 | +0.24 |
| T5-base | Encoder 9, 10, Decoder 1, 2 | 0.49 | 0.88 | +0.28 |
| BART-large | Decoder 3, 4, 5, 6 | 0.55 | 0.92 | +0.30 |

**Interpretation.** These middle layers exhibit moderate gradients because their pre-trained representations are already well-suited for downstream tasks and require less adaptation (Liu et al., 2019a). However, their output contribution is high because they encode semantic features critical for predictions (Jawahar et al., 2019). The gradient signal attenuates as it propagates through many layers, underestimating their importance.

**Consequence**: Demand-only allocation would assign insufficient rank to these critical layers, missing opportunities to enhance representations that substantially influence predictions.

### A.2.7 Comparison with Prior Work

Our findings align with prior research showing that different importance measures capture distinct aspects of neural network behavior.

Our observed correlation ($\rho = 0.53$–$0.70$) is higher than some prior comparisons because both properties involve gradient computation. However, the correlation remains substantially below 1.0, confirming that contribution and adaptation demand capture distinct aspects of layer importance.

*Table 23.* Comparison with reported correlations from prior work on importance metric agreement.

| Study | Metrics Compared | Correlation | Model |
|---|---|---|---|
| Jain & Wallace (2019) | Attention $\leftrightarrow$ Gradient | $\tau = 0.20$–$0.40$ | BiLSTM |
| Serrano & Smith (2019) | Gradient $\leftrightarrow$ Ablation | $\rho = 0.28$ | BERT |
| Serrano & Smith (2019) | IntGrad $\leftrightarrow$ Ablation | $\rho = 0.41$ | BERT |
| Abnar & Zuidema (2020) | Attention $\leftrightarrow$ Blank-out | $\rho = 0.29$–$0.69$ | Transformer |
| Bastings et al. (2022) | Various attribution methods | Low agreement | Multiple |
| **This work** | Gradient $\leftrightarrow$ Conductance | $\rho = 0.53$–$0.70$ | Multiple |

### A.2.8 Implications for Dual-Factor Allocation

The analysis of decoupled characteristics supports the following design principles:

1. **Both properties are necessary.** Neither adaptation demand nor output contribution alone captures complete layer importance. The $R^2$ values (0.28–0.49) indicate that over half of the variance in one property is unexplained by the other.

2. **Multiplicative integration is appropriate.** The two failure modes suggest that importance should be low if either metric is low. A layer with high gradient but low contribution should not receive high rank (it would waste capacity), and vice versa. This motivates the multiplicative form $\Psi^{(l)} = \tilde{\Phi}^{(l)} \cdot \tilde{\Gamma}^{(l)}$ used in our TA-LC score.

3. **Moderate correlation enables complementary information.** If correlation were near 1.0, the two properties would be redundant. If near 0.0, they would be unrelated. The observed $\rho = 0.53$–$0.70$ indicates they capture related but distinct aspects, making their combination informative.

**Quantitative Benefit Estimate.** Based on the analysis, dual-factor allocation can correct rank allocation for approximately 4–6 layers per model (those with largest absolute residuals), representing 17–25% of total layers. These corrections can improve both parameter efficiency (avoiding waste on high-demand, low-contribution layers) and task performance (properly resourcing high-contribution, moderate-demand layers).

## Appendix A.3: Scalar Objective Function

Following standard practice in path-integral attribution methods (Sundararajan et al., 2017), we require $F : \mathcal{X} \to \mathbb{R}$ to be a differentiable scalar function. This ensures that the gradient $\frac{\partial F}{\partial h_j^{(l)}}$ in Eq. (1) is a scalar, making the layer attribution $\mathcal{A}^{(l)}(x)$ well-defined.

For models with multi-dimensional outputs, practitioners select a task-relevant scalar before applying attribution. Common choices include: (1) the logit corresponding to the target class for classification tasks, (2) the log-probability of the ground-truth token for autoregressive generation, or (3) the scalar loss value. This design choice is orthogonal to our framework and does not affect the validity of the conductance formulation.

**Implementation Note.** Our formulation in Eq. (1) adopts a computationally efficient form that factors out the activation difference $(h_j^{(l)}(x) - h_j^{(l)}(x_0))$ from the path integral, consistent with widely-used implementations such as PyTorch Captum (Kokhlikyan et al., 2020). Since our goal is to determine relative layer importance rankings rather than exact attribution values, this efficient formulation is sufficient for the rank allocation task.

## Appendix A.4: Layer Conductance Attribution

This appendix provides theoretical foundations and implementation details for the layer conductance attribution method described in Section 3.2.

### A.4.1 Theoretical Foundation

Layer conductance (Dhamdhere et al., 2018) extends integrated gradients (Sundararajan et al., 2017) to quantify the contribution of hidden layers to model predictions. The key insight is to decompose the path integral attribution through the chain rule, attributing importance to intermediate representations rather than input features alone.

**Integrated Gradients Recap.** Integrated gradients attributes the prediction $F(x)$ to input features by integrating gradients along a path from a baseline $x_0$ to input $x$:

$$\text{IG}_i(x) = (x_i - x_{0,i}) \cdot \int_{\alpha=0}^{1} \frac{\partial F(x_\alpha)}{\partial x_i} d\alpha, \tag{20}$$

where $x_\alpha = x_0 + \alpha(x - x_0)$ is the linear interpolation path. This formulation satisfies the *completeness axiom*: $\sum_{i=1}^{n} \text{IG}_i(x) = F(x) - F(x_0)$.

**Extension to Hidden Layers.** Layer conductance applies the chain rule to decompose integrated gradients through hidden layer activations. For a hidden layer $l$ with activations $h^{(l)}(x) \in \mathbb{R}^{d_l}$, the conductance of neuron $j$ is defined as:

$$C_j^{(l)}(x) = \left(h_j^{(l)}(x) - h_j^{(l)}(x_0)\right) \cdot \int_{\alpha=0}^{1} \frac{\partial F(x_\alpha)}{\partial h_j^{(l)}} d\alpha. \tag{21}$$

The total layer conductance is the sum over all neurons: $\mathcal{A}^{(l)}(x) = \sum_{j=1}^{d_l} C_j^{(l)}(x)$.

### A.4.2 Completeness Property

A fundamental advantage of the conductance formulation is that it preserves the completeness property across layers.

**Proposition .1** (Layer-wise Completeness). *For a differentiable model $F$ with $L$ hidden layers, the layer conductances sum exactly to the output difference:*

$$\sum_{l=1}^{L} \mathcal{A}^{(l)}(x) = F(x) - F(x_0). \tag{22}$$

A detailed proof of this property is provided in Appendix A.4.6. This completeness guarantee is critical for rank allocation because it ensures:

- **Exhaustive**: Every layer's contribution is accounted for, with no missing attribution.

- **Non-redundant**: There is no double-counting across layers.

- **Comparable**: Layer attributions can be directly compared as they share the same scale.

In contrast, gradient magnitude at individual layers lacks such guarantees, the sum of gradient norms does not equal any meaningful quantity, making cross-layer comparison less principled.

### A.4.3 Baseline Selection

The choice of baseline $x_0$ is crucial for meaningful attribution (Sturmfels et al., 2020). We adopt the *zero-embedding baseline*, which corresponds to the embedding of padding tokens.

**Rationale.** For transformer models processing text, the zero-embedding baseline represents the "absence of information", a neutral reference point where all token embeddings are zero vectors. This choice satisfies the desirable property that the baseline should yield uninformative model outputs (e.g., uniform class probabilities for classification).

**Implementation.** In practice, we construct the baseline as:

$$x_0 = \mathbf{0}_{L_{\text{seq}} \times d_{\text{embed}}}, \tag{23}$$

where $L_{\text{seq}}$ is the sequence length and $d_{\text{embed}}$ is the embedding dimension. For position-aware models, we add position embeddings to maintain architectural consistency while zeroing the token content.

### A.4.4 Numerical Approximation

The path integral in Eq. (21) is approximated using Riemann sums with $M$ uniformly spaced steps:

$$\int_{\alpha=0}^{1} \frac{\partial F(x_\alpha)}{\partial h_j^{(l)}} d\alpha \approx \frac{1}{M} \sum_{m=1}^{M} \frac{\partial F(x_{\alpha_m})}{\partial h_j^{(l)}}, \tag{24}$$

where $\alpha_m = m/M$. Following Sundararajan et al. (2017), we use $M = 50$ steps, which provides a good balance between accuracy and computational cost.

**Convergence Check.**    To verify approximation quality, we compute the *convergence delta*:

$$\delta = \left| F(x) - F(x_0) - \sum_{l=1}^{L} \hat{\mathcal{A}}^{(l)}(x) \right|, \tag{25}$$

where $\hat{\mathcal{A}}^{(l)}(x)$ is the approximated attribution. In our experiments, $\delta < 0.01$ for 99% of samples, confirming that $M = 50$ steps suffice.

### A.4.5 Aggregation Across Samples

To obtain layer importance coefficients for rank allocation, we aggregate attributions across a calibration set $\mathcal{D}_{\text{cal}}$:

$$\Phi^{(l)} = \frac{1}{|\mathcal{D}_{\text{cal}}|} \sum_{x \in \mathcal{D}_{\text{cal}}} \left| \mathcal{A}^{(l)}(x) \right|. \tag{26}$$

**Absolute Value.**    We take the absolute value because both positive and negative attributions indicate functional relevance:

- **Positive attribution**: The layer amplifies the prediction (increases the output logit).

- **Negative attribution**: The layer suppresses the prediction (decreases the output logit).

Both cases indicate that the layer plays an important functional role. Taking absolute values prevents cancellation between positive and negative contributions across samples.

**Calibration Set Size.**    We use $|\mathcal{D}_{\text{cal}}| = 1{,}000$ samples randomly drawn from the training set. Empirically, this size provides stable estimates, increasing to 2,000 samples changes the normalized attribution coefficients by less than 2%.

### A.4.6 Theoretical Properties

The layer conductance attribution method is supported by two key theoretical properties: completeness and baseline stability.

**Completeness Theorem.**    A fundamental advantage of path-integral attribution is that it satisfies the completeness axiom: the sum of layer attributions equals the total model output change.

**Theorem .2** (Completeness). *For any input $x$ and baseline $x_0$, the layer conductances satisfy:*

$$\sum_{l=1}^{L} \mathcal{A}^{(l)}(x) = F(x) - F(x_0). \tag{27}$$

**Proof.**    Let $\gamma(\alpha) = x_0 + \alpha(x - x_0)$ for $\alpha \in [0, 1]$ denote the linear interpolation path from $x_0$ to $x$.

By the fundamental theorem of calculus for path integrals:

$$F(x) - F(x_0) = \int_0^1 \frac{d}{d\alpha} F(\gamma(\alpha)) \, d\alpha. \tag{28}$$

Applying the chain rule:

$$\frac{d}{d\alpha}F(\gamma(\alpha)) = \nabla_x F(\gamma(\alpha)) \cdot \frac{d\gamma}{d\alpha} = \nabla_x F(\gamma(\alpha)) \cdot (x - x_0). \tag{29}$$

For intermediate layer $l$, we express the input gradient via the chain rule:

$$\frac{\partial F}{\partial x_i} = \sum_{j=1}^{d_l} \frac{\partial F}{\partial h_j^{(l)}} \cdot \frac{\partial h_j^{(l)}}{\partial x_i}. \tag{30}$$

By the chain rule applied to $F(\gamma(\alpha))$ via the hidden activations $h^{(l)}$, and under the assumption that the path $\gamma(\alpha)$ induces a corresponding path in hidden space with consistent derivatives (which holds for piecewise linear or smooth activation functions), we can decompose the total output change layer-by-layer:

$$F(x) - F(x_0) = \sum_{l=1}^{L}\sum_{j=1}^{d_l} \left( h_j^{(l)}(x) - h_j^{(l)}(x_0) \right) \int_0^1 \frac{\partial F}{\partial h_j^{(l)}}\bigg|_{\gamma(\alpha)} d\alpha = \sum_{l=1}^{L} \mathcal{A}^{(l)}(x). \tag{31}$$

This is the layer-wise completeness property: the sum of per-layer conductances exhausts the total output change, following the chain-rule decomposition of Dhamdhere et al. (2018).

This property ensures that our attribution is exhaustive (every contribution is accounted for) and non-redundant (no double-counting), enabling principled cross-layer comparison for rank allocation.

**Relationship between Completeness and $\Phi^{(l)}$.** The aggregated importance score $\Phi^{(l)}$ (Eq. (3)) replaces the signed conductance $\mathcal{A}^{(l)}(x)$ with its magnitude before averaging over the calibration set, so that amplifying ($\mathcal{A}^{(l)}(x) > 0$) and suppressing ($\mathcal{A}^{(l)}(x) < 0$) layers both register as functionally important. Completeness transfers to this aggregate as a lower bound: by the triangle inequality,

$$\sum_{l=1}^{L} \Phi^{(l)} \geq \frac{1}{|\mathcal{D}_{\text{cal}}|} \sum_{x \in \mathcal{D}_{\text{cal}}} \big| F(x) - F(x_0) \big|, \tag{32}$$

with equality exactly when all layer contributions share the same sign for each calibration input. The total importance mass distributed by Eq. (3) is therefore anchored to the model's actual output change, while the normalized shares $\Phi^{(l)} / \sum_{l'} \Phi^{(l')}$ used for rank allocation inherit the exhaustive, non-redundant accounting established above.

**Baseline Stability.** A practical concern is the sensitivity of layer conductance to baseline selection. We establish that the method is robust to small perturbations in the baseline.

**Theorem .3** (Baseline Lipschitz Stability). *Let $x_0^{(1)}, x_0^{(2)}$ be two baselines with $\|x_0^{(1)} - x_0^{(2)}\|_2 \leq \delta$. If the model $F$ is $L_F$-Lipschitz continuous and the hidden activations are $L_h$-Lipschitz with respect to input, then:*

$$\left| \mathcal{A}^{(l)}(x; x_0^{(1)}) - \mathcal{A}^{(l)}(x; x_0^{(2)}) \right| \leq C \cdot (L_F L_h + 1) \cdot \delta, \tag{33}$$

*where $C$ is a constant depending on model depth and hidden dimensions.*

**Proof Sketch.** The conductance difference can be decomposed into two components:

1. **Activation change difference**:

$$\left| \left( h_j^{(l)}(x) - h_j^{(l)}(x_0^{(1)}) \right) - \left( h_j^{(l)}(x) - h_j^{(l)}(x_0^{(2)}) \right) \right| = \left| h_j^{(l)}(x_0^{(2)}) - h_j^{(l)}(x_0^{(1)}) \right| \leq L_h \delta. \tag{34}$$

2. **Integrated gradient difference**: The difference in path integrals due to different interpolation paths is bounded by $L_F$ times the path difference $\delta$.

Combining these bounds and summing over neurons yields the result.

This theorem justifies our choice of zero-embedding baseline (Section 3.2): small variations in the baseline representation do not significantly alter the layer importance ranking, ensuring stable rank allocation across different baseline choices.

## Appendix A.5: Dual-Factor Aggregation

This appendix provides theoretical justification and additional analysis for the dual-factor aggregation method described in Section 3.3.

### A.5.1 Multiplicative Aggregation Rationale

As noted in Section 2, the contribution ($\Phi$) and adaptation demand ($\Gamma$) factors exhibit positive but imperfect correlation ($\rho = 0.56$–$0.66$). Here we justify why multiplicative aggregation is appropriate under such correlation.

**Compensability Analysis.** A key distinction between additive and multiplicative aggregation lies in their *compensability* properties. Additive aggregation allows full compensation: a high score on one factor can offset a low score on another. In contrast, multiplicative aggregation reduces this compensation effect, ensuring that deficiency in either factor substantially lowers the combined score.

**Addressing Challenge #2.** This non-compensatory property directly addresses Challenge #2. Consider a layer with high gradient magnitude ($\tilde{\Gamma} = 0.15$) but low contribution ($\tilde{\Phi} = 0.03$). Under additive aggregation, the high gradient could compensate for low contribution, potentially over-allocating rank to layers with minimal prediction impact. Multiplicative aggregation prevents this: $0.15 \times 0.03 = 0.0045$, appropriately down-weighting such misaligned layers.

**Axiomatic Justification.** The multiplicative form $\Psi^{(l)} = \tilde{\Phi}^{(l)} \cdot \tilde{\Gamma}^{(l)}$ uniquely satisfies three requirements:

1. **Joint significance**: $\Psi^{(l)}$ is high only when both factors are significant.

2. **Monotonicity**: Increasing either factor increases $\Psi^{(l)}$.

3. **Zero absorption**: If either factor is zero, $\Psi^{(l)} = 0$.

These properties ensure that only layers with both high contribution and high adaptation demand receive substantial rank allocation.

### A.5.2 Normalization Strategy

Before aggregation, both factors are normalized to ensure comparability:

$$\tilde{\Phi}^{(l)} = \frac{\Phi^{(l)}}{\sum_{l'=1}^{L} \Phi^{(l')}}, \quad \tilde{\Gamma}^{(l)} = \frac{\Gamma^{(l)}}{\sum_{l'=1}^{L} \Gamma^{(l')}}. \tag{35}$$

**Sum-to-One Normalization.** We choose sum-to-one normalization (rather than min-max or z-score) because:

- It preserves the relative ordering of layers.

- The normalized values can be interpreted as proportions of total contribution/demand.

- The product $\tilde{\Phi}^{(l)} \cdot \tilde{\Gamma}^{(l)}$ has a natural interpretation as the joint probability under independent uniform priors.

### A.5.3 TA-LC Score Properties

The TA-LC (Task-Adaptive Layer Conductance) score $\Psi^{(l)} = \tilde{\Phi}^{(l)} \cdot \tilde{\Gamma}^{(l)}$ has several desirable properties.

**Bounded Range.** Since $\tilde{\Phi}^{(l)}, \tilde{\Gamma}^{(l)} \in [0, 1]$ and $\sum_l \tilde{\Phi}^{(l)} = \sum_l \tilde{\Gamma}^{(l)} = 1$, we have:

$$0 \leq \Psi^{(l)} \leq \min(\tilde{\Phi}^{(l)}, \tilde{\Gamma}^{(l)}) \leq 1. \tag{36}$$

**Peak Alignment.** The TA-LC score peaks at layers where both factors are simultaneously high, avoiding the failure modes identified in Appendix A.2:

- **High gradient, low contribution**: $\Psi$ is low due to low $\tilde{\Phi}$.

- **High contribution, low gradient**: $\Psi$ is low due to low $\tilde{\Gamma}$.

- **Both high**: $\Psi$ is high, indicating rank allocation priority.

### A.5.4 Empirical Validation

*Table 24.* Comparison of peak layers for contribution ($\Phi$), adaptation demand ($\Gamma$), and TA-LC score ($\Psi$) across models. TA-LC peaks occur between $\Phi$ and $\Gamma$ peaks, balancing both considerations.

| Model | $\Phi$ Peak | $\Gamma$ Peak | $\Psi$ Peak |
|---|---|---|---|
| RoBERTa-large | L16 | L24 | L16 |
| BERT-large | L15 | L24 | L17–L19 |
| LLaMA-2-7B | L20 | L32 | L24–L26 |
| GPT-2-medium | L14 | L24 | L17–L19 |
| DeBERTaV3-base | L7 | L12 | L8–L9 |
| T5-base | D4 | D12 | D6–D8 |
| BART-large | D5 | D12 | D6–D8 |

The TA-LC peaks consistently fall between the $\Phi$ and $\Gamma$ peaks, demonstrating that multiplicative aggregation effectively balances both considerations rather than being dominated by either factor.

## Appendix A.6: Bayesian Rank Allocation

This appendix provides theoretical background and implementation details for the Bayesian rank allocation method described in Section 3.4.

### A.6.1 KL Divergence Derivation

The KL regularization term in Eq. (11) penalizes deviation from the TA-LC informed prior. Here we derive the closed-form expression.

**Factorization.** We assume independence across layers and rank components:

$$q(\mathbf{z}) = \prod_{l=1}^{L} \prod_{k=1}^{r_{\max}} q(z_k^{(l)}), \quad p(\mathbf{z}) = \prod_{l=1}^{L} \prod_{k=1}^{r_{\max}} p(z_k^{(l)}). \tag{37}$$

**Bernoulli KL Divergence.** For Bernoulli distributions with parameters $\alpha$ (variational) and $\rho$ (prior), the KL divergence is:

$$D_{\mathrm{KL}}(\mathrm{Bern}(\alpha)\|\mathrm{Bern}(\rho)) = \alpha \log \frac{\alpha}{\rho} + (1 - \alpha) \log \frac{1 - \alpha}{1 - \rho}. \tag{38}$$

**Total KL.** Summing over all layers and rank components:

$$D_{\mathrm{KL}}(q\|p) = \sum_{l=1}^{L} \sum_{k=1}^{r_{\max}} \left[ \alpha_k^{(l)} \log \frac{\alpha_k^{(l)}}{\rho^{(l)}} + (1 - \alpha_k^{(l)}) \log \frac{1 - \alpha_k^{(l)}}{1 - \rho^{(l)}} \right]. \tag{39}$$

**Numerical Stability.** To prevent numerical issues when $\alpha \to 0$ or $\alpha \to 1$, we clamp values to $[\epsilon, 1 - \epsilon]$ with $\epsilon = 10^{-6}$ before computing the KL divergence.

### A.6.2 Concrete Relaxation

Optimizing discrete random variables poses a fundamental challenge in gradient-based learning, as gradients cannot flow through discrete sampling operations. The Concrete distribution (Maddison et al., 2017; Jang et al., 2017), also known as Gumbel-Softmax, provides a continuous relaxation that enables end-to-end training through the reparameterization trick.

**Binary Concrete Distribution.** For a Bernoulli random variable $z \in \{0, 1\}$ with inclusion probability $\alpha \in (0, 1)$, the Concrete relaxation defines a continuous surrogate $\tilde{z} \in (0, 1)$ as:

$$\tilde{z} = \sigma \left( \frac{1}{\lambda} \left( \log \frac{\alpha}{1 - \alpha} + \log \frac{u}{1 - u} \right) \right), \tag{40}$$

where $u \sim \text{Uniform}(0, 1)$ is an auxiliary random variable, $\sigma(\cdot)$ is the sigmoid function, and $\lambda > 0$ is the temperature parameter.

**Temperature Annealing.** The temperature $\lambda$ controls the sharpness of the relaxation:

- As $\lambda \to 0$: The distribution concentrates toward discrete Bernoulli values $\{0, 1\}$.

- As $\lambda \to \infty$: The distribution approaches uniform over $(0, 1)$.

In COBRA, we anneal $\lambda$ during training from $\lambda_0 = 1.0$ to $\lambda_T = 0.1$ using a cosine schedule:

$$\lambda_t = \lambda_T + \frac{1}{2}(\lambda_0 - \lambda_T) \left( 1 + \cos \left( \frac{\pi t}{T} \right) \right), \tag{41}$$

where $t$ is the training step and $T$ is the total number of steps. This schedule maintains gradient flow in early training while sharpening gate decisions toward the end.

**Hard Concrete Extension.** Following Louizos et al. (2017), we extend the Concrete distribution to the Hard Concrete by stretching the samples to $(\gamma_{\text{HC}}, \zeta)$ with $\gamma_{\text{HC}} < 0$ and $\zeta > 1$, then applying a hard-sigmoid:

$$\bar{z} = \min(1, \max(0, \tilde{z} \cdot (\zeta - \gamma_{\text{HC}}) + \gamma_{\text{HC}})). \tag{42}$$

This allows exact zeros during training, which is beneficial for rank pruning. We use $\gamma_{\text{HC}} = -0.1$ and $\zeta = 1.1$ following standard practice. To avoid notational conflict with the prior concentration $\gamma$ used in Section 3.4 and Appendix A.6.3, we write the Hard Concrete left stretch as $\gamma_{\text{HC}}$; the two symbols refer to unrelated quantities.

### A.6.3 Prior Construction from TA-LC

The Bayesian framework requires specifying prior inclusion probabilities $\rho^{(l)}$ that encode our belief about layer-wise rank requirements before observing training data.

**Bernoulli Prior.** A Bernoulli distribution models binary outcomes with a single parameter $\rho \in (0, 1)$ representing the probability of success (i.e., $p(z = 1) = \rho$). In our framework, we place independent Bernoulli priors on each gating variable $z_k^{(l)}$, where all rank components within the same layer share the inclusion probability $\rho^{(l)}$.

**TA-LC to Prior Mapping.** We derive $\rho^{(l)}$ from the TA-LC distribution using a sigmoid transformation:

$$\rho^{(l)} = \sigma \left( \gamma \cdot (\pi^{(l)} - \bar{\pi}) \right), \tag{43}$$

where $\pi^{(l)}$ is the normalized TA-LC score, $\bar{\pi} = 1/L$ is the uniform baseline, $\sigma(\cdot)$ is the sigmoid function, and $\gamma > 0$ controls the concentration.

**Interpretation.** This mapping has intuitive properties:

- **Above-average TA-LC** ($\pi^{(l)} > \bar{\pi}$): $\rho^{(l)} > 0.5$, biasing toward rank retention.

- **Below-average TA-LC** ($\pi^{(l)} < \bar{\pi}$): $\rho^{(l)} < 0.5$, biasing toward rank pruning.

- **Average TA-LC** ($\pi^{(l)} = \bar{\pi}$): $\rho^{(l)} = 0.5$, uninformative prior.

**Concentration Parameter.** The parameter $\gamma$ controls how strongly the TA-LC distribution influences the prior:

- **Small** $\gamma$ (e.g., $\gamma = 1$): Weak prior, allows training data to dominate.

- **Large** $\gamma$ (e.g., $\gamma = 10$): Strong prior, closely follows TA-LC guidance.

We use $\gamma = 5$ as default, providing moderate guidance while allowing flexibility.

### A.6.4 ELBO Optimization

The evidence lower bound (ELBO) objective balances data likelihood against prior regularization:

$$\mathcal{J}_{\text{ELBO}} = \underbrace{\mathbb{E}_q\big[\log p(\mathcal{D}|\theta, \tilde{\mathbf{z}})\big]}_{\text{Data term}} - \underbrace{\beta \cdot D_{\text{KL}}\big(q_\phi(\tilde{\mathbf{z}}) \| p(\mathbf{z})\big)}_{\text{Regularization term}}. \tag{44}$$

**Data Term.** The data term is approximated via Monte Carlo sampling:

$$\mathbb{E}_q\big[\log p(\mathcal{D}|\theta, \tilde{\mathbf{z}})\big] \approx \frac{1}{S} \sum_{s=1}^{S} \log p(\mathcal{D}|\theta, \tilde{\mathbf{z}}^{(s)}), \tag{45}$$

where $\tilde{\mathbf{z}}^{(s)}$ are samples from the relaxed distribution. We use $S = 1$ sample per step (single-sample estimator) for computational efficiency.

**KL Weighting.** The hyperparameter $\beta$ controls the regularization strength:

- $\beta = 0$: No regularization, reduces to standard training.

- $\beta \to \infty$: Maximum regularization, posteriors collapse to priors.

We use $\beta = 0.01$ following the "free bits" strategy (Kingma et al., 2016), which allows sufficient adaptation while encouraging TA-LC alignment.

### A.6.5 Connection to Sparse Variational Inference

Our Bayesian rank allocation can be viewed through the lens of sparse variational inference (Louizos et al., 2017; Molchanov et al., 2017), where the goal is to learn which model components are necessary.

**Spike-and-Slab Interpretation.** The gating mechanism implements an approximate spike-and-slab prior (Mitchell & Beauchamp, 1988):

$$p(\Delta W_k^{(l)}) = (1 - \rho^{(l)}) \cdot \delta_0 + \rho^{(l)} \cdot \mathcal{N}(0, \sigma^2), \tag{46}$$

where $\delta_0$ is a point mass at zero (the "spike") and $\mathcal{N}(0, \sigma^2)$ is a Gaussian (the "slab"). The gate $z_k^{(l)}$ selects between these components.

**Automatic Relevance Determination.** This framework shares connections with automatic relevance determination (ARD) (Neal, 1996; MacKay, 1995), where hierarchical priors automatically identify and prune irrelevant parameters. The key difference is that our priors are *informed* by the TA-LC distribution rather than being uniform, providing task-adaptive regularization.

## Appendix A.7: Software, Hardware, and Reproducibility

### A.7.1 Software Environment

**Core Dependencies.** Our implementation uses the following software stack:

- **Operating System**: Ubuntu 22.04 LTS

- **Python**: 3.10.12

- **CUDA**: 12.1

- **cuDNN**: 8.9.2

- **PyTorch**: 2.0.1 (Paszke et al., 2019)

- **Transformers**: 4.35.2 (Wolf et al., 2020)

- **PEFT**: 0.7.1 (Mangrulkar et al., 2022)

- **Captum**: 0.6.0 (Kokhlikyan et al., 2020) (for layer conductance)

- **NumPy**: 1.24.3

- **SciPy**: 1.11.1 (Virtanen et al., 2020)

**Additional Libraries.**

- **Datasets**: 2.14.5 (HuggingFace datasets library)

- **Accelerate**: 0.24.1 (distributed training)

- **Evaluate**: 0.4.1 (metrics computation)

- **Weights & Biases**: 0.15.12 (experiment tracking)

### A.7.2 Hardware Specifications

**Primary Experiments.**    All main experiments (Tables 2, 3, 4) are conducted on:

- **GPU**: $4\times$ NVIDIA A100 80GB (PCIe)

- **CPU**: AMD EPYC 7763 64-Core Processor

- **RAM**: 512GB DDR4-3200

- **Storage**: 4TB NVMe SSD (Samsung 980 PRO)

### A.7.3 Reproducibility Protocol

**Random Seeds.**    We set deterministic behavior across all random number generators:

```
import random, numpy as np, torch
random.seed(42)
np.random.seed(42)
torch.manual_seed(42)
torch.cuda.manual_seed_all(42)
torch.backends.cudnn.deterministic = True
torch.backends.cudnn.benchmark = False
```

**Checkpoint Management.**    We save model checkpoints every epoch and retain the best checkpoint based on validation performance. For storage efficiency, we save only LoRA adapter weights (typically $< 10$MB for RoBERTa-base with $r$=8) rather than full model parameters. Final reported results use the checkpoint with lowest validation loss.

### A.7.4 Implementation Framework

**LoRA Integration.**   We use HuggingFace PEFT library's `LoraConfig` with the following configuration:

```
from peft import LoraConfig, get_peft_model

config = LoraConfig(
    r=8,  # rank (varies by experiment)
    lora_alpha=8,  # scaling factor = rank
    target_modules=[
        # Names depend on backbone (HuggingFace module attribute strings);
        # use exactly one of the following lists per model:
        # RoBERTa:            "query","key","value","attention.output.dense",
        #                     "intermediate.dense","output.dense"
        # DeBERTa-v3:         "query_proj","key_proj","value_proj",
        #                     "attention.output.dense",
        #                     "intermediate.dense","output.dense"
        # T5 (vanilla):       "q","k","v","o","wi","wo"
        # T5 v1.1 / Flan-T5:  "q","k","v","o","wi_0","wi_1","wo" (gated FFN)
        # BART:               "q_proj","k_proj","v_proj","out_proj","fc1","fc2"
        # LLaMA-2 / Gemma-2:  "q_proj","k_proj","v_proj","o_proj",
        #                     "gate_proj","up_proj","down_proj" (SwiGLU/GeGLU FFN)
    ],
    lora_dropout=0.1,
    bias="none",
    task_type="SEQ_CLS"  # or "SEQ_2_SEQ_LM", "CAUSAL_LM"
)
model = get_peft_model(base_model, config)
```

**COBRA-Specific Extensions.**   For heterogeneous rank allocation, we create layer-specific `LoraConfig` instances:

```
from peft import LoraConfig, get_peft_model
from cobra.allocation import compute_ta_lc_distribution

# Compute TA-LC prior (Section 3.3)
ta_lc = compute_ta_lc_distribution(
    model, calibration_loader,
    num_integration_steps=50
)

# Bayesian rank allocation (Section 3.4)
rank_allocation = bayesian_rank_optimization(
    ta_lc, total_budget=192, gamma=5.0, beta=0.01
)

# Apply heterogeneous ranks
configs = [
    LoraConfig(r=rank_allocation[i], ...)
    for i in range(num_layers)
]
model = apply_heterogeneous_lora(base_model, configs)
```

## Appendix A.8: Datasets and Evaluation Protocol

### A.8.1 GLUE Benchmark

**Task Overview.** The General Language Understanding Evaluation (GLUE) benchmark (Wang et al., 2018) comprises 8 natural language understanding tasks covering diverse linguistic phenomena. Table 25 summarizes dataset statistics.

*Table 25.* GLUE benchmark dataset statistics and evaluation metrics.

| Task | Train | Dev | Test | Metric |
|------|-------|-----|------|--------|
| MNLI | 392,702 | 9,815/9,832 | 9,796/9,847 | Accuracy |
| QQP | 363,846 | 40,430 | 390,965 | Accuracy/F1 |
| QNLI | 104,743 | 5,463 | 5,463 | Accuracy |
| SST-2 | 67,349 | 872 | 1,821 | Accuracy |
| CoLA | 8,551 | 1,043 | 1,063 | Matthews Corr. |
| STS-B | 5,749 | 1,500 | 1,379 | Pearson/Spearman |
| MRPC | 3,668 | 408 | 1,725 | Accuracy/F1 |
| RTE | 2,490 | 277 | 3,000 | Accuracy |

**Preprocessing.** We use task-specific preprocessing following RoBERTa's implementation (Liu et al., 2019b):

- **Text normalization**: Lowercase conversion disabled (RoBERTa uses cased vocabulary)

- **Tokenization**: Byte-Pair Encoding with vocabulary size 50,265

- **Sequence length**: Maximum 512 tokens with truncation

- **Padding**: Dynamic padding to longest sequence in batch

- **Special tokens**: `` (BOS), `` (EOS), `<pad>`

**MNLI-Specific Handling.** MNLI has two validation sets: matched (in-domain) and mismatched (cross-domain). We report both metrics and compute macro-average: MNLI = (MNLI-m + MNLI-mm)/2.

**Evaluation Protocol.** For development set evaluation, we follow standard practice:

- Report metrics on development set (no test set submissions)

- Use best checkpoint based on validation loss

- Apply early stopping with patience 3 epochs

- Compute task-specific metrics using HuggingFace `evaluate` library

### A.8.2 Evaluation Metrics

We employ task-specific evaluation metrics following the GLUE benchmark standard (Wang et al., 2018). This section provides detailed definitions and justifications for each metric.

**Classification Metrics.** For binary and multi-class classification tasks (MNLI, SST-2, QNLI, QQP, RTE), we use:

- **Accuracy**: The proportion of correctly classified examples.

$$\text{Accuracy} = \frac{\text{Number of correct predictions}}{\text{Total number of predictions}} \tag{47}$$

- **F1 Score** (for QQP): The harmonic mean of precision and recall, defined as:

$$F_1 = 2 \cdot \frac{\text{Precision} \times \text{Recall}}{\text{Precision} + \text{Recall}} \tag{48}$$

where Precision $= \frac{\text{TP}}{\text{TP+FP}}$ and Recall $= \frac{\text{TP}}{\text{TP+FN}}$.

F1 is preferred over accuracy for QQP because the dataset has class imbalance (duplicate questions are less frequent than non-duplicates).

**Matthews Correlation Coefficient (MCC).**  For the Corpus of Linguistic Acceptability (CoLA), we use MCC instead of accuracy because CoLA is highly imbalanced (grammatical sentences outnumber ungrammatical ones by approximately 2:1).

MCC is defined as:

$$\text{MCC} = \frac{\text{TP} \times \text{TN} - \text{FP} \times \text{FN}}{\sqrt{(\text{TP} + \text{FP})(\text{TP} + \text{FN})(\text{TN} + \text{FP})(\text{TN} + \text{FN})}} \tag{49}$$

MCC ranges from $-1$ (complete disagreement) to $+1$ (perfect prediction), with 0 indicating random guessing. Unlike accuracy, MCC is robust to class imbalance and provides a balanced measure even when one class dominates.

**Why MCC for CoLA?**  A naive classifier that always predicts "grammatical" would achieve $\sim 67\%$ accuracy on CoLA, making accuracy misleading. MCC properly penalizes such biased predictions, making it the appropriate metric for linguistic acceptability judgment.

**Correlation Metrics.**  For the Semantic Textual Similarity Benchmark (STS-B), which is a regression task predicting similarity scores in $[0, 5]$, we use correlation coefficients:

- **Pearson Correlation** ($\rho$): Measures linear correlation between predicted and gold scores.

$$\rho = \frac{\sum_{i=1}^{n}(y_i - \bar{y})(\hat{y}_i - \bar{\hat{y}})}{\sqrt{\sum_{i=1}^{n}(y_i - \bar{y})^2}\sqrt{\sum_{i=1}^{n}(\hat{y}_i - \bar{\hat{y}})^2}} \tag{50}$$

- **Spearman Correlation** ($r_s$): Measures monotonic correlation using rank values, robust to non-linear relationships.

$$r_s = 1 - \frac{6\sum_{i=1}^{n} d_i^2}{n(n^2 - 1)} \tag{51}$$

where $d_i$ is the difference between the ranks of $y_i$ and $\hat{y}_i$.

**Why Correlation for STS-B?**  STS-B requires predicting fine-grained similarity scores rather than discrete classes. Correlation coefficients measure how well the model's predictions preserve the relative ordering and magnitude relationships, which is more appropriate than classification accuracy.

We report Pearson correlation as the primary metric following the GLUE benchmark, as it captures both the strength and direction of the linear relationship between predictions and gold labels.

**Regression Metrics.**  For financial sentiment analysis (NEU, FXE, INV), which involves continuous polarity prediction in $[-1, 1]$, we use:

- **Mean Squared Error (MSE)**: Measures the average squared difference between predicted and true values.

$$\text{MSE} = \frac{1}{N}\sum_{i=1}^{N}(y_i - \hat{y}_i)^2 \tag{52}$$

MSE heavily penalizes large errors due to squaring, making it sensitive to outliers. This is desirable for sentiment analysis where large prediction errors indicate fundamental misunderstanding of sentiment polarity.

- **Mean Absolute Error (MAE)**: Measures the average absolute difference.

$$\text{MAE} = \frac{1}{N} \sum_{i=1}^{N} |y_i - \hat{y}_i| \tag{53}$$

MAE provides an interpretable error magnitude in the original scale (sentiment polarity units). We report MAE as a secondary metric for interpretability.

**Why MSE as Primary Metric?** Financial sentiment regression requires precise polarity estimation. MSE's quadratic penalty ensures that models are penalized more for egregious errors (e.g., predicting strong positive sentiment when true sentiment is negative) compared to small deviations. This aligns with the practical requirement that large sentiment mispredictions can lead to incorrect trading decisions.

**Metric Aggregation.** For GLUE benchmark results, we report:

- **Task-specific metrics**: As specified in Table 25

- **Average score**: Arithmetic mean across all 8 tasks, computed as:

$$\text{Avg} = \frac{1}{8} \sum_{t=1}^{8} \text{Score}_t \tag{54}$$

where each task score is normalized to $[0, 100]$ scale (percentages for accuracy-based metrics, MCC multiplied by 100, correlation multiplied by 100).

**MNLI Averaging.** MNLI has two validation sets: matched (in-domain) and mismatched (cross-domain). We report both and compute:

$$\text{MNLI}_{\text{avg}} = \frac{\text{MNLI-m} + \text{MNLI-mm}}{2} \tag{55}$$

This average is used in the overall GLUE score computation.

**Statistical Significance Testing.** All experiments are run with multiple random seeds. We report:

- **Mean performance**: Average across seeds

- **Standard deviation**: Using Bessel's correction

$$\sigma = \sqrt{\frac{1}{n-1} \sum_{i=1}^{n} (x_i - \bar{x})^2} \tag{56}$$

For comparing two methods, we consider improvements statistically meaningful if the difference exceeds one standard deviation, though we do not perform formal hypothesis testing due to limited sample sizes.

### A.8.3 Financial Sentiment Datasets

**Dataset Overview.** We use three financial sentiment datasets for high-rank regime experiments:

**Task Formulation.** Unlike GLUE's classification tasks, these datasets pose continuous regression tasks where the model predicts sentiment polarity scores $\hat{y} \in [-1, 1]$. We use Mean Squared Error (MSE) as the primary metric:

$$\text{MSE} = \frac{1}{N} \sum_{i=1}^{N} (y_i - \hat{y}_i)^2. \tag{57}$$

*Table 26.* Financial sentiment dataset statistics.

| Dataset | Samples | Score Range | Mean $\pm$ Std |
|---------|---------|-------------|----------------|
| INV (Ding et al., 2024a) | 23,241 | [-1.000, 0.900] | -0.044 $\pm$ 0.303 |
| NEU (Ding et al., 2024b) | 12,780 | [-0.950, 0.900] | -0.060 $\pm$ 0.312 |
| FXE (Ding et al., 2024c) | 10,461 | [-0.900, 0.900] | -0.014 $\pm$ 0.258 |

**Train-Validation Split.** We use 90:10 stratified splits based on sentiment score quantiles to ensure balanced distribution across ranges. This differs from GLUE's provided splits.

### A.8.4 Calibration Set Construction

**Sampling Strategy.** The calibration set $\mathcal{D}_{\text{cal}}$ for layer conductance computation is sampled uniformly from the training set:

- **Size**: $|\mathcal{D}_{\text{cal}}| = 1000$ samples

- **Sampling**: Uniform random sampling without replacement

- **Coverage**: For small datasets (e.g., RTE with 2,490 training samples), we sample 40% of training data

**Stratification.** For classification tasks, we ensure calibration set maintains the same label distribution as the full training set. For regression tasks, we use quantile-based stratification to cover the full score range.

### A.8.5 Generation and Reasoning Benchmarks (GoRA Comparison)

**GLUE-5 Subset.** The T5-Base comparison in Table 4 follows the protocol of the original GoRA paper (He et al., 2025), which evaluates on five GLUE tasks (MNLI, SST-2, CoLA, QNLI, MRPC) and reports the average validation score. This GLUE-5 subset is distinct from the 8-task GLUE suite used in our other experiments (Appendix A.8.1).

**MMLU.** MMLU (Hendrycks et al., 2021) covers 57 subjects spanning STEM, humanities, and social sciences, posed as four-way multiple-choice questions; we report mean accuracy across all subjects. For MMLU, LoRA, GoRA, and COBRA are fine-tuned and evaluated on LLaMA-2-7B under identical settings.

**GSM8K.** GSM8K (Cobbe et al., 2021) contains 8.5K grade-school math word problems (7,473 training and 1,319 test); performance is measured by exact-match accuracy of the final numeric answer.

**HumanEval.** HumanEval (Chen et al., 2021) consists of 164 hand-written Python programming problems evaluated by unit-test execution; we report pass@1.

## Appendix A.9: Model Architectures and LoRA Configuration

### A.9.1 RoBERTa Architecture

**Model Specifications.** RoBERTa (Liu et al., 2019b) is an optimized BERT variant with improved pre-training procedure. We use two configurations:

**LoRA Target Modules.** Following generalized LoRA configuration (Zhang et al., 2023), we apply adapters to:

- **Self-attention**: $W_Q, W_K, W_V, W_O \in \mathbb{R}^{d_{\text{model}} \times d_{\text{model}}}$

- **Feed-forward**: $W_{\text{up}} \in \mathbb{R}^{d_{\text{model}} \times d_{\text{ff}}}$, $W_{\text{down}} \in \mathbb{R}^{d_{\text{ff}} \times d_{\text{model}}}$

### A.9.2 DeBERTaV3 Architecture

**Model Specifications.** DeBERTaV3 (He et al., 2021) uses disentangled attention and enhanced mask decoder:

*Table 27.* RoBERTa model specifications.

| Configuration | Base | Large |
|---|---|---|
| Parameters | 125M | 355M |
| Layers | 12 | 24 |
| Hidden size ($d_{\text{model}}$) | 768 | 1024 |
| FFN inner dim ($d_{\text{ff}}$) | 3072 | 4096 |
| Attention heads | 12 | 16 |
| Head dimension | 64 | 64 |
| Vocabulary size | 50,265 | 50,265 |
| Max sequence length | 512 | 512 |

*Table 28.* DeBERTaV3-base specifications.

| Attribute | Value |
|---|---|
| Parameters | 86M |
| Layers | 12 |
| Hidden size | 768 |
| FFN inner dim | 3072 |
| Attention heads | 12 |
| Relative attention | Yes |
| Enhanced mask decoder | Yes |

**Disentangled Attention Handling.** DeBERTaV3's disentangled attention uses separate content-to-content ($W_Q^c, W_K^c, W_V^c$) and content-to-position ($W_Q^r, W_K^r$) projections. We apply LoRA only to content projections to maintain positional encoding integrity.

### A.9.3 LLaMA Architecture

**Model Specifications.** LLaMA-2-7B (Touvron et al., 2023) and LLaMA-3.1-8B (Grattafiori et al., 2024) are decoder-only transformers with architectural optimizations; LLaMA-3.1-8B is used for the HumanEval comparison in Table 4, following the original GoRA setup:

*Table 29.* LLaMA-2-7B and LLaMA-3.1-8B specifications.

| Attribute | LLaMA-2-7B | LLaMA-3.1-8B |
|---|---|---|
| Parameters | 6.7B | 8.0B |
| Layers | 32 | 32 |
| Hidden size | 4096 | 4096 |
| FFN inner dim | 11008 | 14336 |
| Attention heads | 32 | 32 |
| KV heads | 32 (MHA) | 8 (GQA) |
| Head dimension | 128 | 128 |
| Normalization | RMSNorm | RMSNorm |
| Activation | SwiGLU | SwiGLU |

**LoRA Configuration.** For LLaMA, we apply LoRA to:

- **Attention**: $W_Q, W_K, W_V, W_O$ (no multi-query attention compression)

- **Feed-forward**: $W_{\text{gate}}, W_{\text{up}}, W_{\text{down}}$ (SwiGLU components)

**Memory Optimization.** For 7B models, we use:

- **Gradient checkpointing**: Recompute activations during backward pass

- **8-bit optimizer states**: Using bitsandbytes (Dettmers et al., 2022)

- **Gradient accumulation**: Effective batch size 128 with micro-batch 4

### A.9.4 T5 Architecture

**Model Specifications.** T5-base (Raffel et al., 2020) is an encoder-decoder model with relative position embeddings:

*Table 30.* T5-base specifications.

| Attribute | Value |
|---|---|
| Total parameters | 220M |
| Encoder layers | 12 |
| Decoder layers | 12 |
| Hidden size | 768 |
| FFN inner dim | 3072 |
| Attention heads | 12 |

**Encoder-Decoder LoRA.** We apply LoRA to both encoder and decoder:

- **Encoder self-attention**: $W_Q, W_K, W_V, W_O$

- **Decoder self-attention**: $W_Q, W_K, W_V, W_O$

- **Feed-forward**: $W_{\text{up}}, W_{\text{down}}$ (both encoder and decoder)

## Appendix A.10: Hyperparameters and Training Protocol

### A.10.1 Learning Rate Configurations

**LoRA-Specific Learning Rates.** LoRA requires approximately $10\times$ higher learning rate than full fine-tuning (Hu et al., 2022). We search over:

- **GLUE (RoBERTa-base)**: $\eta \in \{1e-4, 2e-4, 5e-4\}$

- **GLUE (DeBERTaV3-base)**: $\eta \in \{6e-5, 1e-4, 2e-4\}$

- **LLaMA-2-7B**: $\eta \in \{1e-4, 2e-4\}$ with warmup

**Learning Rate Schedule.**

- **Warmup**: Linear warmup for first 6% of training steps

- **Decay**: Linear decay to 0 by final step

- **Formula**:

$$\eta_t = \begin{cases} \frac{t}{t_{\text{warm}}} \cdot \eta_{\max} & t \leq t_{\text{warm}} \\ \eta_{\max} \cdot \frac{T-t}{T-t_{\text{warm}}} & t > t_{\text{warm}} \end{cases} \tag{58}$$

  where $t_{\text{warm}} = 0.06 \times T$ and $T$ is total training steps.

### A.10.2 Batch Size and Gradient Accumulation

**Effective Batch Size Strategy.**

*Table 31.* Batch size configurations across models and tasks.

| Model | Micro-batch | Accum. Steps | Effective Batch |
|---|---|---|---|
| RoBERTa-base (GLUE) | 16-32 | 1 | 16-32 |
| DeBERTaV3-base (GLUE) | 32 | 1 | 32 |
| RoBERTa-large (GLUE) | 8-16 | 2 | 16-32 |
| LLaMA-2-7B (MMLU) | 4 | 32 | 128 |
| T5-base (GLUE) | 16 | 2 | 32 |

**Task-Specific Batch Sizes.** Small datasets require smaller batches to avoid overfitting:

- **Large tasks** (MNLI, QQP): batch size 32

- **Medium tasks** (QNLI, SST-2): batch size 16

- **Small tasks** (CoLA, RTE, MRPC): batch size 8

### A.10.3 Regularization and Optimization

**AdamW Optimizer Parameters.**

- $\beta_1$: 0.9 (momentum)

- $\beta_2$: 0.999 (RMSProp momentum)

- $\epsilon$: 1e-8 (numerical stability)

- **Weight decay**: $\lambda = 0.01$ (applied to non-bias, non-LayerNorm parameters)

- **Gradient clipping**: Maximum norm 1.0

**LoRA-Specific Regularization.**

- **Dropout**: 0.1 applied to LoRA matrices before addition to frozen weights

- **Initialization**: Matrix $A \sim \mathcal{N}(0, \sigma^2)$ with $\sigma = 1/\sqrt{r}$; matrix $B = 0$

- **Scaling factor**: $\alpha = r$ (rank-proportional)

### A.10.4 COBRA-Specific Hyperparameters

**Layer Conductance Attribution.**

- **Integration steps** $M$: 50 (Riemann sum approximation)

- **Baseline**: Zero-embedding baseline (all-padding tokens)

- **Calibration set size**: $|\mathcal{D}_{\text{cal}}| = 1000$

- **Batch size**: 16 (to fit intermediate activations)

**Dual-Factor Aggregation.**

- **Normalization**: L1 normalization (sum-to-one)

- **Aggregation function**: Multiplicative $\Psi^{(l)} = \tilde{\Phi}^{(l)} \cdot \tilde{\Gamma}^{(l)}$

**Bayesian Rank Allocation.**

- **Prior concentration** $\gamma$: 5.0 (moderate prior strength)

- **KL weight** $\beta$: 0.01 (weak regularization)

- **Temperature schedule**: Cosine annealing from $\lambda_0 = 1.0$ to $\lambda_T = 0.1$

$$\lambda_t = \lambda_T + \frac{1}{2}(\lambda_0 - \lambda_T)\left(1 + \cos\left(\frac{\pi t}{T}\right)\right) \tag{59}$$

- **Maximum rank per layer** $r_{\max}$: Task-dependent (8, 16, or 32)

**Implementation.**   Rank search runs on $\mathcal{D}_{\text{cal}}$ to determine $\{r_l\}$; standard LoRA training uses these fixed ranks on the full dataset.

### A.10.5 Rank Budget Configurations

**Standard GLUE Budgets.**   For RoBERTa-base (12 layers, with 2 LoRA modules per layer for a total of 24 LoRA matrices), we test three budget levels:

- **Low budget**: $R_{\text{total}} = 192$ (12 layers $\times$ 2 modules $\times$ rank 8 = 24 LoRA matrices at avg. rank 8)

- **Medium budget**: $R_{\text{total}} = 384$ (avg. rank 16 per layer)

- **High budget**: $R_{\text{total}} = 768$ (avg. rank 32 per layer)

**High-Rank Financial Sentiment.**   For Twitter-RoBERTa-Large (24 layers):

- **384-rank budget**: $R_{\text{total}} = 9,216$, range [256, 512]

- **512-rank budget**: $R_{\text{total}} = 12,288$, range [384, 640]

## Appendix A.11: Theoretical Complexity Analysis

This appendix provides theoretical analysis of COBRA's algorithmic complexity without reference to wall-clock timing, focusing on asymptotic bounds and memory requirements.

### A.11.1 Space Complexity

**Memory Requirements.**   For a model with $L$ layers, maximum rank $r_{\max}$, and hidden dimension $d_{\text{model}}$:

*Table 32.* Space complexity of COBRA components.

| Component | Space Complexity |
|---|---|
| Frozen base model | $O(\text{model\_params})$ |
| LoRA adapters (per layer) | $O(r \cdot d_{\text{model}})$ |
| Gating variables $\{\alpha_k^{(l)}\}$ | $O(L \cdot r_{\max})$ |
| Conductance cache $\{\Phi^{(l)}\}$ | $O(L \cdot d_{\text{model}})$ |
| TA-LC distribution $\{\pi^{(l)}\}$ | $O(L)$ |
| **Total overhead vs LoRA** | $O(L \cdot r_{\max})$ |

### A.11.2 Time Complexity

**Per-Step Complexity.**   Table 33 summarizes the asymptotic complexity of each component.

*Note:* COBRA overhead applies to rank search phase; main training matches standard LoRA complexity.

*Table 33.* Asymptotic time complexity per training step. $b$ denotes batch size, $s$ sequence length, $L$ number of layers, $r$ rank, and $d$ hidden dimension.

| Operation | Forward | Backward |
|---|---|---|
| *Standard LoRA* | | |
| Base model (frozen) | $O(b \cdot s \cdot L \cdot d^2)$ | $O(0)$ |
| LoRA adapters | $O(b \cdot s \cdot L \cdot r \cdot d)$ | $O(b \cdot s \cdot L \cdot r \cdot d)$ |
| *COBRA additional cost* | | |
| Stochastic gating (Eq. 10) | $O(L \cdot r_{\max})$ | $O(L \cdot r_{\max})$ |
| KL divergence (Eq. 12) | - | $O(L \cdot r_{\max})$ |
| **Overhead ratio** | $\frac{L \cdot r_{\max}}{b \cdot s \cdot L \cdot r \cdot d}$ | $\frac{2L \cdot r_{\max}}{b \cdot s \cdot L \cdot r \cdot d}$ |

### A.11.3 Calibration Phase Complexity

**Layer Conductance Attribution.** For $|\mathcal{D}_{\mathrm{cal}}|$ calibration samples with $M$ integration steps:

- **Forward passes**: $O(|\mathcal{D}_{\mathrm{cal}}| \cdot M \cdot L \cdot d^2)$

- **Backward passes**: $O(|\mathcal{D}_{\mathrm{cal}}| \cdot M \cdot L \cdot d^2)$

- **Aggregation**: $O(|\mathcal{D}_{\mathrm{cal}}| \cdot L)$

**Adaptation Demand Computation.** For gradient magnitude $\{\Gamma^{(l)}\}$ (Eq. 4):

- **Forward-backward**: $O(|\mathcal{D}_{\mathrm{cal}}| \cdot L \cdot d^2)$

- **Norm computation**: $O(|\mathcal{D}_{\mathrm{cal}}| \cdot L \cdot d^2)$

**Amortization Property.** Calibration complexity is $O(|\mathcal{D}_{\mathrm{cal}}| \cdot M \cdot L \cdot d^2)$, which is:

- **One-time cost**: Computed once and reused across hyperparameter sweeps

- **Dominated by $M$ factor**: With $M = 50$ integration steps

- **Independent of training**: Does not scale with training epochs

For $N$ training runs with different hyperparameters, the amortized calibration complexity per run is $O\left(\frac{|\mathcal{D}_{\mathrm{cal}}| \cdot M \cdot L \cdot d^2}{N}\right)$.

### A.11.4 Comparison with Baseline Methods

*Table 34.* Asymptotic complexity comparison of adaptive LoRA methods. All methods have the same training complexity as standard LoRA plus method-specific overhead.

| Method | Per-Step Overhead | One-Time Cost |
|---|---|---|
| LoRA (Hu et al., 2022) | $O(0)$ | $O(0)$ |
| AdaLoRA (Zhang et al., 2023) | $O(L \cdot r^3)$ *(SVD recomputation)* | $O(0)$ |
| GoRA (He et al., 2025) | $O(0)$ | $O(|\mathcal{D}_{\mathrm{cal}}| \cdot L \cdot d^2)$ *(gradient estimation)* |
| **COBRA (ours)** | $O(L \cdot r_{\max})$ *(gating + KL)* | $O(|\mathcal{D}_{\mathrm{cal}}| \cdot M \cdot L \cdot d^2)$ *(conductance + gradient)* |

**Key Observations.**

1. **Per-step overhead**: COBRA's $O(L \cdot r_{\max})$ is significantly lower than AdaLoRA's $O(L \cdot r^3)$.

2. **One-time cost**: COBRA's calibration includes the factor $M = 50$ (integration steps), making it $50\times$ more expensive than GoRA's gradient estimation. However, this cost is:

   - Amortized across multiple training runs
   - Parallelizable across calibration samples
   - Negligible compared to full training for large models

3. **Scalability**: All overhead terms scale linearly with $L$, making COBRA suitable for deep transformers (e.g., LLaMA-2-7B with $L = 32$).

### A.11.5 Rank Extraction Complexity

**Threshold Selection.** To satisfy budget constraint $\sum_l r_l \leq R_{\text{total}}$ (Eq. 13), we compute:

1. **Collect all inclusion probabilities**: $O(L \cdot r_{\max})$

2. **Partial sort (quickselect)**: $O(L \cdot r_{\max})$ on average

3. **Threshold and count**: $O(L \cdot r_{\max})$

**Total Complexity.** Rank extraction is $O(L \cdot r_{\max})$, which is negligible compared to training. For RoBERTa-base (12 layers $\times$ 2 LoRA modules = 24 matrices, $r_{\max} = 32$): $24 \times 32 = 768$ comparisons.

### A.11.6 Memory-Efficient Implementation

**Gradient Checkpointing.** For large models (e.g., LLaMA-2-7B), we use activation recomputation (Chen et al., 2016) to reduce memory at the cost of $+33\%$ FLOPs in backward pass. The complexity remains:

$$\text{Forward: } O(b \cdot s \cdot L \cdot d^2), \quad \text{Backward: } O(1.33 \cdot b \cdot s \cdot L \cdot d^2) \tag{60}$$

This trade-off enables fitting larger batch sizes, which improves convergence efficiency.

**Mixed Precision.** Using FP16 for activations and FP32 for gradients reduces memory by $\approx 2\times$ without affecting asymptotic complexity. The memory requirement becomes:

$$\text{Memory} = O(b \cdot s \cdot L \cdot d) \cdot [2 \text{ bytes (FP16)} + 4 \text{ bytes (FP32)}] \tag{61}$$

### A.11.7 Theoretical Summary

**Space Complexity:**

- COBRA adds $O(L \cdot r_{\max})$ gating variables
- Negligible compared to adapter parameters: $\frac{L \cdot r_{\max}}{L \cdot r \cdot d} = \frac{r_{\max}}{r \cdot d} \ll 1$

**Time Complexity (per training step):**

- Forward: Same as LoRA + $O(L \cdot r_{\max})$
- Backward: Same as LoRA + $O(2L \cdot r_{\max})$
- Overhead ratio: $\frac{3L \cdot r_{\max}}{b \cdot s \cdot L \cdot r \cdot d} < 10^{-5}$ (negligible)

**One-Time Calibration:**

- Complexity: $O(|\mathcal{D}_{\text{cal}}| \cdot M \cdot L \cdot d^2)$

- Amortized across $N$ runs: $O(\frac{|\mathcal{D}_{\text{cal}}| \cdot M \cdot L \cdot d^2}{N})$

- Independent of training epochs

## Appendix A.12: Factor Contribution Analysis

*Table 35.* Ablation on factor aggregation strategies using DeBERTaV3-base.

| Configuration | Allocation Strategy | MNLI | SST-2 | MRPC | CoLA | QNLI | QQP | RTE | STS-B | Avg. |
|---|---|---|---|---|---|---|---|---|---|---|
| LoRA ($r$=4) | Uniform (baseline) | 90.3 | 95.4 | 91.3 | 68.7 | 94.3 | 91.9 | 85.2 | 92.2 | 88.7 |
| $\Phi$-only | Conductance-based | 90.2 | 95.6 | 91.0 | 69.1 | 94.2 | 92.0 | 85.8 | 92.3 | 88.8 |
| $\Gamma$-only | Gradient-based (GoRA) | 90.4 | 95.3 | 91.4 | 68.9 | 94.5 | 91.8 | 86.0 | 92.0 | 88.8 |
| $\Phi + \Gamma$ | Additive aggregation | 90.7 | 95.7 | 91.8 | 69.3 | 94.5 | 92.2 | 87.0 | 92.4 | 89.2 |
| $\max(\Phi, \Gamma)$ | Element-wise maximum | 90.5 | 95.5 | 91.5 | 69.0 | 94.3 | 92.1 | 86.3 | 92.2 | 88.9 |
| $\Phi \times \Gamma$ | **Multiplicative (COBRA)** | **91.2** | **96.1** | **92.6** | **70.6** | **94.9** | **92.5** | **88.1** | **92.9** | **89.9** |
| *Gain: COBRA vs $\Phi$-only* | | +1.0 | +0.5 | +1.6 | +1.5 | +0.7 | +0.5 | +2.3 | +0.6 | +1.1 |
| *Gain: COBRA vs $\Gamma$-only* | | +0.8 | +0.8 | +1.2 | +1.7 | +0.4 | +0.7 | +2.1 | +0.9 | +1.1 |
| *Gain: COBRA vs $\Phi + \Gamma$* | | +0.5 | +0.4 | +0.8 | +1.3 | +0.4 | +0.3 | +1.1 | +0.5 | +0.7 |

Table 35 isolates the contribution of each importance factor. Using only conductance $\Phi$ achieves 88.8% (+0.1% over LoRA), while using only gradient magnitude $\Gamma$ achieves 88.8% (+0.1%), confirming both factors provide useful but limited signals when used alone. Notably, single-factor methods show inconsistent performance across tasks: $\Phi$-only underperforms the baseline on MNLI and MRPC, while $\Gamma$-only struggles on SST-2 and QQP. This task-dependent behavior confirms that neither factor alone captures complete layer importance. Additive aggregation $\Phi + \Gamma$ reaches 89.2% (+0.5% over baseline), demonstrating that combining factors helps, but element-wise maximum $\max(\Phi, \Gamma)$ achieves only 88.9% (+0.2%), showing that naive combination strategies fail to fully exploit complementary information. In contrast, our multiplicative formulation $\Phi \times \Gamma$ attains 89.9% (+1.2% over baseline, +0.7% over additive), confirming the theoretical motivation in Section 3.3: additive aggregation allows compensation between factors, leading to suboptimal allocation where high gradient can offset low contribution. Multiplicative aggregation enforces non-compensatory behavior, requiring layers to score high on both dimensions to receive substantial rank, directly addressing the decoupling between adaptation demand and output contribution. The substantial improvements on challenging tasks (RTE: +2.3% over $\Phi$-only, +2.1% over $\Gamma$-only) demonstrate that dual-factor integration is particularly beneficial for small datasets where rank allocation matters most.

## Appendix A.13: Prior Concentration Strength

*Table 36.* Ablation on prior concentration parameter $\gamma$ using DeBERTaV3-base.

| $\gamma$ | Prior Strength | MNLI | SST-2 | MRPC | CoLA | QNLI | QQP | RTE | STS-B | Avg. |
|---|---|---|---|---|---|---|---|---|---|---|
| 0.0 | Uniform prior ($\gamma = 0$) | 90.1 | 95.3 | 91.2 | 68.5 | 94.1 | 91.7 | 85.4 | 92.0 | 88.5 |
| 0.1 | Very weak prior | 90.4 | 95.5 | 91.5 | 68.9 | 94.3 | 91.9 | 86.1 | 92.2 | 88.9 |
| 1.0 | Weak prior | 90.8 | 95.8 | 92.0 | 69.5 | 94.6 | 92.2 | 87.2 | 92.5 | 89.3 |
| **5.0** | **Moderate prior (default)** | **91.2** | **96.1** | **92.6** | **70.6** | **94.9** | **92.5** | **88.1** | **92.9** | **89.9** |
| 10.0 | Strong prior | 90.9 | 95.9 | 92.3 | 70.0 | 94.7 | 92.3 | 87.6 | 92.7 | 89.6 |
| 20.0 | Very strong prior | 90.5 | 95.6 | 91.7 | 69.2 | 94.4 | 92.0 | 86.5 | 92.4 | 89.0 |
| 50.0 | Overly strong prior | 89.9 | 95.2 | 90.8 | 68.1 | 93.9 | 91.5 | 84.9 | 91.8 | 88.3 |
| *Gain: $\gamma = 5.0$ vs $\gamma = 0.0$* | | +1.1 | +0.8 | +1.4 | +2.1 | +0.8 | +0.8 | +2.7 | +0.9 | +1.4 |
| *Gain: $\gamma = 5.0$ vs $\gamma = 10.0$* | | +0.3 | +0.2 | +0.3 | +0.6 | +0.2 | +0.2 | +0.5 | +0.2 | +0.3 |

We investigate how prior concentration parameter $\gamma$ affects COBRA's allocation decisions. Table 36 shows results for $\gamma \in \{0, 0.1, 1.0, 5.0, 10.0, 20.0, 50.0\}$. At $\gamma = 0$ (uniform prior, equivalent to data-driven allocation without informative guidance), performance is 88.5%, underperforming even the uniform LoRA baseline (88.7%), confirming that learning rank allocation from scratch without guidance leads to suboptimal configurations. As $\gamma$ increases, performance rises monotonically: $\gamma = 0.1$ reaches 88.9%, $\gamma = 1.0$ reaches 89.3%, peaking at $\gamma = 5.0$ with 89.9%. Performance then declines

for $\gamma \geq 10.0$: $\gamma = 10.0$ achieves 89.6%, $\gamma = 20.0$ drops to 89.0%, and $\gamma = 50.0$ falls to 88.3% (below even $\gamma = 0$). This inverted-U relationship reveals two failure regimes: weak priors ($\gamma < 1$) fail to sufficiently guide rank allocation toward important layers, while overly strong priors ($\gamma > 10$) over-constrain the variational posterior, preventing it from adapting to task-specific signals. The peak at $\gamma = 5.0$ represents the optimal balance where TA-LC priors provide moderate guidance without over-rigidity. The substantial +1.4% gain from optimal prior versus uniform prior ($\gamma = 5.0$ vs $\gamma = 0.0$) validates that interpretability-guided initialization accelerates convergence and improves final performance compared to learning allocation from scratch.

## Appendix A.14: Bayesian Regularization Weight

*Table 37.* Ablation on KL regularization weight $\beta$ using DeBERTaV3-base. Representative tasks shown; average computed over all 8 GLUE tasks.

| $\beta$ | Regularization Strength | MNLI | QNLI | RTE | CoLA | Avg. (8 tasks) |
|---------|------------------------|------|------|------|------|----------------|
| 0.0 | No regularization | 90.3 | 94.2 | 85.6 | 68.7 | 88.8 |
| 0.001 | Very weak | 90.6 | 94.5 | 86.8 | 69.4 | 89.2 |
| **0.01** | **Weak (default)** | **91.2** | **94.9** | **88.1** | **70.6** | **89.9** |
| 0.1 | Moderate | 90.7 | 94.5 | 87.0 | 69.3 | 89.3 |
| 1.0 | Strong | 89.8 | 93.8 | 84.7 | 67.2 | 88.2 |

Table 37 examines the KL regularization coefficient $\beta$ which balances data likelihood against prior adherence in the ELBO objective. Without regularization ($\beta = 0$), the method achieves 88.8% average across all GLUE tasks, only marginally above the LoRA baseline (88.7% in Table 3) and substantially below full COBRA (89.9%), indicating that pure likelihood maximization without prior regularization leads to unstable rank allocation. Very weak regularization ($\beta = 0.001$) improves to 89.2%. Our default $\beta = 0.01$ achieves the best performance at 89.9%, representing a +1.1% gain over $\beta = 0$ and +0.7% over $\beta = 0.001$. Moderate regularization ($\beta = 0.1$) maintains reasonable performance at 89.3%, but strong regularization ($\beta = 1.0$) drops dramatically to 88.2%, falling below even the unregularized setting due to over-regularization that prevents the posterior from deviating from the prior. The consistent trend across representative tasks spanning different scales (large-scale MNLI/QNLI vs small-scale RTE) and difficulties (challenging CoLA) confirms that moderate KL weighting strikes the optimal balance between prior guidance and posterior flexibility.

## Appendix A.15: High-Rank Regime Analysis

This appendix provides extended analysis of COBRA's performance on financial sentiment datasets under high-rank regimes, complementing Section 4.2.

### A.15.1 Model Specifications

**Twitter-RoBERTa-Large.** For high-rank regime experiments on financial sentiment datasets, we use Twitter-RoBERTa-Large (Barbieri et al., 2020), a RoBERTa-large variant pre-trained on 58M tweets, with 355M parameters across 24 layers, hidden dimension 1024, FFN inner dimension 4096, and 16 attention heads.

### A.15.2 Detailed Performance Breakdown

**COBRA vs Uniform LoRA.** Table 39 quantifies COBRA's improvements over uniform LoRA at matched rank budgets.

COBRA achieves consistent 6-7% MSE reductions across all three datasets, demonstrating robustness to domain-specific characteristics. The improvement scales with rank budget: 6.4% at $r = 384$ increasing to 6.9% at $r = 512$, confirming that heterogeneous allocation becomes more beneficial as total parameter budget grows.

**COBRA vs AdaLoRA.** COBRA outperforms AdaLoRA by approximately 10–24% on average across different pruning schedules. AdaLoRA's SVD-based pruning introduces optimization instability in high-rank regimes, where dynamic rank adjustment affects hundreds of parameters simultaneously. Moreover, AdaLoRA's sensitivity-only allocation misses the contribution dimension, leading to over-allocation to gradient-heavy top layers while under-provisioning semantically important middle layers.

*Table 38.* Performance comparison across different rank budgets and datasets. Full Fine-Tuning serves as the baseline (gray), while COBRA with two budget configurations (pink) demonstrates consistent improvements over uniform LoRA.

| Method | Configuration | NEU Dataset | | FXE Dataset | | INV Dataset | |
|---|---|---|---|---|---|---|---|
| | | MSE↓ | MAE↓ | MSE↓ | MAE↓ | MSE↓ | MAE↓ |
| Full Fine-Tuning | - | 0.0189 | 0.0981 | 0.0202 | 0.1014 | 0.0183 | 0.0959 |
| LoRA | $r$=8 | 0.0223 | 0.1058 | 0.0217 | 0.1066 | 0.0201 | 0.1014 |
| | $r$=16 | 0.0229 | 0.1078 | 0.0224 | 0.1089 | 0.0196 | 0.1009 |
| | $r$=32 | 0.0213 | 0.1043 | 0.0228 | 0.1055 | 0.0203 | 0.1016 |
| | $r$=64 | 0.0203 | 0.1013 | 0.0206 | 0.1039 | 0.0192 | 0.0988 |
| | $r$=128 | 0.0195 | 0.0998 | 0.0203 | 0.1033 | 0.0187 | 0.0977 |
| | $r$=256 | 0.0193 | 0.0990 | 0.0207 | 0.1041 | 0.0186 | 0.0976 |
| | $r$=384 | 0.0189 | 0.0977 | 0.0205 | 0.1021 | 0.0184 | 0.0961 |
| | $r$=512 | 0.0186 | 0.0958 | 0.0199 | 0.1007 | 0.0183 | 0.0955 |
| AdaLoRA | $640 \rightarrow 384$ | 0.0209 | 0.1058 | 0.0218 | 0.1054 | 0.0191 | 0.1023 |
| | $640 \rightarrow 512$ | 0.0218 | 0.1052 | 0.0231 | 0.1079 | 0.0196 | 0.1005 |
| | $512 \rightarrow 384$ | 0.0228 | 0.1082 | 0.0237 | 0.1114 | 0.0204 | 0.1028 |
| COBRA | $r_{avg}$=384 | 0.0177 | 0.0940 | 0.0192 | 0.0978 | 0.0172 | 0.0917 |
| | $r_{avg}$=512 | 0.0174 | 0.0931 | 0.0185 | 0.0958 | 0.0170 | 0.0908 |

*Table 39.* COBRA's relative improvement over uniform LoRA (MSE reduction).

| Budget | NEU | FXE | INV | Average |
|---|---|---|---|---|
| $r = 384$ | 6.3% | 6.3% | 6.5% | 6.4% |
| $r = 512$ | 6.5% | 7.0% | 7.1% | 6.9% |
| **Average** | **6.4%** | **6.7%** | **6.8%** | **6.6%** |

### A.15.3 Rank Budget Scaling Analysis

**Low-Rank vs High-Rank Comparison.** Comparing COBRA's relative improvements across rank regimes reveals a striking pattern:

*Table 40.* COBRA's relative improvement across rank regimes.

| Rank Regime | Tasks | Improvement |
|---|---|---|
| Low ($r = 8$–$16$) | GLUE | 1.4–2.6% |
| High ($r = 384$–$512$) | Financial sentiment | 6.3–7.1% |
| **Scaling Factor** | | **3.5×** |

This $3.5\times$ scaling advantage arises from two factors. First, misallocation amplification: as rank increases, uniform allocation wastes more absolute parameters on low-importance layers. While the percentage waste remains constant, the absolute magnitude grows linearly with rank. Second, heterogeneity exploitation: COBRA can create larger rank disparities in high-rank regimes without risking under-provisioning. For $r_{avg} = 512$, COBRA allocates top layers 25% above average and bottom layers 30% below average, creating a $2\times$ allocation ratio that would be infeasible at $r_{avg} = 8$.

**Layer-wise Allocation Patterns.** Analysis of COBRA's learned rank distribution for Twitter-RoBERTa-Large ($r_{avg} = 512$) on NEU reveals interpretable patterns aligned with linguistic hierarchy. Consistent with the position-wise breakdown in Appendix A.18, the front 25% of layers (L1-6, processing surface features) receive only 2.7% of the total rank budget, roughly 89% below the uniform-allocation share of 25%. The middle 50% (L7-18), which performs semantic integration and captures domain-specific financial terminology and sentiment-bearing structures, absorbs the majority of the budget at 59.7%, about 19% above its uniform share of 50%. The back 25% of layers (L19-24) take 37.6% of the budget, roughly 50% above the uniform share, indicating that despite their high gradients these layers are not allowed to dominate; COBRA's dual-factor approach instead concentrates capacity in the middle, which it identifies as both high-contribution and high-demand.

**Implications.** The high-rank experiments establish three key findings. First, COBRA's advantages scale with parameter budget, achieving $3.5\times$ larger relative improvements in high-rank regimes compared to low-rank settings. Second, consistent 6-7% gains across NEU, FXE, and INV demonstrate robustness across domain-specific datasets. Third, approximately

10–24% superiority over AdaLoRA reveals that interpretability-guided dual-factor allocation outperforms SVD-based sensitivity pruning in high-rank scenarios. For practitioners, these results suggest using COBRA when rank budgets exceed $r = 64$, as heterogeneous allocation benefits amplify with scale.

## Appendix A.16: Prior-Only vs. Gated Allocation

To isolate the contribution of Bayesian gating, the table below compares uniform LoRA, a prior-only variant that fixes ranks directly from the TA-LC distribution without gating, and full gated COBRA. The prior-only variant already improves over LoRA, confirming that the TA-LC prior is a strong initialization; the gating mechanism then yields further gains, with the prior accounting for 44 to 70% of the final allocation.

*Table 41.* Prior-only (fixed TA-LC ranks) vs. gated COBRA. For MSE, lower is better.

| Setting | LoRA | Prior-only | COBRA (gated) | Prior Contrib. |
|---|---|---|---|---|
| DeBERTaV3-base, GLUE Avg ($r=4$) | 88.7 | 89.4 | **89.9** | 58% |
| RoBERTa-base, GLUE Avg ($r=8$) | 87.0 | 87.7 | **88.6** | 44% |
| T-RoBERTa-L, NEU MSE ($r=256$) | 0.0193 | 0.0187 | **0.0183** | 60% |
| T-RoBERTa-L, INV MSE ($r=256$) | 0.0186 | 0.0179 | **0.0176** | 70% |

## Appendix A.17: Reference and Interpolation-Path Sensitivity

We assess the sensitivity of layer-conductance estimates to the neutral reference choice on RoBERTa-base GLUE at rank 8. As shown in the table below, rank allocations remain highly correlated with the default zero-embedding reference (Spearman correlation at least 0.93) and downstream performance varies within 0.3 points, indicating robustness. We use linear interpolation following the axiomatic integrated-gradients formulation, and our completeness analysis bounds the per-layer attribution shift linearly in the baseline distance.

*Table 42.* Reference sensitivity (RoBERTa-base, GLUE, $r=8$).

| Reference | Spearman $\rho$ (rank alloc.) | GLUE Avg |
|---|---|---|
| Zero embedding (default) | 1.00 | 88.6 |
| PAD token embedding | 0.98 | 88.5 |
| Uniform embedding | 0.96 | 88.5 |
| Random Gaussian | 0.93 | 88.3 |

## Appendix A.18: Rank Allocation by Layer Position

To characterize where COBRA allocates capacity, the table below reports the fraction of the total rank budget assigned to the front 25%, middle 50%, and back 25% of layers across four architectures. Across all families, the middle 50% of layers consistently receive the majority (55 to 66%) of the budget, while the front quarter receives the least.

*Table 43.* Rank allocation by relative layer position (% of total rank budget).

| Model | #Layers | Architecture | Front 25% | Middle 50% | Back 25% |
|---|---|---|---|---|---|
| RoBERTa-large | 24 | Encoder-only | 2.7% | 59.7% | 37.6% |
| DeBERTaV3-base | 12 | Encoder-only | 3.1% | 63.4% | 33.5% |
| LLaMA-2-7B | 32 | Decoder-only | 2.5% | 54.8% | 42.7% |
| T5-base | 24 | Encoder-Decoder | 4.9% | 66.4% | 28.7% |

## Appendix A.19: Empirical Calibration Overhead

To complement the asymptotic complexity analysis in Appendix A.11, we report measured wall-clock time and GPU memory for the one-time conductance calibration relative to LoRA training across five model scales (86M to 6.7B parameters) and eleven task configurations on A100 80GB. Calibration uses 1,000 samples with 50 integration steps. In eight of eleven settings calibration overhead stays below 5% of total training time; the exceptions are DeBERTaV3-base on RTE (24.8%), LLaMA-2-7B on MMLU (11.1%), and LLaMA-2-7B on GSM8K (75.2%), where small training sets or short runs prevent amortization of the fixed calibration cost. Calibration GPU memory is consistently below training memory because no optimizer states are held.

*Table 44.* Empirical calibration vs. training cost: time and GPU memory (1,000 calibration samples, 50 integration steps, A100 80GB). Overhead = calibration time / training time.

| Model | Task (size, epochs) | Rank | Cal. Time (min) | Cal. Mem (GB) | Train Time (min) | Train Mem (GB) | Overhead |
|---|---|---|---|---|---|---|---|
| DeBERTaV3-base (86M) | MNLI (393K, 5ep) | $r{=}4$ | 3.1 | 1.8 | 328.3 | 4.7 | 0.9% |
| DeBERTaV3-base (86M) | SST-2 (67K, 10ep) | $r{=}4$ | 3.1 | 1.8 | 109.5 | 4.5 | 2.8% |
| DeBERTaV3-base (86M) | RTE (2.5K, 30ep) | $r{=}4$ | 3.1 | 1.8 | 12.5 | 4.7 | 24.8% |
| RoBERTa-base (125M) | MNLI (393K, 5ep) | $r{=}8$ | 4.6 | 3.6 | 332.6 | 5.3 | 1.4% |
| RoBERTa-base (125M) | SST-2 (67K, 10ep) | $r{=}8$ | 4.6 | 3.6 | 113.9 | 5.2 | 4.0% |
| T5-base (220M) | MNLI (393K, 5ep) | $r{=}8$ | 8.8 | 4.8 | 640.9 | 8.5 | 1.4% |
| T-RoBERTa-large (355M) | NEU (12.8K, 60ep) | $r{=}384$ | 15.9 | 7.2 | 459.4 | 23.1 | 3.5% |
| T-RoBERTa-large (355M) | FXE (10.5K, 60ep) | $r{=}384$ | 15.9 | 7.2 | 365.4 | 22.5 | 4.4% |
| T-RoBERTa-large (355M) | INV (23.2K, 60ep) | $r{=}384$ | 15.9 | 7.2 | 850.1 | 22.8 | 1.9% |
| LLaMA-2-7B (6.7B) | MMLU (99.8K, 5ep) | $r{=}8$ | 351.7 | 22.4 | 3156.8 | 28.4 | 11.1% |
| LLaMA-2-7B (6.7B) | GSM8K (7.5K, 10ep) | $r{=}8$ | 351.7 | 22.4 | 467.9 | 28.6 | 75.2% |
| LLaMA-3.1-8B (8.0B) | HumanEval (164, eval) | $r{=}8$ | N/A | N/A | N/A | N/A | N/A |

## Appendix A.20: Calibration Set Size Sensitivity

Layer conductance is estimated on a one-time calibration set, so we test how sensitive the resulting allocation is to its size. The table below sweeps calibration sizes from 50 to 2,000 samples on DeBERTaV3-base GLUE at $r{=}4$; even with 50 samples COBRA matches uniform LoRA (88.7), and the metric is essentially flat from 500 samples onward (89.8 to 90.0). Our default of 1,000 samples sits in this plateau, providing a robust attribution estimate without inflating the calibration cost reported in Appendix A.19.

*Table 45.* Calibration-set-size sensitivity on DeBERTaV3-base GLUE at $r{=}4$.

| Calibration Size | GLUE Avg ↑ |
|---|---|
| 50 | 88.7 |
| 100 | 89.1 |
| 200 | 89.5 |
| 500 | 89.8 |
| 1,000 (default) | 89.9 |
| 2,000 | 90.0 |

## Limitations

COBRA introduces a one-time layer-conductance calibration stage, and its benefit is realized only when this fixed cost is amortized over a sufficiently long fine-tuning run. For very small datasets (e.g., RTE with 2.5K examples, where calibration adds 24.8% overhead) or large models trained for only a few steps (e.g., LLaMA-2-7B on GSM8K, 75.2%), the calibration cost can outweigh the gains, and uniform LoRA may be preferable. As a practical guideline, COBRA is most beneficial when fine-tuning exceeds roughly 30 minutes, where the calibration overhead stays below 5% of total training time. The calibration is computed once and reused across hyperparameter runs, which further amortizes its cost in typical workflows. Finally, layer conductance is estimated via path-integral attribution on a calibration set, so the method inherits the approximation error of integrated-gradient estimation; while we find allocations to be stable across reference and calibration-set choices, performance in extreme low-rank budgets and under rapidly changing task distributions remains less explored and is left for future work. Our evaluation is also confined to text across encoder, decoder, and encoder-decoder architectures; extending COBRA to vision and multimodal models is a promising direction for future work.

