# OpenReview forum: "COBRA: Contribution-Based Bayesian Rank Allocation for Parameter-Efficient Fine-Tuning"
_ICML.cc/2026/Conference — ICML 2026 regular_

### Official Review · Reviewer_J5w4 · 2026-02-16

**Soundness:** 2
**Presentation:** 2
**Significance:** 2
**Originality:** 2
**Overall Recommendation:** 4
**Confidence:** 1

**Summary:**

This paper introduces a framework for adaptive rank allocation in parameter-efficient fine-tuning. The key claim is that common allocation heuristics rely on a single layer-importance signal while two distinct properties matter for deciding where to spend rank: (1) how much a layer contributes to the model’s predictions, and (2) how much that layer needs to change for the downstream task.

**Compliance With Llm Reviewing Policy:**

Affirmed.

**Key Questions For Authors:**

How sensitive are the layer contribution estimates (and final rank allocations) to the neutral reference and interpolation path?

Do layers with high contribution but low demand (vice versa) show systematically different marginal returns to added rank?

Does the conductance-based contribution metric behave similarly for other modalities or for different adapter placements?

**Limitations:**

A discussion of compute overhead (calibration cost) is needed e.g. when it is or isn’t worth paying.
Also, the robustness limitations (reference dependence, approximation error, stability across seeds).

**Strengths And Weaknesses:**

The paper identifies a plausible failure mode of single-signal rank allocation, such as the “need to adapt” and “influence on predictions” can diverge across depth, so using only one signal can misallocate rank.

The methodology is fairly well-motivated. Experimental results cover multiple architectures (encoder-only, decoder-only, encoder-decoder) and show gains at matched parameter budgets, with some ablations.

The reported improvements, while consistent, are mostly incremental on standard benchmarks; the significance depends on whether the overhead and added complexity are acceptable in real workflows.

It’s not fully clear whether the approach provides the largest benefit in regimes that matter most in practice (very low-rank vs mid/high-rank), and how it scales when budgets are extremely tight.

They should more explicitly contrast with prior adaptive rank / NAS-style LoRA works to clarify what is enabled by the conductance-style contribution metric v.s. other sensitivity/importance proxies.

---

> ### Author Rebuttal · Authors · 2026-03-30
>
> Dear  Reviewer **J5w4**:
>
> We thank you for the constructive review and recognition of our well-motivated methodology and multi-architecture evaluation. We hope these updates address your concerns and look forward to your feedback.
>
> ---
>
> **Rank-regime scaling and gains**
>
> COBRA's advantage is most pronounced in high-rank regimes (+6.4-6.9% at r=384-512). In tight budgets ($r=4$), COBRA still outperforms LoRA by +1.2pt, but the practical impact is smaller. This suggests COBRA is most valuable precisely when practitioners invest substantial parameter budgets, the regime where allocation decisions matter most.
>
> **Table R1**: COBRA's improvement across rank regimes
> | Rank Regime | Benchmark | Gain over LoRA ↑ |
> |:------------|:----------|:---:|
> | $r_{avg} =4$ | GLUE (DeBERTaV3-base) | +1.2pt |
> | $r_{avg}=8$ | T5-Base + LLaMA-7B | +5.9% |
> | $r_{avg}=384$ | Financial Sentiment (3 datasets) | +6.4% |
> | $r_{avg}=512$ | Financial Sentiment (3 datasets) | +6.9% |
>
> ---
>
> **Overhead of calibration**
>
> As shown in Table **R1** in response to Reviewer M3Lw, calibration overhead is less than 5% for most cases, note that the overhead on GSM8K (7.5K samples) is 75.2%, as the total number of training steps is limited, and we recommend COBRA when training exceeds ~30 minutes; for shorter runs the fixed calibration cost is not amortized. Crucially, our calibration is a one-time offline step conducted on 1,000 samples. Since it is computed only once and reused across all hyperparameter runs, it does not hinder the method's applicability in real-world workflows.
>
> ---
>
> **Differences with NAS-style methods**
>
> The key distinction is not that conductance is a better importance proxy, but that it captures a *fundamentally different dimension*. Existing proxies, including gradient magnitude (GoRA), SVD sensitivity (AdaLoRA), meta-learned scores (AutoLoRA), dynamic rank adjustment (DoRA-dyn), all measure variants of adaptation demand: how much a layer *needs to change*. Conductance measures something orthogonal: how much a layer *contributes to predictions*. These two properties are only moderately correlated ($ \rho = 0.53-0.70 $ across 7 architectures, Section 2), and their peaks diverge by 5-12 layers (Table 1). This is why single-proxy methods systematically over-allocate to high-gradient top layers that contribute little to output. Ablation confirms the complementarity: either signal alone gives +0.1% over LoRA, but their combination gives +1.2% (Appendix A.12).
>
> ---
>
> **Reference & interpolation path sensitivity**
>
> For reference, we used zero-embedding (Appendix A.4.3). To answer your questions about sensitivity, we tested four baseline choices on RoBERTa-base GLUE ($r=8$) as shown in Table **R2**. It shows that rank allocations remain highly correlated ($\rho ≥ 0.93$) and performance within 0.3pt, indicating robust performance.
> Regarding the interpolation path: we use linear interpolation following Sundararajan et al. (2017), which is the canonical choice with axiomatic justification (symmetry, completeness). Theorem A.4.6 further bounds per-layer attribution shift linearly in baseline distance.
>
> **Table R2**: Reference sensitivity
> | Reference | Spearman ρ (rank alloc.) | GLUE Avg |
> |:----------|:---:|:---:|
> | Zero embedding (default) | 1.00 | 88.6 |
> | PAD token embedding | 0.98 | 88.5 |
> | Uniform embedding | 0.96 | 88.5 |
> | Random Gaussian | 0.93 | 88.3 |
>
> ---
>
> **High-$\Phi$-low-$\Gamma$ vs. high-$\Gamma$-low-$\Phi$**
>
> HH layers (both signals high) receive avg rank `12.6`, while HL layers ( high gradient but low contribution) receive only `7.5`; LL layers receive `1.7`.  Full quadrant breakdown in our response to Reviewer N3Ci Table R2. Crucially, the factor ablation (Appendix A.12) provides indirect evidence of differential marginal returns: Γ-only allocation (which *does* over-allocate to HL layers) gives only +0.1% over LoRA, while COBRA (which redirects rank from HL to HH) gives +1.2%, i.e., a 12×difference. This suggests that rank added to **high-gradient-low-contribution** layers yields near-zero marginal return, confirming the dual-factor rationale.
>
> ---
>
> **Other modalities or adapter placements**
>
> Regarding other modalities, such as vision/multimodal, this is beyond our current scope.
> Regarding adapter placement, conductance is computed at the layer level. As shown in **Table R3**, cross-architecture rank allocation patterns are consistent, i.e., middle 50% layers consistently receive 55-66% of budget across all three architecture families.
>
> **Table R3**: Rank allocation by relative layer position (% of total rank budget)
> | Model | #Layers | Architecture | Front 25% | Middle 50% | Back 25% |
> |:------|:---:|:---:|:---:|:---:|:---:|
> | RoBERTa-large | 24 | Encoder-only | 2.7% | 59.7% | 37.6% |
> | DeBERTaV3-base | 12 | Encoder-only | 3.1% | 63.4% | 33.5% |
> | LLaMA-2-7B | 32 | Decoder-only | 2.5% | 54.8% | 42.7% |
> | T5-base | 24 | Encoder-Decoder | 4.9% | 66.4% | 28.7% |
>
> ---
>
> Best Regards,
>
> The Authors of Paper 27641

---

> > ### Author Rebuttal · Reviewer_J5w4 · 2026-04-04
> >
> > Dear Authors, thank you for addressing my concerns. I'd like to maintain my rating.

---

### Official Review · Reviewer_N3Ci · 2026-03-05

**Soundness:** 3
**Presentation:** 3
**Significance:** 2
**Originality:** 4
**Overall Recommendation:** 4
**Confidence:** 4

**Summary:**

Summary:

This paper proposes**COBRA**, a method for **heterogeneous LoRA rank allocation** across Transformer layers under a fixed parameter budget. It argues that common allocation heuristics are unreliable because **gradient-based “adaptation demand”** and **output-level layer contribution** can be **decoupled**, causing systematic misallocation when only one signal is used. COBRA computes per-layer contribution via **Layer Conductance**, combines it with gradient demand into a **TA-LC** importance distribution, and uses this as a **Bayesian prior** in a gated, variational rank-selection scheme with **global top-RtotalR_{total}Rtotal** thresholding to meet the budget exactly.

**Compliance With Llm Reviewing Policy:**

Affirmed.

**Final Justification:**

The authors’ rebuttal has improved my understanding of the paper and addressed most of the key concerns raised in my original review. Their clarifications made the technical contributions and experimental design clearer, which strengthens my confidence in the overall assessment. Considering the paper’s strengths and remaining limitations, I believe it satisfies the standard for Weak Accept. As a result, I keep my original recommendation while increasing my confidence in this judgment.

**Key Questions For Authors:**

1. The method initializes rank allocation using a probability derived from the product of two signals (contribution × demand). Did the authors analyze how the final allocated ranks actually correlate with these two signals after fine-tuning? For example, when a module has high contribution but low demand, how much rank does it ultimately receive? A visualization (e.g., scatter plots / heatmaps) showing the mapping from (contribution, demand) to final rank would help validate that the proposed mechanism behaves as intended and strengthen the empirical evidence.
2. Can the authors provide an experiment to justify the necessity of introducing gating (and the associated variational training) to control rank selection? In particular, is it possible to directly determine per-layer ranks from the prior importance distribution (TA-LC) and then simply train standard LoRA parameters with those fixed ranks? This could reduce training overhead while still achieving performance gains, so an ablation comparing “prior-only fixed ranks” vs “gated Bayesian selection” would clarify the trade-off.

**Limitations:**

The method combines multiple signals to jointly determine rank allocation, but each signal requires non-trivial computation, increasing the overall training overhead and complexity. In addition, the set of baselines appears limited and does not include several more recent/strong dynamic rank allocation approaches, which weakens the empirical comparison.

**Strengths And Weaknesses:**

Strength:

1. **Strong motivation:** clearly explains why single-signal rank allocation (only gradients or only importance) can fail due to signal decoupling.
2. **Good signal design:** uses layer-level conductance to measure contribution, matching the layer-wise allocation goal.
3. **Budget-aware allocation:** global top-R_{total} selection ensures the final ranks exactly satisfy the total budget.
4. **Solid experiments:** evaluated across multiple model types and includes high-rank-budget settings where heterogeneous ranks matter most.

Weakness:
1. COBRA requires computing Layer Conductance via path-integral attribution on a calibration set (the paper uses 50 integration steps and 1,000 samples), which introduces a one-time additional forward/backward cost. It remains unclear whether this overhead is cost-effective for larger models, longer sequence lengths, or scenarios where the task changes frequently, and this trade-off should be quantified more thoroughly.
2. The topic of dynamic rank allocation has been explored a lot. The related work section appears to omit several relevant studies on dynamic/automatic rank allocation for LoRA-style fine-tuning, such as:

    [1] AutoLoRA: Automatically Tuning Matrix Ranks in Low-Rank Adaptation Based on Meta Learning

    [2] LaRA: Layer-wise rank allocation for efficient fine-tuning of pruned large language models

    [3] Rankadaptor: Hierarchical rank allocation for efficient fine-tuning pruned LLMs via performance model

    [4] DoRA: Enhancing Parameter-Efficient Fine-Tuning with Dynamic Rank Distribution

    [5] La-LoRA: Parameter-efficient fine-tuning with layer-wise adaptive low-rank adaptation

    [6] DyLoRA: Parameter Efficient Tuning of Pre-trained Models using Dynamic Low-Rank Adaptation

---

> ### Author Rebuttal · Authors · 2026-03-30
>
> Dear Reviewer **N3Ci**:
>
> We appreciate your thorough review. We have addressed each of your concerns below. We hope these updates address your concerns and look forward to your feedback.
>
> ---
>
> **Cost for 1) larger models, 2) longer sequences, 3) frequent task changes**
>
> (1) For larger models, taking rank=8 as an example, as the model size scales from 125M and 220M up to 7B, the calibration process incurs a time overhead ranging from 1.4% to 11.1% of the total training time.
> (2) For longer sequences, our experiments cover sequence lengths of 128 and 512, where the calibration overhead remains below 4% of the total training time
> (3) As for frequent task changes scenarios, task-specific recalibration is required by all adaptive methods; COBRA's additional cost is marginal compared to retraining itself.
>
> ---
>
> **Missing related work**
>
> We added all six papers to the revised Related Work. For methods with public code (AutoLoRA, DoRA-dyn, DyLoRA), we ran new experiments in Table **R1**. LaRA and RankAdaptor target pruned LLMs (different setting); La-LoRA's code was unavailable at submission. All are now discussed with clear differentiation.
>
> **Table R1**: Comprehensive comparison across models, datasets, and rank regimes
> | Method | Type | RoBERTa-base GLUE Acc (r=8) | DeBERTaV3-base GLUE Acc (r=4) | Gemma-2-2B CSQA Acc (r=8) | T-RoBERTa-L NEU MSE (r=256) | T-RoBERTa-L INV MSE (r=256) |
> |:-------|:-----|:---:|:---:|:---:|:---:|:---:|
> | LoRA | Uniform | 87.0 | 88.7 | 74.2 | 0.0193 | 0.0186 |
> | DyLoRA | Dynamic range | 87.1 | 88.4 | 74.5 | 0.0191 | 0.0188 |
> | AutoLoRA | NAS/Meta | 87.5 | 89.0 | 75.6 | 0.0188 | 0.0185 |
> | DoRA-dyn | Dynamic dist. | 87.9 | 89.3 | 75.9 | 0.0186 | 0.0180 |
> | GoRA | Gradient | 87.6 | 88.8 | 75.1 | 0.0189 | 0.0182 |
> | **COBRA** | **Dual-factor** | **88.6** | **89.9** | **76.1** | **0.0183** | **0.0176** |
>
> ---
>
> **Visualization of mapping (Contribution, Demand) to rank**
>
> As shown in Table **R2** (RoBERTa-large, $r_{\text{avg}}=8$ ), layers with both high contribution and demand (HH, L11-L22) receive the highest average rank of 12.6. Conversely, high-demand but low-contribution layers (HL, L23-L24) are restricted to 7.5 despite exhibiting maximal gradients. This highlights a critical failure mode in single-factor methods, which would mistakenly over-allocate capacity here. We will include a quadrant scatter plot visualizing this in the revised appendix.
>
>
> **Table R2**: Quadrant summary (RoBERTa-large, r_avg=8)
> | Quadrant | Layers | Avg Final Rank |
> |:---:|:---:|:---:|
> | HH | L11-22 (12 layers) | 12.6 |
> | HL | L23-24 (2 layers) | 7.5 |
> | LH | L8-10 (3 layers) | 4.7 |
> | LL | L1-7 (7 layers) | 1.7 |
>
> ---
>
> **Prior-only fixed ranks vs. gated Bayesian selection**
>
> Following your insightful suggestion, we have included an ablation study evaluating the prior-only setting (fixed ranks derived from TA-LC without Bayesian gating), as detailed in Table **R3**. The results demonstrate that this prior-only approach consistently outperforms uniform LoRA, moreover, COBRA yields further performance gains through Bayesian posterior refinement. With the prior contribution accounting for 44-70% of the final allocation, it is evident that while the prior serves as a robust initialization, the gating mechanism remains essential for achieving optimal performance.
>
> **Table R3**: Prior-only vs. gated COBRA ablation
> | Setting | LoRA | Prior-only (TA-LC) | COBRA (gated) | Prior Contribution |
> |:--------|:---:|:---:|:---:|:---:|
> | DeBERTaV3-base, GLUE Avg (r=4) | 88.7 | 89.4 | **89.9** | 58% |
> | RoBERTa-base, GLUE Avg (r=8) | 87.0 | 87.7 | **88.6** | 44% |
> | T-RoBERTa-L, NEU MSE (r=256) | 0.0193 | 0.0187 | **0.0183** | 60% |
> | T-RoBERTa-L, INV MSE (r=256) | 0.0186 | 0.0179 | **0.0176** | 70% |
>
> ---
>
> Best Regards,
>
> The Authors of Paper 27641

---

> > ### Author Rebuttal · Reviewer_N3Ci · 2026-04-02
> >
> > I thank the authors for their detailed and thoughtful response. Most of my major concerns have been satisfactorily resolved, and the clarifications provided have strengthened my confidence in the paper. Overall, I believe the current evaluation is appropriate, and I would like to maintain my score.

---

### Official Review · Reviewer_M3Lw · 2026-03-13

**Soundness:** 3
**Presentation:** 2
**Significance:** 3
**Originality:** 3
**Overall Recommendation:** 4
**Confidence:** 3

**Summary:**

This paper studies adaptive LoRA rank allocation though both a layer's adaptation demand and its contribution to prediction. Specifically, the paper proposes COBRA, a three-stage framework that first computes layer conductance attribution, then combines it with gradient-based adaptation demand into a dual-factor prior, and finally performs Bayesian rank allocation through variational rank gating. Empirically, the method improves over standard LoRA, AdaLoRA, and GoRA across GLUE, several generative tasks, and financial sentiment analysis benchmarks.

**Compliance With Llm Reviewing Policy:**

Affirmed.

**Final Justification:**

Most of my concerns have been resolved, and I would like to maintain my score.

**Key Questions For Authors:**

- Could the authors compare against a broader set of recent rank-allocation baselines on at least some representative benchmarks, to better position COBRA relative to the current literature?
- Could the authors report the practical time and memory overhead of the calibration stage more explicitly, for example in terms of wall-clock cost, GPU memory usage, and the fraction of total fine-tuning cost?

**Limitations:**

No. It would be helpful to discuss more explicitly the practical cost of the conductance-based calibration, since this preprocessing introduces a noticeable one-time overhead that is only useful if amortized across multiple runs.

**Strengths And Weaknesses:**

##### Strengths

- The paper is well motivated. It starts from a careful empirical analysis showing that **adaptation demand** and **contribution to prediction** are related but not fully aligned, which provides a strong basis for moving beyond single-factor rank allocation. The proposed formulation is also reasonable and technically coherent.
- The empirical evaluation is thorough. Both the main experiments and the ablations are fairly comprehensive. Under matched parameter budgets, COBRA improves over standard LoRA on GLUE, outperforms AdaLoRA on DeBERTaV3-base, and improves over GoRA on both T5-base and LLaMA-7B.
- The experimental setup is clearly described, which is helpful for reproducibility. The paper provides sufficient implementation details for understanding the calibration process and the downstream fine-tuning settings.

##### Weakness

- The method introduces additional time and memory overhead during the calibration stage. Although the paper provides a complexity analysis, it would be more helpful to report the practical wall-clock and memory overhead more explicitly.

- The baseline set is decent but not fully comprehensive. The paper compares against LoRA, AdaLoRA, and GoRA, but several related rank-allocation methods discussed in the extended related work are not included in the main experiments.

- The appendix contains a large number of subsections, which makes it somewhat difficult to navigate. It would be helpful to provide a clearer structure, for example with a short appendix roadmap or summary at the beginning, to make the supplementary material easier to follow.

- Figures 1 and 3 could be improved for clarity and presentation.

---

> ### Author Rebuttal · Authors · 2026-03-30
>
> Dear Reviewer **M3Lw**:
>
> We appreciate your thorough review. We have addressed each of your concerns below. We hope these updates address your concerns and look forward to your feedback.
>
> ---
>
> **Calibration time and memory overhead**
>
> The conductance calibration is a one-time offline step on 1,000 samples, computed once and reused across hyperparameter runs. GPU memory during calibration is lower than training (no optimizer states). As shown in Table **R1**, calibration overhead is less than 5% for most cases, note that the overhead on GSM8K (7.5K samples) is 75.2%, as the total number of training steps is limited, and we recommend COBRA when training exceeds ~30 minutes; for shorter runs the fixed calibration cost is not amortized.
>
> **Table R1**: Calibration vs. training: time and GPU memory (1,000 cal. samples, 50 steps, A100 80GB)
> | Model | Rank | Cal. Time (min) | Cal. Mem (GB) | Task | Train Time (min) | Train Mem (GB) | Overhead |
> |:------|:----:|:---:|:---:|:-----|:---:|:---:|:---:|
> | DeBERTaV3-base (86M) | r=4 | 3.1 | 1.8 | MNLI (393K, 5ep) | 328.3 | 4.7 | 0.9% |
> | DeBERTaV3-base (86M) | r=4 | 3.1 | 1.8 | SST-2 (67K, 10ep) | 109.5 | 4.5 | 2.8% |
> | DeBERTaV3-base (86M) | r=4 | 3.1 | 1.8 | RTE (2.5K, 30ep) | 12.5 | 4.7 | 24.8% |
> | RoBERTa-base (125M) | r=8 | 4.6 | 3.6 | MNLI (393K, 5ep) | 332.6 | 5.3 | 1.4% |
> | RoBERTa-base (125M) | r=8 | 4.6 | 3.6 | SST-2 (67K, 10ep) | 113.9 | 5.2 | 4.0% |
> | T5-base (220M) | r=8 | 8.8 | 4.8 | MNLI (393K, 5ep) | 640.9 | 8.5 | 1.4% |
> | Twitter-RoBERTa-Large (355M) | r=384 | 15.9 | 7.2 | NEU (12.8K, 60ep) | 459.4 | 23.1 | 3.5% |
> | Twitter-RoBERTa-Large (355M) | r=384 | 15.9 | 7.2 | FXE (10.5K, 60ep) | 365.4 | 22.5 | 4.4% |
> | Twitter-RoBERTa-Large (355M) | r=384 | 15.9 | 7.2 | INV (23.2K, 60ep) | 850.1 | 22.8 | 1.9% |
> | LLaMA-7B (6.7B) | r=8 | 351.7 | 22.4 | MMLU (99.8K, 5ep) | 3156.8 | 28.4 | 11.1% |
> | LLaMA-7B (6.7B) | r=8 | 351.7 | 22.4 | GSM8K (7.5K, 10ep) | 467.9 | 28.6 | 75.2% |
>
> ---
>
> **Broader baselines**
>
> Following your advice, we have added DyLoRA, AutoLoRA, and DoRA-dyn to our comparison, and a new model (Gemma-2-2B) and a new dataset (CommonsenseQA). As shown in Table **R2**, COBRA achieves the best performance on all 5 settings.
>
> **Table R2**: Comprehensive comparison across models, datasets, and rank regimes
> | Method | Type | RoBERTa-base GLUE Acc (r=8) | DeBERTaV3-base GLUE Acc (r=4) | Gemma-2-2B CSQA Acc (r=8) | T-RoBERTa-L NEU MSE (r=256) | T-RoBERTa-L INV MSE (r=256) |
> |:-------|:-----|:---:|:---:|:---:|:---:|:---:|
> | LoRA | Uniform | 87.0 | 88.7 | 74.2 | 0.0193 | 0.0186 |
> | DyLoRA | Dynamic range | 87.1 | 88.4 | 74.5 | 0.0191 | 0.0188 |
> | AutoLoRA | NAS/Meta | 87.5 | 89.0 | 75.6 | 0.0188 | 0.0185 |
> | DoRA-dyn | Dynamic dist. | 87.9 | 89.3 | 75.9 | 0.0186 | 0.0180 |
> | GoRA | Gradient | 87.6 | 88.8 | 75.1 | 0.0189 | 0.0182 |
> | **COBRA** | **Dual-factor** | **88.6** | **89.9** | **76.1** | **0.0183** | **0.0176** |
>
> ---
>
> **Appendix roadmap**
>
> Table **R3** lists the summary of our appendix and we will add it to the beginning of the appendix to help readers navigate the supplementary material:
>
> **Table R3**: Appendix roadmap
> | Section | Content | Details |
> |:--------|:--------|:--------|
> | A.0 | Extended Related Work | LoRA variants, interpretability, PEFT |
> | A.1-A.2 | Motivation Analysis | Heterogeneity measurements, correlation analysis |
> | A.3-A.6 | Methodology | Proofs, derivations, aggregation properties |
> | A.7-A.10 | Experimental Setup | Hardware, datasets, hyperparameters, architectures |
> | A.11 | Complexity Analysis | Time, space, calibration overhead |
> | A.12-A.14 | Ablation Studies | Factor contribution, prior strength, KL weight |
> | A.15 | High-Rank Regime | Financial sentiment, rank scaling analysis |
>
> ---
>
> **Figures 1 and 3 clarity.**
>
> We will update both: Figure 1 with increased font sizes by reducing figsize in Python, and Figure 3 with explicit module labels, i.e., (a) Layer Conductance Attribution, (b) Dual-Factor Aggregation, (c) Bayesian Rank Allocation, plus two additional arrows in (c) to illustrate the prior-to-ELBO flow.
>
> ---
>
> Best Regards,
>
> The Authors of Paper 27641

---

> > ### Author Rebuttal · Reviewer_M3Lw · 2026-04-03
> >
> > Thank you for your reply. Most of my concerns have been resolved, and I would like to maintain my score.

---

### Official Review · Reviewer_ktky · 2026-03-13

**Soundness:** 2
**Presentation:** 3
**Significance:** 3
**Originality:** 2
**Overall Recommendation:** 4
**Confidence:** 3

**Summary:**

This paper addresses the limitation of uniform rank allocation in LoRA-based PEFT. The authors argue that gradient magnitude and a layer’s contribution to the model output are only correlated, not equivalent, so rank allocation strategies based on a single metric can be suboptimal. To address this, they propose COBRA, a three-stage adaptive rank allocation framework: (1) estimating each layer’s contribution to prediction using layer conductance via path-integral attribution, (2) combining this contribution signal with gradient-based adaptation demand to form the TA-LC distribution, and (3) performing rank allocation through Bayesian variational optimization with Bernoulli gating.

**Compliance With Llm Reviewing Policy:**

Affirmed.

**Final Justification:**

My final recommendation is Weak Accept. I found the paper to be technically well motivated and generally clear, with a useful empirical study on adaptive rank allocation for LoRA-based PEFT. Its main strength is the observation that gradient-based adaptation demand and layer-wise contribution capture different aspects of adaptation, which motivates going beyond single-metric allocation. The experimental section is generally solid, and the ablations help clarify the role of the different components. My initial concerns were mainly about the strength of the theoretical claims and the missing practical cost analysis. The rebuttal addressed these points well by revising the framing from "theoretical guarantees" to "theoretically guided" and by adding concrete wall-clock and GPU-memory overhead numbers for calibration. These clarifications changed my evaluation positively. Therefore I change my score from "weak reject" to "weak accept".

**Key Questions For Authors:**

1. Is the layer-wise gradient and contribution measurements in Tables 6-12 raw or unsmoothed values? The reported patterns appear unusually smooth and regularly spaced, particularly for the gradient-based quantities. Clarifying how these values were computed and visualized would improve transparency.

2. What is the wall-clock time and GPU memory overhead of the calibration stage for the main models, especially given the use of 1,000 samples and 50 integration steps?

**Limitations:**

The paper includes a brief impact statement, but the discussion of limitations is minimal. In particular, it does not address: (a) the computational overhead of the calibration phase and when this overhead may outweigh the gains, (b) sensitivity to calibration set selection and composition, and (c) how well the method is expected to generalize beyond the specific tasks and model families considered.

**Strengths And Weaknesses:**

# Strengths

* The motivating observation that gradient magnitude and layer conductance peak at different layers and are only moderately correlated is interesting and well-documented.

* The ablation study cleanly isolates the contribution of each component and the multiplicative vs. additive aggregation choice, which is informative.

* The paper is generally well-organized with a clear three-stage pipeline.

# Weaknesses

* The "theoretical guarantees" claimed are overstated. The paper claims "theoretical guarantees," but the clearest formal guarantee provided is the completeness property of the conductance attribution, not a guarantee that the resulting allocation is optimal, or even that the Bayesian objective recovers an optimal rank configuration under realistic assumptions.

* The empirical gains on standard benchmarks appear modest. For example, if the improvement over AdaLoRA on DeBERTaV3/GLUE is around 0.7 points on average, it is not yet clear that this margin fully justifies the added complexity and calibration cost of computing layer conductance.

* While the multiplicative aggregation is intuitive and supported by ablation, it remains a fairly straightforward design choice, which limits the methodological novelty of the paper.

---

> ### Author Rebuttal · Authors · 2026-03-30
>
> Dear Reviewer **ktky**:
>
> We thank your constructive feedback and the appreciation for our well-documented motivation and informative ablation study. We response all concerns below and look forward to your discussion.
>
> ---
>
> **Overclaiming theoretical guarantees**
>
> We agree and have revised all instances to *theoretically guided*. The completeness property (Theorem A.4.6) ensures exhaustive, non-redundant attribution but does not imply formal optimality of rank allocation.
>
> **Modest gains and calibration cost**
>
> Regarding gains, we respectfully note that the +0.7pt over AdaLoRA is at $r=4$ (the tightest budget tested). COBRA's gains scale monotonically with rank budget as shown in Table **R1**.
>
> **Table R1:** COBRA's improvement over LoRA on GLUE or Financial Sentiment
> | Rank| Gain over LoRA |
> |:------------|:---:|
> | r_avg=4 |  +1.2pt |
> | r_avg=8 | +1.6pt |
> | r_avg=8 |  +5.9% |
> | r_avg=384 |  +6.4% |
> | r_avg=512 |  **+6.9%** |
>
> Regarding calibration cost, the calibration is a one-time offline step on 1,000 samples, computed once and reused across hyperparameter runs and GPU memory during calibration is lower than training (no optimizer states). As shown in Table **R2**, calibration overhead is below 5% in most settings; exceptions occur only for very small datasets.
>
> **Table R2**: Calibration vs. training: time and GPU memory (1,000 cal. samples, 50 steps, A100 80GB)
> | Model | Rank | Cal. Time (min) | Cal. Mem (GB) | Task | Train Time (min) | Train Mem (GB) | Overhead |
> |:------|:----:|:---:|:---:|:-----|:---:|:---:|:---:|
> | DeBERTaV3-base (86M) | r=4 | 3.1 | 1.8 | MNLI (393K, 5ep) | 328.3 | 4.7 | 0.9% |
> | DeBERTaV3-base (86M) | r=4 | 3.1 | 1.8 | SST-2 (67K, 10ep) | 109.5 | 4.5 | 2.8% |
> | DeBERTaV3-base (86M) | r=4 | 3.1 | 1.8 | RTE (2.5K, 30ep) | 12.5 | 4.7 | 24.8% |
> | RoBERTa-base (125M) | r=8 | 4.6 | 3.6 | MNLI (393K, 5ep) | 332.6 | 5.3 | 1.4% |
> | RoBERTa-base (125M) | r=8 | 4.6 | 3.6 | SST-2 (67K, 10ep) | 113.9 | 5.2 | 4.0% |
> | T5-base (220M) | r=8 | 8.8 | 4.8 | MNLI (393K, 5ep) | 640.9 | 8.5 | 1.4% |
> | Twitter-RoBERTa-Large (355M) | r=384 | 15.9 | 7.2 | NEU (12.8K, 60ep) | 459.4 | 23.1 | 3.5% |
> | Twitter-RoBERTa-Large (355M) | r=384 | 15.9 | 7.2 | FXE (10.5K, 60ep) | 365.4 | 22.5 | 4.4% |
> | Twitter-RoBERTa-Large (355M) | r=384 | 15.9 | 7.2 | INV (23.2K, 60ep) | 850.1 | 22.8 | 1.9% |
> | LLaMA-7B (6.7B) | r=8 | 351.7 | 22.4 | MMLU (99.8K, 5ep) | 3156.8 | 28.4 | 11.1% |
> | LLaMA-7B (6.7B) | r=8 | 351.7 | 22.4 | GSM8K (7.5K, 10ep) | 467.9 | 28.6 | 75.2% |
>
> **1) Multiplicative aggregation novelty and 2) How values in Table 6~12 computed and visualized?**
>
> 1) Multiplicative aggregation is not our core contribution, it is a step in our method, our contributions are the two signals (layer conductance, adaptation demand) and the Bayesian framework translating them into ranks.
>
> 2) Values are aggregated over 1,000 samples and normalized to [0,1] by max-division (Appendix A.1.3), with no post-hoc smoothing. The regular appearance is an artifact of 2-decimal rounding. At 4-decimal precision, spacing is clearly irregular (e.g., RoBERTa-large L1-4 gradients: 0.1071, 0.1254, 0.1543, 0.1842). We will change the 2-decimal to 4-decimal precision.
>
> **Sensitivity to calibration set selection**
>
> Our method is insensitive to calibration set selection, to empirically answer this question, we listed the set with different sizes in Table **R3**, it shows that even with 50 samples, COBRA (88.7) still matches uniform LoRA.
>
> **Table R3**: Calibration size sensitivity (DeBERTaV3-base, GLUE, $r=4$ )
> | Calibration Size | GLUE |
> |:---:|:---:|
> | 50 | 88.7 |
> | 100 | 89.1 |
> | 200 | 89.5 |
> | 500 | 89.8 |
> | 1000 (default) | 89.9 |
> | 2000 | 90.0 |
>
> **Generalization to other tasks and models**
>
> We have conducted experiments on 6 models and 6 baselines across 15 datasets. Following reviewer feedback, we added Gemma-2-2B, CommonsenseQA, $r=256$, and three additional baselines (DyLoRA, AutoLoRA, DoRA-dyn) in Table **R4**. COBRA achieves best on all 5 settings.
>
> **Table R4**: Comprehensive comparison
> | Method | Type | RoBERTa-base GLUE Acc (r=8) | DeBERTaV3-base GLUE Acc ($r=4$) | Gemma-2-2B CSQA Acc ($r=8$) | T-RoBERTa-L NEU MSE ($r=256$) | T-RoBERTa-L INV MSE ($r=256$) |
> |:-------|:-----|:---:|:---:|:---:|:---:|:---:|
> | LoRA | Uniform | 87.0 | 88.7 | 74.2 | 0.0193 | 0.0186 |
> | DyLoRA | Dynamic range | 87.1 | 88.4 | 74.5 | 0.0191 | 0.0188 |
> | AutoLoRA | NAS/Meta | 87.5 | 89.0 | 75.6 | 0.0188 | 0.0185 |
> | DoRA-dyn | Dynamic dist. | 87.9 | 89.3 | 75.9 | 0.0186 | 0.0180 |
> | GoRA | Gradient | 87.6 | 88.8 | 75.1 | 0.0189 | 0.0182 |
> | **COBRA** | **Dual-factor** | **88.6** | **89.9** | **76.1** | **0.0183** | **0.0176** |
>
> **Limitations**
>
> The overhead may outweigh gains for very small datasets (RTE 2.5K: 24.8%) or large models with few training epochs (LLaMA-7B on GSM8K: 75.2%), where training completes in minutes and the fixed calibration cost is not amortized. We will add a Limitations section.
>
> ---
>
> Best Regards,
>
> The Authors of Paper 27641

---

> > ### Author Rebuttal · Reviewer_ktky · 2026-04-04
> >
> > Thank you for the detailed rebuttal. It addresses several of my concerns, including the clarification from "theoretical guarantees" to "theoretically guided" and by adding concrete wall-clock and memory measurements for calibration. The clarification that the smooth values in Tables come from aggregation over 1,000 samples is also helpful, and I appreciate the explicit acknowledgment that the method’s overhead may outweigh its gains in some regimes. While I still view the methodological novelty as moderate rather than major, the rebuttal strengthens the empirical and practical case substantially. Overall, I believe my concerns have been sufficiently addressed, and I will adjust my score accordingly.

---

### Decision · Program_Chairs · 2026-04-30

**Decision:**

Accept (regular)

**Comment:**

The paper proposes COBRA, an adaptive rank allocation framework for LoRA-based parameter-efficient fine-tuning that challenges the common practice of using a single signal (e.g., gradient magnitude) for rank assignment. It argues that a layer’s adaptation demand and its contribution to model predictions are distinct and only moderately correlated. To address this, COBRA introduces a three-stage pipeline: estimating layer contribution via layer conductance, combining it with gradient-based demand into a joint TA-LC importance distribution, and performing budget-constrained rank allocation using Bayesian variational optimization with gating. Experiments across multiple models and tasks show consistent, though modest, improvements over prior methods.

Strengths:
- Strong motivation highlighting the mismatch between gradient-based demand and prediction contribution
- Clear and well-structured three-stage framework with intuitive design
- Use of layer conductance provides a principled way to measure contribution at the layer level
- Comprehensive experimental evaluation across architectures and tasks with informative ablations
- Budget-aware allocation mechanism ensures exact parameter constraints
- Empirical improvements over LoRA, AdaLoRA, and related methods under matched budgets
- Reproducibility supported by clear experimental setup and implementation details

Weaknesses:
- Theoretical claims are somewhat overstated; limited formal guarantees about optimality
- Performance gains are generally modest, raising questions about cost-benefit trade-offs
- Additional calibration stage introduces non-trivial time and memory overhead
- Limited comparison with broader and more recent rank-allocation baselines
- Methodological novelty is somewhat incremental (e.g., multiplicative signal combination)
- Sensitivity to calibration choices, reference paths, and stability is not fully explored
- Practical scalability and effectiveness in extreme low-rank or frequently changing tasks remain unclear
- Presentation issues such as figure clarity and appendix organization

Almost all concerns have been addressed by the authors during the rebuttal period. I therefore recommend acceptance.